# Learning Time-Aware Causal Representation for Model Generalization in Evolving Domains

**Zhuo He** [1]   **Shuang Li** [1]   **Wenze Song** [1]   **Longhui Yuan** [1]   **Jian Liang** [2]   **Han Li** [2]   **Kun Gai** [2]

## Abstract

Endowing deep models with the ability to generalize in dynamic scenarios is of vital significance for real-world deployment, given the continuous and complex changes in data distribution. Recently, evolving domain generalization (EDG) has emerged to address distribution shifts over time, aiming to capture evolving patterns for improved model generalization. However, existing EDG methods may suffer from spurious correlations by modeling only the dependence between data and targets across domains, creating a shortcut between task-irrelevant factors and the target, which hinders generalization. To this end, we design a time-aware structural causal model (SCM) that incorporates dynamic causal factors and the causal mechanism drifts, and propose **S**tatic-**DYN**amic **C**ausal Representation Learning (**SYNC**), an approach that effectively learns time-aware causal representations. Specifically, it integrates specially designed information-theoretic objectives into a sequential VAE framework which captures evolving patterns, and produces the desired representations by preserving intra-class compactness of causal factors both across and within domains. Moreover, we theoretically show that our method can yield the optimal causal predictor for each time domain. Results on both synthetic and real-world datasets exhibit that SYNC can achieve superior temporal generalization performance. 🔗

## 1. Introduction

Deep neural networks (DNNs) have achieved excellent performance in various applications, yet suffer from performance degradation when the *i.i.d.* assumption is violated (Recht et al., 2019; Taori et al., 2020). Domain Gen-

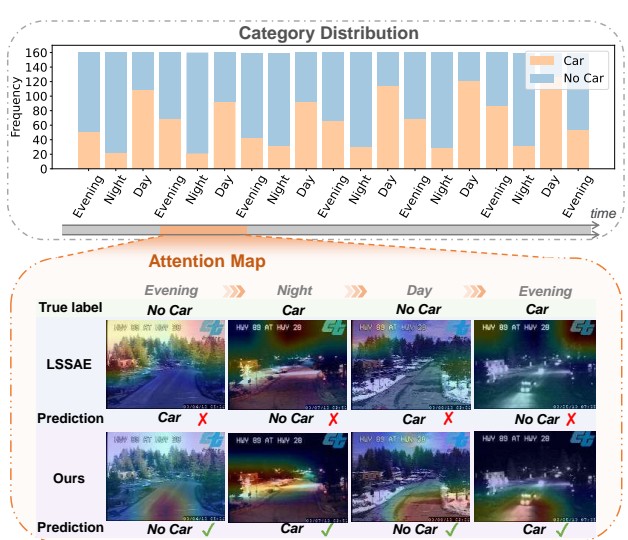

Figure 1: We plot the category distribution of training domains on Caltran. The lower part shows visualization of attention maps of the convolutional Layer. Compared with LSSAE, it can be observed that our method better focuses on target-related features and makes accurate predictions.

eralization (DG) (Liu et al., 2021b; Wang et al., 2023; Fan et al., 2024) can effectively address the issue of distribution shift, but it still struggles to handle the dynamic scenarios that are widespread in the real world (Yao et al., 2022a; Qin et al., 2022). In order to adapt to changing environments over time, evolving domain generalization (EDG) (Nasery et al., 2021; Qin et al., 2022; Zeng et al., 2023b; Bai et al., 2023; Xie et al., 2024; Zeng et al., 2023a; Qin et al., 2023) has emerged in recent years and is garnering increasing attention, aiming to capture the underlying evolving patterns in data distribution to generalize well to the near future.

Despite promising results, existing EDG methods may suffer from spurious correlations by solely modeling the statistical correlation between data and targets. Fig. 1 illustrates the behavior of the baseline model (Qin et al., 2022) trained on the Caltran dataset (Hoffman et al., 2014), tasked with determining whether a vehicle is present in an image. Among the images captured by the camera, most of those containing vehicles were taken during the day, while images without vehicles were captured at evening or night. Therefore, there

[1]Independent Researcher, China [2]Kuaishou Technology, China. Correspondence to: Shuang Li <shuangliai@buaa.edu.cn>.

*Proceedings of the 42nd International Conference on Machine Learning*, Vancouver, Canada. PMLR 267, 2025. Copyright 2025 by the author(s).

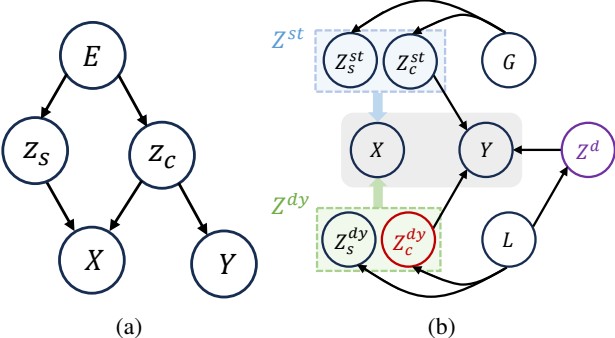

Figure 2: (a) SCM for DG. (b) Our proposed time-aware SCM for EDG. By introducing $Z_c^{dy}$ and $Z^d$, dynamic causal factors and causal mechanism drifts are explicitly modeled.

exists a shortcut from the background representation to the target, *i.e.*, the trained model tends to rely on lighting (spurious feature) to determine the presence of a vehicle, rather than the object of the vehicle (causal feature). Due to spurious correlations, the model during the inference stage tends to classify day (night) images as "Car" ("No Car"), which compromises its generalization performance.

Causality (Pearl, 2009; Pearl et al., 2016) has proven to be a powerful tool for addressing spurious correlations and has therefore been widely explored. (Arjovsky et al., 2019; Mahajan et al., 2021; Sun et al., 2021; Lv et al., 2022; Sheth et al., 2022). However, since causal factors and causal mechanisms behave more complex and changeable in non-stationary environments, its implementation in dynamic scenarios still remains underexplored. Fig. 2a shows a classical structural causal model (SCM) for static DG (Liu et al., 2021a; Sun et al., 2021; Lv et al., 2022), where category-irrelevant (spurious) factors $Z_s$ and category-related (causal) factors $Z_c$ are modeled as time-invariant. Such a temporal homogeneous SCM fails to account for the underlying continuous structure in non-stationary tasks, rendering static causal representations insufficient for generalizing to evolving new domains. Moreover, as the statistical properties of the target variables change over time (Gama et al., 2014; Lu et al., 2018), the influence of causal factors on the target may also vary, suggesting the mapping from causal factors to labels gradually shifts, leading to causal mechanism drifts.

To address the aforementioned issues, we propose a novel time-aware SCM for EDG. As shown in Fig. 2b, the local variable $L$ is introduced to model instability and dynamics, and the global variable $G$ is utilized to represent globally unchanging knowledge. We then split the causal factors into static causal factors $Z_c^{st}$ and time-varying dynamic causal factors $Z_c^{dy}$, thereby incorporating the evolving patterns into causal factors. To model the causal mechanism drifts, we introduce a time-varying drift factor $Z^d$. Moreover, $Z_c^{st}$ and $Z_c^{dy}$, along with the static spurious factors $Z_s^{st}$ and dynamic spurious factors $Z_s^{dy}$, constitute the static factors $Z_s^{st}$

and dynamic factors $Z^{dy}$, respectively. Therefore, in our new SCM, the data $X$ is generated by $X \leftarrow (Z^{st}, Z^{dy})$ while the label $Y$ is generated via $Y \leftarrow (Z_c^{st}, Z_c^{dy}, Z^d)$. It can be observed that the time factor acts as a latent confounder, with its components $G$ and $L$ forming backdoor paths between static and dynamic spurious factors and the target. These paths introduce spurious correlations, which can be mitigated by identifying the true causal factors and constructing a causal model accordingly.

In order to learn the causal model with the help of the time-aware SCM, we propose **S**tatic-**DYN**amic **C**ausal Representation Learning (**SYNC**), a method designed to effectively capture time-aware causal representations by simultaneously uncovering static and dynamic causal representations. Concretely, a sequential VAE framework is employed to capture the underlying evolving patterns in data distribution. To separate causal factors from the complex and entangled mixture of factors, we first minimize the mutual information (MI) loss between static and dynamic factors to encourage disentanglement between them. After that, static and dynamic causal factors can be obtained by maximally preserving the intra-class compactness of causal factors both across and within temporal domains, implemented by optimizing the designed cross-domain and intra-domain conditional MI losses that account for continuous domain structure in EDG. We theoretically show that, with the devised objectives, our method can yield the optimal causal predictor for each time domain, ultimately excluding spurious correlations and achieving superior temporal generalization performance.

**Contributions**: **(i)** By taking a novel causal perspective towards EDG problem, we design a time-aware SCM that enables the refined modeling of both dynamic causal factors and causal mechanism drifts. After that, we propose SYNC, an approach for effectively learning time-aware causal representations, thereby mitigating spurious correlations. **(ii)** Theoretically, we show that SYNC can build the optimal causal predictor for each time domain, resulting in improved model generalization. **(iii)** Results on both synthetic and real-world datasets, along with extensive analytic experiments demonstrate the effectiveness of proposed approach.

## 2. Related Work

**Evolving Domain Generalization (EDG)** is introduced to address the issue of generalizing across temporally drift domains (Nasery et al., 2021; Qin et al., 2022; Zeng et al., 2023b; Bai et al., 2023; Xie et al., 2024; Zeng et al., 2024; 2023a; Qin et al., 2023; Jin et al., 2024). Most existing methods learn a time-sensitive model. To name a few, GI (Nasery et al., 2021) suggests training a time-sensitive model by utilizing a gradient interpolation loss. LSSAE and MMD-LSAE (Qin et al., 2022; 2023) account for covariate shifts and concept drifts and introduce a probabilistic

model with variational inference to capture evolving patterns. MISTS (Xie et al., 2024) further proposes to learn both invariant and dynamic features through mutual information regularization. Additionally, DRAIN (Bai et al., 2023) launches a Bayesian framework and adaptively generates network parameters through dynamic graphs. SDE-EDG (Zeng et al., 2023a) proposes to learn continuous trajectories captured by the stochastic differential equations. Different from the aforementioned methods, DDA (Zeng et al., 2023b) simulates the unseen target data by a meta-learned domain transformer. However, these methods do not account for spurious correlations, which could hinder the model generalization. In this work, we introduce a novel causal perspective to EDG problem and learn time-aware causal representations to address this issue.

**Causality and Generalization.** Causality describes the fundamental relationship between cause and effect, which goes beyond the statistical correlation simply obtained from observational data (Pearl, 2009; Pearl et al., 2016). Since causal models can effectively mitigate the impact of spurious correlations, many studies have proposed leveraging causal theories to develop more robust models. Causal methods in DG can be broadly categorized into two types. The first type of methods (Arjovsky et al., 2019; Ahuja et al., 2020; Li et al., 2022; Ahuja et al., 2021; Sun et al., 2021; Yong et al., 2024) focuses on constructing an optimal invariant predictor through invariant learning. However, by neglecting dynamic causal information or the correlations between temporal domains, these methods struggle to perform effectively in dynamic environments. The second type (Mahajan et al., 2021; Lv et al., 2022; Wang et al., 2022; Hu et al., 2022; Mao et al., 2022; Chen et al., 2023; Yue et al., 2021) leverages interventions to extract causal features. Nevertheless, in dynamic scenarios, the presence of dynamical causal factors complicates the rational intervention in spurious factors. Regarding causal methods for time series modeling (Entner & Hoyer, 2010; Malinsky & Spirtes, 2018; Li et al., 2021; Yao et al., 2022b), they often need somewhat strong assumptions on models such as the reversibility of the generation function to infer underlying causal structures. Furthermore, the aforementioned methods overlook the fact that the causal mechanisms of the target may drift (Lu et al., 2018) in dynamic scenarios.

# 3. Methodology

## 3.1. Preliminaries

**Evolving Domain Generalization.** Let $x \in \mathcal{X}$ and $y \in \mathcal{Y}$ denote the input data and its label, respectively, where $\mathcal{X}$ and $\mathcal{Y}$ represent the nonempty input space and label space, respectively. $P(x, y, t)$ characterizes the temporal dynamic of the probability distribution for $(x, y)$, wherein underlying evolving patterns over time $t$ exist. Suppose that we are

given $T$ sequential source domains $\mathcal{S} = \{\mathcal{D}_t\}_{t=1}^{T}$, where each domain $\mathcal{D}_t$ consists of $N_t$ data $(x_{t,i}, y_{t,i})$ which are drawn from $P(x, y|t)$, i.e., $\mathcal{D}_t = \{(x_{t,i}, y_{t,i})\}_{i=1}^{N_t}$. The goal of EDG is to leverage given $T$ training domains to establish a robust model $h : \mathcal{X} \to \mathcal{Y}$ which can generalize well to $K$ unseen target domains $\mathcal{T} = \{\mathcal{D}_t\}_{t=T+1}^{T+K}$ arriving sequentially in the near future. Due to the existence of evolving patterns across the sequential domains, the ideal model should be capable of capturing these patterns.

**Structural Causal Model.** The structual causal model (SCM) over $n$ random variables $X_1, X_2, \cdots, X_n$, according to (Pearl, 2009), is defined as a triple $\langle \mathcal{G}, P(\epsilon), \mathcal{F} \rangle$. $\mathcal{G}$ denotes the causal directed acyclic graph (DAG) which is comprised of the nodes $V$ and the directed edges $E$, where the starting point and end point of each directed edge represent cause and effect, respectively. $\epsilon$ is the independent exogenous noise variables. These noise variables follow the joint distribution $P(\epsilon)$. $\mathcal{F}$ denotes a set of assignments $\{X_i := f_i(\mathbf{PA}_i, \epsilon_i)\}_{i=1}^{n}$, where $\mathbf{PA}_i$ is the set of parents of $X_i$, i.e., direct causes, $f_i$ represents a deterministic function. Each SCM induces a corresponding causal DAG, and the data generation process can be characterized by the SCM.

## 3.2. Temporally Evolving SCM

Spurious correlations may mislead the model into learning features that are irrelevant to the target, posing a notable challenge to model generalization in EDG. Since causal models can effectively alleviate spurious correlations, we aim to construct an appropriate SCM for EDG and acquire the causal model by learning causal representations.

Given that causal-based methods (Lv et al., 2022; Liu et al., 2021a; Lu et al., 2021; Sun et al., 2021) have been extensively explored in static DG, we start by briefly reviewing the SCM employed in static DG. As mentioned in Section 1, the target $Y$ is determined solely by causal factors $Z_c$, and the causal mechanism $P(Y|Z_c)$ remains consistent across environments. On this basis, a causal model can be established by extracting $Z_c$ and constructing the corresponding invariant causal mechanisms. Nevertheless, such time-independent SCM is not suitable for dynamic scenarios. Specifically, first, ignoring the underlying continuous structure in non-stationary tasks limits performance. Second, the statistical properties of the target may change over time (Gama et al., 2014; Lu et al., 2018), implying that the influence of causal factors on the target may vary, resulting in causal mechanism drifts. Therefore, to apply causality in EDG, designing a new and suitable SCM is essential.

Regarding the first issue, we further partition the causal factors into static causal factors $Z_c^{st}$ and dynamic causal factors $Z_c^{dy}$, where $Z_c^{st}$ contains stable category-related information, while $Z_c^{dy}$ is related to domain-specific category-related information. $Z_c^{st}$ and $Z_c^{dy}$, along with the static

spurious factors $Z_s^{st}$ and dynamic spurious factors $Z_s^{dy}$, constitute the static factors $Z^{st}$ and dynamic factor $Z^{dy}$, respectively. Then, $Z^{st}$ and $Z^{dy}$ collaborate to generate the observed data $X$. Regarding the second issue, in order to characterize the evolving causal structure $P(Y|Z_c^{st}, Z_c^{dy})$, we introduce drift factors $Z^d$ to characterize the evolution of causal mechanisms. Finally, we introduce the local variable $L$, which consist of unstable and dynamic information, along with the stable global variable $G$, to represent the generation of $Z^{st}$, $Z^{dy}$ and $Z^d$. Combined with the above analysis, we propose the time-aware SCM as follows:

**Definition 1** (Time-aware SCM). The time-aware SCM $\mathcal{M}_T$ built on observed variables $X, Y$, latent variables $Z_c^{st}, Z_s^{st}, Z_c^{dy}, Z_s^{dy}, Z^d$ and information factors $G$ and $L$, is a triple, *i.e.*, $\mathcal{M}_T = \langle \mathcal{G}, P(\epsilon), \mathcal{F} \rangle$, $\mathcal{G}$ is the corresponding directed acyclic causal graph shown in Fig. 2b, $P(\epsilon)$ denotes the distribution that exogenous noise follows, and $\mathcal{F}$ denotes the assignments $\{f^x, f^y, f_c^{st}, f_s^{st}, f_c^{dy}, f_s^{dy}, f^d\}$:

$$X := f^x(Z_c^{st}, Z_s^{st}, Z_c^{dy}, Z_s^{dy}, \epsilon_x), Y := f^y(Z_c^{st}, Z_c^{dy}, Z^d, \epsilon_y),$$
$$Z_c^{st} := f_c^{st}(G, \epsilon_c^{st}), Z_s^{st} := f_s^{st}(G, \epsilon_s^{st}),$$
$$Z_c^{dy} := f_c^{dy}(L, \epsilon_c^{dy}), Z_s^{dy} := f_s^{dy}(L, \epsilon_s^{dy}), Z^d := f^d(L, \epsilon_d).$$

From Def. 1, the generation of the observed variables is fully determined by five latent factors, with no additional unobserved factors involved. Under the global Markov condition and the causal faithfulness (Pearl et al., 2016), it follows that $Y \perp\!\!\!\perp [Z_s^{st}, Z_s^{dy}] \mid [Z_c^{st}, Z_c^{dy}, Z^d]$. This observation suggests that, within a given time domain $\mathcal{D}_t$, predictors built on causal representations are immune to the influence of spurious factors. By incorporating a mutual information (MI) term to filter out information irrelevant to the target, the optimal causal predictor for $\mathcal{D}_t$ can be defined as follows:

**Definition 2** (Optimal Causal Predictor). For a time domain $\mathcal{D}_t$, the corresponding drift factors $Z_t^d$ are given. Let $Z_{c,t} = (Z_{c,t}^{st}, Z_{c,t}^{dy})$ be the causal factors in domain $\mathcal{D}_t$ that satisfy $Z_{c,t} \in \arg\max_{Z_{c,t}} I(Y; Z_{c,t}, Z_t^d)$, and $Y \perp\!\!\!\perp [Z_{s,t}^{st}, Z_{s,t}^{dy}] \mid [Z_{c,t}, Z_t^d]$. Then the predictor based on factors $Z_{c,t}$ and $Z_t^d$ is the optimal causal predictor.

From Def. 2, when static and dynamic causal factors, along with drifts in causal mechanisms, are modeled together, the causal model for each time domain can be accurately learned. Building on the time-aware SCM $\mathcal{M}_T$, we formulate a temporally evolving SCM $\mathcal{M}_{evo}$, which is composed of $\mathcal{M}_T$ as the basic element, connected sequentially along the time axis. As shown in Fig. 3, static factors remain invariant to temporal changes, whereas dynamic and drift factors evolve according to specific patterns.

Existing causal methods developed for static DG suffer from the following two problems in non-stationary environments: **(i)** The presence of dynamic causal factors complicates the

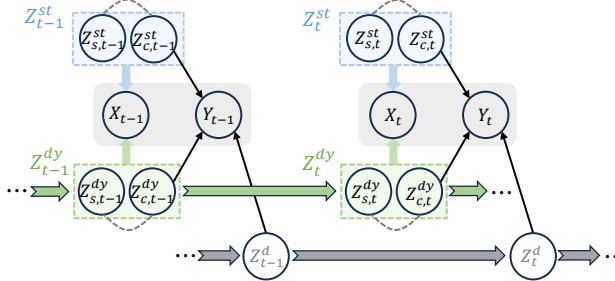

Figure 3: Temporally Evolving SCM for EDG. The dashed line indicates that the two factors are correlated.

rational intervention in spurious factors; **(ii)** Learning with invariant causal mechanisms, as in a static environment, is not feasible. To address these issues, we propose **S**tatic-**DYN**amic **C**ausal Representation Learning (**SYNC**), which effectively learns time-aware causal representations composed of both static and dynamic causal components.

### 3.3. Static-Dynamic Causal Representation Learning

#### 3.3.1. EVOLVING PATTERN LEARNING

We consider the evolving patterns of both static and dynamic latent factors. For data $(x_t, y_t)$ collected at time stamp $t$, let $z_t^{st}$ and $z_t^{dy}$ denote the static factors and dynamic factors, respectively. Our goal is to learn the latent evolving patterns by modeling the condition distributions $p(z_t^{st}|x_t)$ and $p(z_t^{dy}|z_{<t}^{dy}, x_t)$. To this end, a neural network $q_\theta(z_t^{dy}|z_{<t}^{dy}, x_t) = \mathcal{N}(\mu(z_t^{dy}), \sigma^2(z_t^{dy}))$ is employed to approximate $p(z_t^{dy}|z_{<t}^{dy}, x_t)$. This approximation is achieved by minimizing $\mathbb{D}_{KL}(q_\theta(z_t^{dy}|z_{<t}^{dy}, x_t), p(z_t^{dy}|z_{<t}^{dy}, x_t))$, which is equivalent to maximizing

$$\mathbb{E}_{q_\theta} \log p(x_t|z_t^{dy}) - \mathbb{D}_{KL}(q_\theta(z_t^{dy}|z_{<t}^{dy}, x_t), p(z_t^{dy}|z_{<t}^{dy})),$$

where $p(z_t^{dy}|z_{<t}^{dy})$ is the prior distribution of $z_t^{dy}$. Inspired by (Qin et al., 2022), we model it using LSTM (Hochreiter & Schmidhuber, 1997), by setting the initial state as $z_0^{dy} = 0$. For $p(z_t^{st}|x_t)$, we utilize a network $q_\psi(z_t^{st}|x_t) = \mathcal{N}(\mu(z_t^{st}), \sigma^2(z_t^{st}))$ to approximate. Analogously, to minimize $\mathbb{D}_{KL}(q_\psi(z_t^{st}|x_t), p(z_t^{st}|x_t))$, we need to maximize

$$\mathbb{E}_{q_\psi} \log p(x_t|z_t^{st}) - \mathbb{D}_{KL}(q_\psi(z_t^{st}|x_t), p(z_t^{st})),$$

where $p(z_t^{st})$ is set as $\mathcal{N}(0, I)$. In general, the objective function for learning the latent patterns of features is

$$\mathcal{L}_{\text{fp}} = -\sum_{t=1}^{T} \mathbb{E}_{q_\theta q_\psi}[\log p(x_t|z_t^{st}, z_t^{dy})]$$
$$+ \sum_{t=1}^{T} \mathbb{D}_{KL}(q_\psi(z_t^{st}|x_t), p(z_t^{st})) \quad (1)$$
$$+ \sum_{t=1}^{T} \mathbb{D}_{KL}(q_\theta(z_t^{dy}|z_{<t}^{dy}, x_t), p(z_t^{dy}|z_{<t}^{dy})),$$

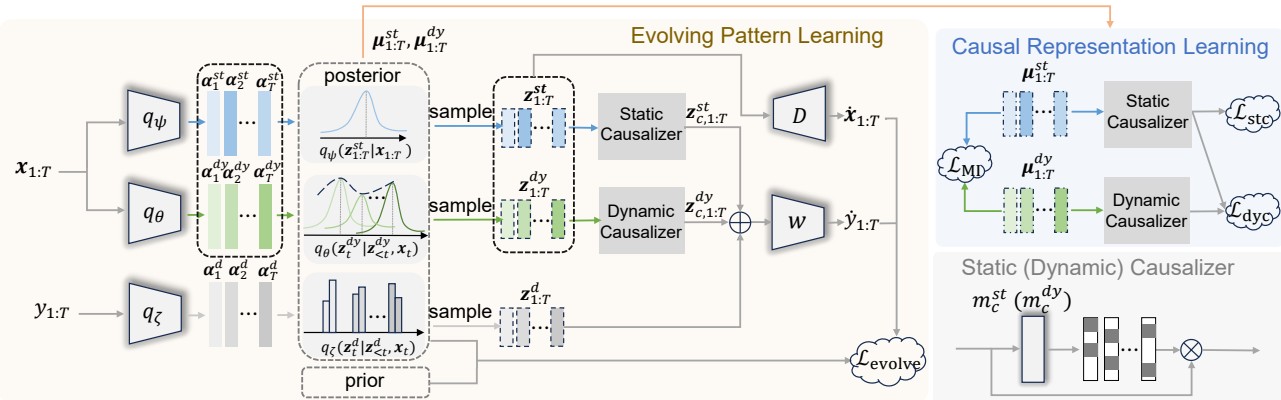

Figure 4: The training framework of SYNC. We employ $q_\psi, q_\theta, q_\zeta$ to model the posteriors, where $\alpha_i^\cdot = (\boldsymbol{\mu}_i^\cdot, \boldsymbol{\sigma}_i^{2,\cdot})$, $\boldsymbol{\mu}, \boldsymbol{\sigma}^2$ is the mean and variance of the posterior, · denotes "st", "dy" or "d". Then, the sampled $z_{1:T}^{st}$ and $z_{1:T}^{dy}$ are utilized to reconstruct the data through the decoder $D$. After passing the corresponding causalizer (in which $m_c^{st}$ or $m_c^{dy}$ is a 0-1 masker), they are combined with $z_{1:T}^d$ to make predition through the classifier $w$. The reconstruction term and the divergency between the priors and the posteriors is used to calculate the evolving pattern loss $\mathcal{L}_{\text{evolve}}$. The means of the static and dynamic posteriors are utilized to calculate MI losses $\mathcal{L}_{\text{MI}}, \mathcal{L}_{\text{stc}}, \mathcal{L}_{\text{dyc}}$, aiming to obtain static and dynamic causal factors.

where the log-likelihood term represents the reconstruction term for input $x_t$, two KL divergence terms are used to align the posterior distributions of $z_t^{st}$ and $z_t^{dy}$ with the corresponding prior distributions.

### 3.3.2. DISENTANGLEMENT OF STATIC AND DYNAMIC REPRESENTATIONS

Although static and dynamic factors are modeled separately in Eq. (1), the absence of additional constraints prevents us from guaranteeing that static factors do not contain dynamic information, the same is true for dynamic factors. MI quantifies the degree of dependence between two variables. By minimizing the MI, the exclusivity of the information contained in these variables can be promoted. Therefore, in order to achieve the disentanglement of $z_t^{st}$ and $z_t^{dy}$, we propose to minimize the MI between them. The MI between $z_t^{st}$ and $z_t^{dy}$ can be written as $I(z_t^{st}; z_t^{dy}) = H(z_t^{st}) + H(z_t^{dy}) - H(z_t^{st}, z_t^{dy})$. Following (Chen et al., 2018b), we use a standard mini-batch weighted sampling (MWS) to estimate the entropy term $H(\cdot) = -\mathbb{E}_{q(\cdot)}[\log q(\cdot)]$. Specifically, when provided with a mini-batch of samples $\{x_t^1, x_t^2, \cdots, x_t^B\}$ at time $t$, we can use the estimator

$$\mathbb{E}_{q(\cdot)}[\log q(\cdot)] \approx \frac{1}{B} \sum_{i=1}^{B} [\log \frac{1}{BB'} \sum_{j=1}^{B} q(z(x_t^i)|x_t^j)], \quad (2)$$

where $z(x_t^i)$ denotes a sample from $q(\cdot|x_t^i)$, and $B$ and $B'$ denote the batch size and data size, respectively. The MI loss can be written as

$$\mathcal{L}_{\text{MI}} = \sum_{t=1}^{T} I(z_t^{st}, z_t^{dy}). \quad (3)$$

### 3.3.3. UNEARTHING STATIC AND DYNAMIC CAUSAL REPRESENTATIONS

As mentioned in Section 3.2, learning causal representations poses significant challenges. To address this, we need to find some properties of causal factors and translate them into equivalent and feasible objectives. Prop. 1 below gives the properties of causal factors in the continuous time domain:

**Proposition 1.** *Let $\mathcal{D}_t$ be a time domain, and for a given class $Y$, it can be conclude that:*

*(i) If there is $H(Z_{c,t}^{st}|Z_{c,t-1}^{st}, Y) < H(Z_{c,t}^{st}|Z_{s,t}^{st}, Y)$, then $I(Z_{c,t}^{st}; Z_{c,t-1}^{st}|Y) > I(Z_{s,t}^{st}; Z_{c,t-1}^{st}|Y)$.*

*(ii) If there is $H(Z_{c,t}^{st}|Z_{c,t}^{dy}, Y) < H(Z_{c,t}^{st}|Z_{s,t}^{dy}, Y)$, then $I(Z_{c,t}^{dy}; Z_{c,t}^{st}|Y) > I(Z_{s,t}^{dy}; Z_{c,t}^{st}|Y)$.*

In fact, the condition in Prop. 1 is generally easy to meet, as causal factors typically exhibit greater similarity, leading to smaller conditional entropy.

**Remark 1.** Although Prop. 1 appears straightforward, its inspiration comes from $\mathcal{M}_T$ and can be intuitively understood within this framework. Since the two claims are similar, we explain only the second one for simplicity. As shown in Fig. 2b, there exists a collider $Z_c^{st} \to Y \leftarrow Z_c^{dy}$ and a fork $Z_s^{dy} \leftarrow L \to Z_c^{dy}$. According to d-separation (Pearl et al., 2016), when conditioned on $Y$, both $Z_c^{dy}$ and $Z_s^{dy}$ are related to $Z_c^{st}$. Consider a boundary case where $Z_c^{st}$ and $Z_s^{dy}$ are independent, this implies that the backdoor path between $Z_c^{dy}$ and $Z_s^{dy}$ is blocked, leading to independence between $Z_c^{st}$ and $Z_s^{dy}$, while $Z_c^{st}$ remains related to $Z_c^{dy}$. Prop. 1 generalizes this observation, under certain entropy inequalities, static causal factors are more strongly related

to dynamic causal factors than dynamic spurious factors. As mentioned above, the conditions are easy to satisfy.

Prop. 1 (i) provides a way to learn the static causal factors:

$$\max_{\Phi_c^{st}} I(\Phi_c^{st}(X_t); \Phi_c^{st}(X_{t-1})|Y), \tag{4}$$

where $\Phi_c^{st}$ consists of the feature extraction component of $q_\psi$, namely $q_\psi^{ext}$, and a neural-network-based masker $m_c^{st}$, such that $\Phi_c^{st} = m_c^{st} \circ q_\psi^{ext}$. $m_c^{st}$ is implemented by

$$m_c^{st} = \text{Gumbel-Softmax}\left(w_c^{st}(q_\psi^{ext}(X_t)), \kappa N\right), \tag{5}$$

where $w_c^{st}$ learns the scores of each dimensions of $q_\psi^{ext}(X_t)$, and the dimensions correspond to the mask ratio $\kappa \in (0, 1)$ are regarded as causal dimensions, $N$ is the dimension of $q_\psi^{ext}(X_t)$. We employ the Gumbel-Softmax trick (Jang et al., 2017) to sample a mask with $\kappa N$ values approaching 1.

Since an exact estimate of Eq. (4) could be highly expensive (Oord et al., 2018), we use supervised contrastive learning (Khosla et al., 2020; Belghazi et al., 2018) as a practical solution for the approximating $I(\Phi_c^{st}(X_t); \Phi_c^{st}(X_{t-1})|Y)$:

$$\mathbb{E}_{\substack{\{X_{t,j}, X_{t-1,k}^p\} \sim P(X|y=Y) \\ \{X_{t-1,i}^n\}_{i=1}^M \sim P(X|y\neq Y)}} \log \frac{e^{l_{t+1}^{st}(j,k)/\tau}}{e^{l_{t+1}^{st}(j,k)/\tau} + \sum_{i=1}^M e^{l_t^{st}(j,i)/\tau}}, \tag{6}$$

where $l_{t+1}^{st}(j,k) = \text{sim}(\Phi_c^{st}(X_{t,j}), \Phi_c^{st}(X_{t-1,k}^p))$, and $l_t^{st}(j,i) = \text{sim}(\Phi_c^{st}(X_{t,j}), \Phi_c^{st}(X_{t-1,i}^n))$, $\text{sim}(\cdot, \cdot)$ is the cosine similarity. In Eq. (6), the positive sample $X_{t-1,k}^p$ shares the same label as $X_{t,j}$, while the negative samples $\{X_{t-1,i}^n\}_{i=1}^M$ have different labels. $\tau$ is the temperature hyperparameter. In addition, we utilize the MI term $I(Y; \Phi_c^{st}(X_t))$ to filter out information irrelevant to the target. Lem. 1 as follows demonstrates the effectiveness of designed objective.

**Lemma 1.** *For a time domain $\mathcal{D}_t$, the static causal factors can be obtained by solving the following objective:*

$$\max_{\Phi_c^{st}} I(Y; \Phi_c^{st}(X_t)),$$
$$s.t. \ \Phi_c^{st} \in \arg\max_{\Phi_c^{st}} I(\Phi_c^{st}(X_t); \Phi_c^{st}(X_{t-1})|Y), \tag{7}$$

where maximizing the MI term can be achieved by minimizing the cross-entropy loss between the representation $\Phi_c^{st}(X_t)$ and the target $Y$. Prop. 1 (ii) demonstrates that by using the static causal factors $Z_{c,t}^{st}$ as the anchor, the dynamic causal representations can be learned through

$$\max_{\Phi_c^{dy}} I(\Phi_c^{dy}(X_t); Z_{c,t}^{st}|Y), \tag{8}$$

where $\Phi_c^{dy}$ is comprised of the feature extraction component of $q_\theta$, *i.e.*, $q_\theta^{ext}$, and a neural-network-based masker $m_c^{dy}$,

such that $\Phi_c^{dy} = m_c^{dy} \circ q_\theta^{ext}$. $m_c^{dy}$ is implemented is similar to Eq. (5). We estimate $I(\Phi_c^{dy}(X_t); Z_{c,t}^{st}|Y)$ by

$$\mathbb{E}_{\substack{\{X_{t,j}, X_{t,k}^p\} \sim P(X|y=Y) \\ \{X_{t,i}^n\}_{i=1}^M \sim P(X|y\neq Y)}} \log \frac{e^{l_t^{dy}(j,k)/\tau}}{e^{l_t^{dy}(j,k)/\tau} + \sum_{i=1}^M e^{l_t^{dy}(j,i)/\tau}}, \tag{9}$$

where $l_t^{dy}(j,k) = \text{sim}(\Phi_c^{dy}(X_{t,j}), \hat{z}_c^{st}(X_{t,k}^p))$, $l_t^{dy}(j,i) = \text{sim}(\Phi_c^{dy}(X_{t,j}), \hat{z}_c^{st}(X_{t,i}^n))$, with $X_{t,k}^p$ being a positive sample that shares the same label as $X_{t,j}$, and the negative samples $\{X_{t,i}^n\}_{i=1}^M$ are those having different labels. $\hat{z}_c^{st}(X_{t,\cdot})$ denotes the static causal factors of $X_{t,\cdot}$, which are implemented by $\Phi_c^{st}(X_{t,\cdot})$. We provide the following Lem. 2, which demonstrates the effectiveness of devised objective.

**Lemma 2.** *For a time domain $\mathcal{D}_t$, let $\Phi_c^{st,\star}(X_t)$ be the static causal factors learned by the model, then the dynamic causal factors can be obtained by the following objective:*

$$\max_{\Phi_c^{dy}} I(Y; \Phi_c^{dy}(X_t)),$$
$$s.t. \ \Phi_c^{dy} \in \arg\max_{\Phi_c^{dy}} I(\Phi_c^{dy}(X_t); \Phi_c^{st,\star}(X_t)|Y). \tag{10}$$

Overall, the objective in Eq. (4) brings the static causal representations of the same category across domain closer, while the objective in Eq. (8) aligns the static and dynamic causal representations of the same category within the domain.

### 3.3.4. LEARNING EVOLVING CAUSAL MECHANISMS

We introduce drift factors $Z^d$ to model the evolving causal mechanisms. In order to approximate $p(z_t^d|z_{<t}^d, y_t)$, we use a network parameterized by a recurrent neural network $q_\zeta(z_t^d|z_{<t}^d, y_t)$ with a categorical distribution as the output. The initial state is set as $z_0^d = 0$. Then minimizing the KL divergency between $q_\zeta$ and $p$ can be achieved by maximizing

$$\mathbb{E}_{q_\zeta} \log p(y_t|z_t^d) - \mathbb{D}_{KL}(q_\zeta(z_t^d|z_{<t}^d, y_t), p(z_t^d|z_{<t}^d)),$$

where $p(z_t^d|z_{<t}^d)$ is presented as a learnable categorical distribution. Together with the classification loss, the loss function for learning the evolving causal mechanisms is

$$\mathcal{L}_{\text{mp}} = -\sum_{t=1}^T \mathbb{E}_{q_\theta q_\psi q_\zeta}[\log p(y_t|z_{c,t}^{dy}, z_{c,t}^{st}, z_t^d)]$$
$$+ \sum_{t=1}^T \mathbb{D}_{KL}(q_\zeta(z_t^d|z_{<t}^d, y_t), p(z_t^d|z_{<t}^d)). \tag{11}$$

### 3.3.5. OVERALL OPTIMIZATION OBJECTIVE

Let $\mathcal{L}_{\text{evolve}} = \mathcal{L}_{\text{fp}} + \mathcal{L}_{\text{mp}}$, where $\mathcal{L}_{\text{evolve}}$ represents the loss function for capturing the evolving pattern. Let $\mathcal{L}_{\text{causal}} = \mathcal{L}_{\text{stc}} + \mathcal{L}_{\text{dyc}}$, where $\mathcal{L}_{\text{stc}} = -\sum_{t=2}^T I_{st}(t)$ and $\mathcal{L}_{\text{dyc}} = -\sum_{t=1}^T I_{dy}(t)$, $I_{st}(t)$ and $I_{dy}(t)$ represent the term

Table 1: Comparison of accuracy (%) between SYNC and other baseline methods. "Wst" and "Avg" denote worst-case and mean performance across all test domains for a given dataset, respectively. The best and second best results are marked in **bold** and underline, respectively. "Overall" refers to the mean of the two metrics averaged across all datasets.

| Method | Circle | | Sine | | RMNIST | | Portraits | | Caltran | | PowerSupply | | ONP | | Overall | |
|---|---|---|---|---|---|---|---|---|---|---|---|---|---|---|---|---|
| | Wst | Avg | Wst | Avg | Wst | Avg | Wst | Avg | Wst | Avg | Wst | Avg | Wst | Avg | Wst | Avg |
| ERM | 41.7 | 49.9 | 49.5 | 63.0 | 37.8 | 43.6 | 75.5 | 87.8 | 29.9 | 66.3 | 64.4 | 71.0 | 64.2 | 65.9 | 51.9 | 63.9 |
| Mixup | 41.7 | 48.4 | 49.3 | 62.9 | 38.3 | 44.9 | 75.5 | 87.8 | 52.3 | 66.0 | 64.3 | 70.8 | 64.1 | 66.0 | 55.1 | 63.8 |
| MMD | 41.7 | 50.7 | 47.6 | 55.8 | 39.0 | 44.8 | 74.0 | 87.3 | 29.4 | 57.1 | 64.8 | 70.9 | 47.1 | 51.3 | 49.1 | 59.7 |
| MLDG | 41.7 | 50.8 | 49.5 | 63.2 | 37.5 | 43.1 | 76.4 | 88.5 | 51.7 | 66.2 | 64.6 | 70.8 | 63.9 | 65.9 | 55.0 | 64.1 |
| RSC | 41.7 | 48.0 | 49.5 | 61.5 | 35.8 | 41.7 | 75.2 | 87.3 | 51.9 | 67.0 | 64.4 | 70.9 | 63.3 | 64.7 | 54.5 | 63.0 |
| MTL | 42.2 | 51.2 | 46.9 | 62.9 | 36.1 | 41.7 | 78.2 | 89.0 | 52.6 | 68.2 | 64.2 | 70.7 | 63.4 | 65.9 | 54.8 | 64.2 |
| FISH | 41.7 | 48.8 | 49.5 | 62.3 | 37.6 | 44.2 | 78.6 | 88.8 | 57.5 | 68.6 | 64.2 | 70.8 | 63.2 | 65.9 | 56.0 | 64.2 |
| CORAL | 41.7 | 53.9 | 46.3 | 51.6 | 38.5 | 44.5 | 74.6 | 87.4 | 50.9 | 65.7 | 64.6 | 71.0 | 63.8 | 65.8 | 54.3 | 62.8 |
| AndMask | 41.7 | 47.9 | 42.9 | 69.3 | 37.8 | 42.8 | 62.0 | 70.3 | 29.9 | 56.9 | 64.0 | 70.7 | 51.2 | 54.6 | 47.1 | 58.9 |
| DIVA | 56.7 | 67.9 | 38.6 | 52.9 | 36.9 | 42.7 | 76.2 | 88.2 | 53.8 | 69.2 | 63.9 | 70.7 | 64.5 | 66.0 | 55.8 | 65.4 |
| IRM | 41.7 | 51.3 | 49.5 | 63.2 | 34.1 | 39.0 | 74.2 | 85.4 | 43.1 | 60.6 | 64.1 | 70.8 | 61.9 | 64.5 | 52.7 | 62.1 |
| IIB | 42.0 | 53.9 | 47.6 | 61.3 | 38.5 | 45.0 | 78.1 | 89.7 | 54.8 | 69.3 | 64.5 | 70.8 | 63.2 | 66.3 | 55.2 | 62.1 |
| iDAG | 42.0 | 49.0 | 47.6 | 57.1 | 39.8 | 44.1 | 79.8 | 88.6 | 53.5 | 69.7 | 66.1 | 71.2 | 63.8 | 66.4 | 56.1 | 63.7 |
| GI | 42.0 | 54.4 | 49.8 | 65.2 | 39.2 | 44.6 | 77.8 | 88.1 | 53.1 | 70.7 | 65.1 | 71.0 | 63.5 | 65.7 | 55.8 | 65.7 |
| LSSAE | 42.0 | 73.8 | 49.0 | 71.4 | 40.3 | 46.4 | 77.7 | 89.1 | 54.3 | 70.3 | 65.4 | 71.1 | 64.5 | 66.0 | 56.2 | 69.7 |
| DDA | 42.0 | 51.2 | 43.0 | 66.6 | 38.0 | 45.1 | 76.0 | 87.9 | 31.0 | 66.1 | 64.4 | 70.9 | 63.5 | 65.3 | 51.2 | 64.7 |
| DRAIN | 45.0 | 50.7 | 43.0 | 71.3 | 37.2 | 43.8 | 77.7 | 89.4 | 55.7 | 69.0 | 64.9 | 71.0 | 59.8 | 61.1 | 54.8 | 65.2 |
| MMD-LSAE | 54.0 | 79.5 | 43.0 | 71.4 | 42.9 | 49.2 | 80.9 | 90.4 | 56.9 | 69.6 | 65.2 | 71.4 | 61.8 | 66.4 | 58.1 | 70.9 |
| CTOT | 54.0 | 75.2 | 43.2 | 67.2 | 31.7 | 44.8 | 79.2 | 86.4 | 48.0 | 66.9 | 63.6 | 71.1 | 61.1 | 65.6 | 54.4 | 68.2 |
| SDE-EDG | 45.0 | 81.5 | 42.3 | 72.2 | 39.7 | 52.6 | 78.6 | 89.6 | 55.1 | 71.3 | 64.1 | 70.8 | 63.1 | 65.6 | 55.4 | 71.9 |
| SYNC (Ours) | 67.0 | 84.7 | 60.0 | 76.1 | 45.8 | 50.8 | 81.0 | 90.8 | 58.8 | 72.2 | 66.9 | 71.7 | 64.6 | 65.6 | 63.4 | 73.1 |

at time $t$ as defined in Eq. (6) and Eq. (9), respectively. The objective function of SYNC is

$$\mathcal{L}_{\text{SYNC}} = \mathcal{L}_{\text{evolve}} + \alpha_1 \mathcal{L}_{\text{MI}} + \alpha_2 \mathcal{L}_{\text{causal}}, \qquad (12)$$

where $\alpha_1$ and $\alpha_2$ are the trade-off hyperparameters.

### 3.4. Theoretical Analysis

In this section, we aim to provide a theoretical insight on our proposed SYNC. More details on the theoretical proof and discussion can be found in Appendix B. First, we show that optimizing $\mathcal{L}_{\text{evolve}}$ approximates the joint distribution $p(\boldsymbol{x}_{1:T}, y_{1:T})$ and captures the underlying evolving pattern.

**Theorem 1.** *Assume that the underlying data generation process at each time step is characterized by SCM $\mathcal{M}_T$. Then, by optimizing $\mathcal{L}_{evolve}$, the model can learn the data distribution $p(\boldsymbol{x}_{1:T}, y_{1:T})$ on training domains.*

Thm. 1 demonstrates that minimizing $\mathcal{L}_{\text{evolve}}$ can effectively capture the underlying evolving pattern in data distribution. Now, we provide Thm. 2 as follows to state that minimizing $\mathcal{L}_{\text{SYNC}}$ leads to the optimal causal predictor at each time domain $\mathcal{D}_t$, thereby addressing spurious correlations and improving model generalization.

**Theorem 2.** *Let $\Phi_c^{st}(X_t)$, $\Phi_c^{dy}(X_t)$ and $Z_t^d$ denote the representations and drift factors at the time domain $\mathcal{D}_t$, obtained by training the network through the optimization of $\mathcal{L}_{SYNC}$. Then the predictor constructed upon these components is the optimal causal predictor as defined in Def. 2.*

## 4. Experiments

### 4.1. Experimental setup

**Datasets.** We evaluate SYNC on several commonly used benchmarks, including two synthetic datasets (Circle, Sine) and five real-world datasets (RMNIST, Portraits, Caltran, PowerSupply, ONP). **Circle** (Pesaranghader & Viktor, 2016) contains evolving 30 domains, where the instance are sampled from 30 2D Gaussian distributions. **Sine** (Pesaranghader & Viktor, 2016) includes 24 evolving domains, which is achieved by extending and rearranging the original dataset. **RMNIST** (Ghifary et al., 2015) consists of MNIST digits with varying rotations. Following (Qin et al., 2022), we generate 19 evolving domains by sequentially applying rotations from 0° to 180° in 15° increments. **Portraits** (Yao et al., 2022a) is a real-world dataset of American high school senior photos spanning 108 years across 26 states. We split it into 34 evolving domains using fixed temporal intervals. **Caltran** (Hoffman et al., 2014) is a surveillance dataset captured by fixed traffic cameras deployed at intersections and is split into 34 domains by time to predict the type of scene. **PowerSupply** (Dau et al., 2019) is created by an Italian electricity company and contains 30 evolving domains. **ONP** (Fernandes et al., 2015) is collected from the Mashable website within two years and is divided into 24 domains according to month. All domains are split into source domains, intermediate domains and target domains according to the ratio of $\{1/2 : 1/6 : 1/3\}$. The intermediate domains

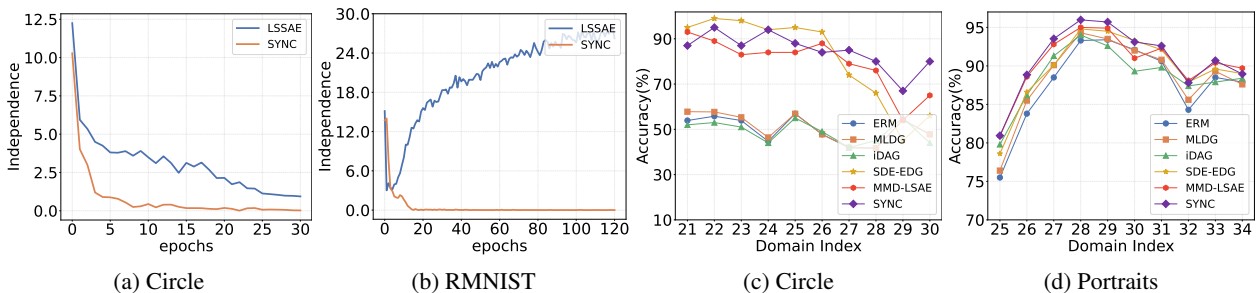

(a) Circle         (b) RMNIST         (c) Circle         (d) Portraits

Figure 5: (a) and (b) show the independence degree of LSSAE and SYNC, respectively, during the training process on the Circle and RMNIST. (c) and (d) present the test accuracy trajectory of SYNC and various baselines on Circle and Portraits.

are utilized as validation set for model selection.

**Baselines.** We select three categories of related methods for comparison: **(i) Non-Causal DG**: ERM (Vapnik, 1999), Mixup (Zhang et al., 2018), MMD (Li et al., 2018b), MLDG (Li et al., 2018a), RSC (Huang et al., 2020), MTL (Blanchard et al., 2021), FISH (Shi et al., 2022), CORAL (Sun & Saenko, 2016), AndMask (Parascandolo et al., 2021), DIVA (Ilse et al., 2020). **(ii) Causal DG**: IRM (Arjovsky et al., 2019), IIB (Li et al., 2022), iDAG (Huang et al., 2023). **(iii) EDG**: GI (Nasery et al., 2021), LSSAE (Qin et al., 2022), DDA (Zeng et al., 2023b), DRAIN (Bai et al., 2023), CTOT (Jin et al., 2024), SDE-EDG (Zeng et al., 2023a), MMD-LSAE (Qin et al., 2023). We keep the neural network architecture of encoding and classification part constant across all baselines used in different benchmarks for a fair comparison.

**Implementation.** All experiments in this work are performed on a single NVIDIA GeForce RTX 4090 GPU with 24GB memory, using the PyTorch packages, and are based on DomainBed (Gulrajani & Lopez-Paz, 2021). Please refer Appendix E.2 for additional training details and Appendix C.3 for network architecture details.

**Evaluation Metrics.** In this work, we report the generalization performance on $K$ target domains in the future, including the worst accuracy performance "Wst" ($\min_{k \in \{1,2,\cdots,K\}} \text{Accuracy}(\mathcal{D}^{T+k})$) and the average accuracy performance "Avg" ($\frac{1}{K} \sum_{k=1}^{K} \text{Accuracy}(\mathcal{D}^{T+k})$).

### 4.2. Quantitative results

The results of our proposed SYNC, along with various baseline methods, are presented in Table 1. It can be observed that the average performance of EDG methods generally exceeds that of traditional DG methods, underscoring the importance of modeling and leveraging evolving patterns in dynamic environments. In terms of worst-case performance, EDG methods do not exhibit a significant improvement over DG methods, in contrast to their advantage in average performance. This may be attributed to the fact that, if EDG methods fail to capture robust evolutionary patterns,

their generalization ability may deteriorate over longer temporal spans. In contrast, DG methods focus on learning domain-invariant representations, which can serve as a performance lower bound and offer a more stable guarantee for generalization. These results demonstrate the importance of leveraging stable representations to enhance model generalization in dynamic environments.

SYNC consistently outperforms other baselines over all benchmarks, achieving an accuracy of 63.4% in the worst-case scenario and 73.1% in terms of average performance. Specifically, compared to existing causal DG methods, SYNC achieves significantly better results on both metrics, with improvements of at least 7.6% in worst-case performance and 7.7% in average performance. This highlights the importance of incorporating dynamic causal information and modeling causal mechanism shifts. Moreover, when compared with EDG methods, SYNC outperforms the state-of-the-art approach by 5.3% and 1.2% in the two metrics, respectively. These results demonstrates that, by learning the causal model, SYNC achieves superior generalization and captures more robust evolving patterns. Lastly, it can be observed that VAE-based methods, such as LSSAE, achieve better performance, suggesting that generative approaches are effective in capturing more robust evolving patterns. Building on this, SYNC further learns both static and dynamic causal representations, which enhances model generalization and advances the modeling of evolving patterns.

### 4.3. Analytical Experiments

**Ablation study** We conduct an ablation study on RM-NIST to evaluate the effectiveness of various components in our method, and results are presented in Table 2. Variant A serves as the base method trained solely with evolving pattern loss $\mathcal{L}_{\text{evolve}}$. Variant B builds upon the base method by additionally training with MI loss $\mathcal{L}_{\text{MI}}$. It can be observed that Variant B achieves better performance than Variant A, suggesting that better separation enables more effective utilization of features from each part. Variants C and D build upon Variant B by incorporating additional training with static causal loss $\mathcal{L}_{\text{stc}}$ and dynamic causal

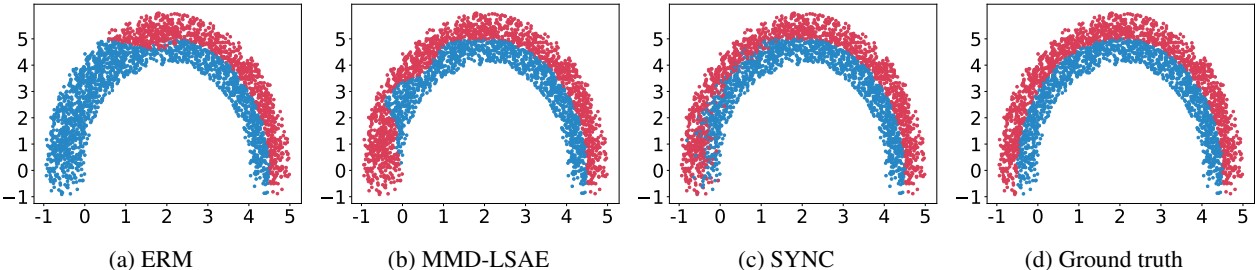

Figure 6: Visualization of decision boundary of the Circle dataset, where the positive and negative classes are colored in red and blue. (a)-(c) show the decision boundaries learned by ERM, MMD-LSAE and SYNC. (d) presents the ground truth.

Table 2: Ablation study of SYNC on RMNIST dataset.

| Ablation | Evo. | Disent. | $Z_c^{st}$ | $Z_c^{dy}$ | Wst | Avg | Overall |
|---|---|---|---|---|---|---|---|
| Variant A | ✓ | - | - | - | 40.5 | 44.1 | 42.3 |
| Variant B | ✓ | ✓ | - | - | 41.9 | 45.7 | 43.8 |
| Variant C | ✓ | ✓ | ✓ | - | 44.1 | 48.7 | 46.4 |
| Variant D | ✓ | ✓ | - | ✓ | 42.9 | 49.2 | 46.1 |
| SYNC | ✓ | ✓ | ✓ | ✓ | **45.8** | **50.8** | **48.3** |

loss $\mathcal{L}_{\text{dyc}}$, respectively. Variant C and D achieve higher accuracy than Variant B, demonstrating that learning causal representations enhances model generalization. Notably, it is clear that Variant C achieves a greater improvement in worst-case performance compared to Variant D, indicating that static causal factors can ensure stable generalization under continuous distribution shifts. However, focusing solely on them ignores evolving pattern, limiting further generalization gains. Learning dynamic causal factors captures features related to task evolving over time, enabling to generalize better to the current distribution. Variant D outperforms Variant C in average performance and provides evidence for this claim. SYNC jointly learns static and dynamic causal representations and achieves the best performance, highlighting their significant contributions to overall effectiveness. This demonstrates that static and dynamic causal factors are complementary and play a crucial role in enhancing model generalization.

**Disentanglement** To illustrate the extent of disentanglement between static and dynamic factors in our approach, we use mutual information as the measure of independence and estimate it using mini-batch weighted sampling between the static and dynamic representations produced by LSSAE and SYNC on the Circle and RMNIST datasets. As shown in Fig. 5a and Fig. 5b, compared to LSSAE, SYNC exhibits a faster and more stable decline in the independence indicator, indicating that our method achieves a more effective disentanglement of static and dynamic factors.

**Temporal Robustness** Fig. 5c and Fig. 5d show the accuracy trajectories of our method and multiple baseline

methods on Circle and Portraits. SYNC demonstrates more stable predictive performance while maintaining high accuracy. Notably, compared to SDE-EDG on the Circle dataset, SYNC continues to perform well in the later domains, highlighting the effectiveness of time-aware causal representation learning in enhancing long-term generalization.

**Evaluation on evolving pattern captured** In order to verify whether the evolving patterns learned through causal representations are more accurate, we visualize the decision boundaries of our proposed approach along with ERM and MMD-LSAE on the Circle dataset. As shown in Fig. 6, ERM struggles to generalize to unseen target domains due to its inability to model latent evolving patterns. In contrast, MMD-LSAE demonstrates improved generalization to future domains. SYNC goes one step further and achieves decision boundaries that most closely align with the ground truth, highlighting its effectiveness in capturing evolving patterns and enhancing generalization to unseen target data.

## 5. Conclusion

In this work, we investigate the issue that existing EDG methods may experience poor generalization performance due to the presence of spurious correlations. To address this problem, we propose a time-aware SCM by considering both dynamic causal factors and causal mechanism drifts. We propose **S**tatic-**DYN**amic **C**ausal Representation Learning (**SYNC**), an approach that can effectively learn time-aware causal representations and reconstruct causal mechanisms, thereby obtaining the causal model. Extensive experiments demonstrate the effectiveness and superiority of our method in improving the temporal generalization performance. We hope this work can offer valuable insights into improving model generalization in dynamic environments.

## Acknowledgements

This paper was supported by the National Natural Science Foundation of China (No. 62376026), Beijing Nova Program (No. 20230484296) and KuaiShou.

## Impact Statement

In this work, we highlight the potential negative impact of spurious correlations on the model generalization in the evolving domain generalization problem and provide insights into how to build and learn causal models in non-stationary environments. By learning time-aware causal representations, the model can effectively mitigate the impact of task-irrelevant factors in dynamic tasks, thereby enabling reliable predictions and decision-making in complex, changing environments. This allows the model to cope with challenges in non-stationary dynamic environments, offering potential benefits for real-world applications such as autonomous driving and advertisement recommendation.

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

# A. Related Work

**Domain Generalization (DG)** aims to generalize knowledge extracted from multiple source domains to unseen target domain. In recent years, extensive research has been conducted to address this issue in the literature (Wang et al., 2023; Liu et al., 2021b), which can be roughly categorized into three groups: (i) Invariant representation learning: This category of methods focus on learning invariant representations across distributions to generalize well to unseen target domains, which is achieved through domain alignment (Muandet et al., 2013; Ghifary et al., 2017; Li et al., 2018c) and feature disentangement (Piratla et al., 2020; Chattopadhyay et al., 2020). Domain alignment methods learn invariant representation via kernel-based optimization (Muandet et al., 2013; Ghifary et al., 2017), adversarial training (Li et al., 2018c; Motiian et al., 2017) and using additional normalization (Peng & Saenko, 2018). Feature disentaglement methods (Piratla et al., 2020; Chattopadhyay et al., 2020) aim to disentagle the features into domain-specific and domain-invariant features, and the latter are utilized for better generalization. (ii) Domain augmentation: These methods enrich the diversity of the training domain by manipulating domain data, thereby enhancing the model's generalization. They encompass operations at both the image levels (Zhang et al., 2018; Hendrycks et al., 2020) and feature levels (Zhou et al., 2021a; 2020; Xu et al., 2021). (iii) Learning strategies: This category of methods utilize effective learning strategies to improve model generalization performance, such as meta-learning (Balaji et al., 2018; Li et al., 2018a), ensemble learning (Zhou et al., 2021b; Arpit et al., 2022) and self-supervised learning (Carlucci et al., 2019; Huang et al., 2020). Although significant progress has been made in DG, when faced with non-stationary environments that are more common in the real world (Grossniklaus et al., 2013; Wang et al., 2025), it still faces major challenges (Yao et al., 2022a; Qin et al., 2022).

# B. Theoretical Details

## B.1. Proof of Proposition 1

**Proposition 1.** *Let $\mathcal{D}_t$ be a time domain, and for a given class $Y$, it can be conclude that:*

*(i) If there is $H(Z_{c,t}^{st}|Z_{c,t-1}^{st},Y) < H(Z_{s,t}^{st}|Z_{c,t-1}^{st},Y)$, then $I(Z_{c,t}^{st};Z_{c,t-1}^{st}|Y) > I(Z_{s,t}^{st};Z_{c,t-1}^{st}|Y)$.*

*(ii) If there is $H(Z_{c,t}^{st}|Z_{c,t}^{dy},Y) < H(Z_{c,t}^{st}|Z_{s,t}^{dy},Y)$, then $I(Z_{c,t}^{dy};Z_{c,t}^{st}|Y) > I(Z_{s,t}^{dy};Z_{c,t}^{st}|Y)$.*

*Proof.* For $I(\cdot;Z_c^{st}|Y)$, according to the properties of conditional MI, it can be derived that

$$
\begin{aligned}
I(\cdot;Z_{c,t-1}^{st}|Y) &= H(Z_{c,t-1}^{st}|Y) - H(Z_{c,t-1}^{st}|\cdot,Y), \\
I(\cdot;Z_{c,t}^{st}|Y) &= H(Z_{c,t}^{st}|Y) - H(Z_{c,t}^{st}|\cdot,Y),
\end{aligned}
\tag{13}
$$

thus we have

$$
\begin{aligned}
H(Z_{c,t}^{st}|Z_{c,t-1}^{st},Y) < H(Z_{s,t}^{st}|Z_{c,t-1}^{st},Y) &\iff I(Z_{c,t}^{st};Z_{c,t-1}^{st}|Y) > I(Z_{s,t}^{st};Z_{c,t-1}^{st}|Y), \\
H(Z_{c,t}^{st}|Z_{c,t}^{dy},Y) < H(Z_{c,t}^{st}|Z_{s,t}^{dy},Y) &\iff I(Z_{c,t}^{dy};Z_{c,t}^{st}|Y) > I(Z_{s,t}^{dy};Z_{c,t}^{st}|Y).
\end{aligned}
\tag{14}
$$

$\square$

These two propositions regarding the properties of causal factors demonstrate that, as long as the causal factors are sufficiently similar, the conditional MI can be maximized to distinguish the causal factors from the spurious factors.

## B.2. Proof of Theorem 1

**Theorem 1.** *Assume that the underlying data generation process at each time step is characterized by SCM $\mathcal{M}_T$. Then, by optimizing $\mathcal{L}_{evolve}$, the model can learn the data distribution $p(\boldsymbol{x}_{1:T}, y_{1:T})$ on training domains.*

*Proof.* According to Eq. (1) and Eq. (11), minimizing $\mathcal{L}_{\text{evolve}}$ is equivalent to maximizing

$$-\mathcal{L}_{\text{evolve}} = \sum_{t=1}^{T} \mathbb{E}_{q_\theta q_\psi}[\log p(\boldsymbol{x}_t|\boldsymbol{z}_t^{st}, \boldsymbol{z}_t^{dy})] - \sum_{t=1}^{T} \mathbb{D}_{KL}(q_\psi(\boldsymbol{z}_t^{st}|\boldsymbol{x}_t), p(\boldsymbol{z}_t^{st})) - \sum_{t=1}^{T} \mathbb{D}_{KL}(q_\theta(\boldsymbol{z}_t^{dy}|\boldsymbol{z}_{<t}^{dy}, \boldsymbol{x}_t), p(\boldsymbol{z}_t^{dy}|\boldsymbol{z}_{<t}^{dy}))$$
$$+ \sum_{t=1}^{T} \mathbb{E}_{q_\theta q_\psi q_\zeta}[\log p(y_t|\boldsymbol{z}_{c,t}^{st}, \boldsymbol{z}_{c,t}^{dy}, \boldsymbol{z}_t^{d})] - \sum_{t=1}^{T} \mathbb{D}_{KL}(q_\zeta(\boldsymbol{z}_t^{d}|\boldsymbol{z}_{<t}^{d}, y_t), p(\boldsymbol{z}_t^{d}|\boldsymbol{z}_{<t}^{d})) \,. \tag{15}$$

The above equation can be further simplified to

$$-\mathcal{L}_{\text{evolve}} = \sum_{t=1}^{T} \mathbb{E}_{q_\theta q_\psi}[\log p(\boldsymbol{x}_t|\boldsymbol{z}_t^{st}, \boldsymbol{z}_t^{dy})] + \sum_{t=1}^{T} \mathbb{E}_{q_\theta q_\psi q_\zeta}[\log p(y_t|\boldsymbol{z}_{c,t}^{st}, \boldsymbol{z}_{c,t}^{dy}, \boldsymbol{z}_t^{d})]$$
$$- \sum_{t=1}^{T} \mathbb{E}_{q_\psi} \log \frac{q_\psi(\boldsymbol{z}_t^{st}|\boldsymbol{x}_t)}{p(\boldsymbol{z}_t^{st})} - \sum_{t=1}^{T} \mathbb{E}_{q_\theta} \log \frac{q_\theta(\boldsymbol{z}_t^{dy}|\boldsymbol{z}_{<t}^{dy}, \boldsymbol{x}_t)}{p(\boldsymbol{z}_t^{dy}|\boldsymbol{z}_{<t}^{dy})} - \sum_{t=1}^{T} \mathbb{E}_{q_\zeta} \log \frac{q_\zeta(\boldsymbol{z}_t^{d}|\boldsymbol{z}_{<t}^{d}, y_t)}{p(\boldsymbol{z}_t^{d}|\boldsymbol{z}_{<t}^{d})} \tag{16}$$
$$= \sum_{t=1}^{T} \mathbb{E}_{q_\psi q_\theta q_\zeta} \log \frac{p(\boldsymbol{x}_t|\boldsymbol{z}_t^{st}, \boldsymbol{z}_t^{dy})p(y_t|\boldsymbol{z}_{c,t}^{st}, \boldsymbol{z}_{c,t}^{dy}, \boldsymbol{z}_t^{d})p(\boldsymbol{z}^{st})p(\boldsymbol{z}_t^{dy}|\boldsymbol{z}_{<t}^{dy})p(\boldsymbol{z}_t^{dy}|\boldsymbol{z}_{<t}^{dy})}{q_\psi(\boldsymbol{z}^{st}|\boldsymbol{x}_t)q_\theta(\boldsymbol{z}_t^{dy}|\boldsymbol{z}_{<t}^{dy}, \boldsymbol{x}_t)q_\zeta(\boldsymbol{z}_t^{d}|\boldsymbol{z}_{<t}^{d}, y_t)} \,.$$

Based on the time-aware SCM $\mathcal{M}_T$ and the temporally evolving SCM $\mathcal{M}_{evo}$, and let $q(\boldsymbol{z}_t^{st}, \boldsymbol{z}_t^{dy}, \boldsymbol{z}_t^{d}|\boldsymbol{x}_t, y_t) := q_\psi(\boldsymbol{z}_t^{st}|\boldsymbol{x}_t)q_\theta(\boldsymbol{z}_t^{dy}|\boldsymbol{z}_{<t}^{dy}, \boldsymbol{x}_t)q_\zeta(\boldsymbol{z}_t^{d}|\boldsymbol{z}_{<t}^{d}, y_t)$, then it can be obtained that

$$-\mathcal{L}_{\text{evolve}} = \sum_{t=1}^{T} \mathbb{E}_{q_\psi q_\theta q_\zeta} \log \frac{p(\boldsymbol{x}_t|\boldsymbol{z}_t^{st}, \boldsymbol{z}_t^{dy})p(y_t|\boldsymbol{z}_{c,t}^{st}, \boldsymbol{z}_{c,t}^{dy}, \boldsymbol{z}_t^{d})p(\boldsymbol{z}_t^{st})p(\boldsymbol{z}_t^{dy}|\boldsymbol{z}_{<t}^{dy})p(\boldsymbol{z}_t^{dy}|\boldsymbol{z}_{<t}^{dy})}{q_\psi(\boldsymbol{z}_t^{st}|\boldsymbol{x}_t)q_\theta(\boldsymbol{z}_t^{dy}|\boldsymbol{z}_{<t}^{dy}, \boldsymbol{x}_t)q_\zeta(\boldsymbol{z}_t^{d}|\boldsymbol{z}_{<t}^{d}, y_t)}$$
$$\leq \mathbb{E}_{q_\psi q_\theta q_\zeta} \sum_{t=1}^{T} \log \frac{p(\boldsymbol{x}_t|\boldsymbol{z}_t^{st}, \boldsymbol{z}_t^{dy})p(y_t|\boldsymbol{z}_{c,t}^{st}, \boldsymbol{z}_{c,t}^{dy}, \boldsymbol{z}_t^{d})p(\boldsymbol{z}_t^{st})p(\boldsymbol{z}_t^{dy}|\boldsymbol{z}_{<t}^{dy})p(\boldsymbol{z}_t^{dy}|\boldsymbol{z}_{<t}^{dy})}{q_\psi(\boldsymbol{z}_t^{st}|\boldsymbol{x}_t)q_\theta(\boldsymbol{z}_t^{dy}|\boldsymbol{z}_{<t}^{dy}, \boldsymbol{x}_t)q_\zeta(\boldsymbol{z}_t^{d}|\boldsymbol{z}_{<t}^{d}, y_t)} \tag{17}$$
$$= \mathbb{E}_{q_\psi q_\theta q_\zeta} \log \frac{\prod_{t=1}^{T} p(\boldsymbol{x}_t|\boldsymbol{z}_t^{st}, \boldsymbol{z}_t^{dy})p(y_t|\boldsymbol{z}_{c,t}^{st}, \boldsymbol{z}_{c,t}^{dy}, \boldsymbol{z}_t^{d})p(\boldsymbol{z}_t^{st})p(\boldsymbol{z}_t^{dy}|\boldsymbol{z}_{<t}^{dy})p(\boldsymbol{z}_t^{dy}|\boldsymbol{z}_{<t}^{dy})}{\prod_{t=1}^{T} q_\psi(\boldsymbol{z}_t^{st}|\boldsymbol{x}_t)q_\theta(\boldsymbol{z}_t^{dy}|\boldsymbol{z}_{<t}^{dy}, \boldsymbol{x}_t)q_\zeta(\boldsymbol{z}_t^{d}|\boldsymbol{z}_{<t}^{d}, y_t)}$$
$$= \mathbb{E}_{q} \log \frac{p(\boldsymbol{x}_{1:T}, y_{1:T}, \boldsymbol{z}_{1:T}^{st}, \boldsymbol{z}_{1:T}^{dy} \boldsymbol{z}_{1:T}^{d})}{q(\boldsymbol{z}_{1:T}^{st}, \boldsymbol{z}_{1:T}^{dy}, \boldsymbol{z}_{1:T}^{d}|\boldsymbol{x}_{1:T}, y_{1:T})} \,.$$

Since the distribution of observational data $p(\boldsymbol{x}_{1:T}, y_{1:T})$ is generated by the latent factors $Z^{st}$, $Z^{dy}$, and $Z^{d}$, we aim to model this distribution by approximating the posterior $p(\boldsymbol{z}_{1:T}^{st}, \boldsymbol{z}_{1:T}^{dy}, \boldsymbol{z}_{1:T}^{d}|\boldsymbol{x}_{1:T}, y_{1:T})$ with a variational network $q$:

$$\mathbb{D}_{KL}(q, p) = \mathbb{E}_{q} \log \frac{q(\boldsymbol{z}_{1:T}^{st}, \boldsymbol{z}_{1:T}^{dy}, \boldsymbol{z}_{1:T}^{d}|\boldsymbol{x}_{1:T}, y_{1:T})}{p(\boldsymbol{z}_{1:T}^{st}, \boldsymbol{z}_{1:T}^{dy}, \boldsymbol{z}_{1:T}^{d}|\boldsymbol{x}_{1:T}, y_{1:T})}$$
$$= -\mathbb{E}_{q} \log \frac{p(\boldsymbol{z}_{1:T}^{st}, \boldsymbol{z}_{1:T}^{dy}, \boldsymbol{z}_{1:T}^{d}, \boldsymbol{x}_{1:T}, y_{1:T})}{q(\boldsymbol{z}_{1:T}^{st}, \boldsymbol{z}_{1:T}^{dy}, \boldsymbol{z}_{1:T}^{d}|\boldsymbol{x}_{1:T}, y_{1:T})} + \log p(\boldsymbol{x}_{1:T}, y_{1:T}) \,. \tag{18}$$

Combing with Eq. (17), it is clear that optimizing $\mathcal{L}_{\text{evolve}}$ is equivalent to minimizing $\mathbb{D}_{KL}(q, p)$, which facilitates learning the data distribution $p(\boldsymbol{x}_{1:T}, y_{1:T})$. $\qquad\square$

## B.3. Proof of Theorem 2

In this section, we elaborate on how SYNC derives the optimal causal predictor for each time domain. We first briefly introduce the relevant theories of causal graphical models and causal assumptions in Sec. B.3.1. Then, we provide the proofs of Lem. 1 and Lem. 2 in Sec. B.3.2 and Sec. B.3.3, respectively. Finally, the proof of Thm. 2 is presented in Sec. B.3.4.

### B.3.1. PRELIMINARIES

**Causal Graphical Models.** In causal research (Pearl, 2009; Pearl et al., 2016), a causal graphical model $\mathcal{G}$ is a directed acyclic graph (DAG) that can be derived from a SCM $\mathcal{M}$, characterizing the joint probability distribution $P$ over $n$ random

variables $X_1, X_2, \cdots, X_n$. The causal DAG $\mathcal{G}$ is comprised of the nodes $V$ and the directed edges $E$, where the starting point and end point of each directed edge ($\rightarrow$) represent cause and effect, respectively. The directed edges encode the causal relationships between the variables. The joint distribution $P$ has causal factorization as follows:

$$P(X_{1:n}) = \prod_{i=1}^{n} P(X_i | \boldsymbol{PA}(X_i)), \tag{19}$$

where $\boldsymbol{PA}(X_i)$ denotes the parents of variable $X_i$ in DAG $\mathcal{G}$. Causal graphical models typically consist of three fundamental structures: chain, fork, and collider. The three basic causal structures and their metaphorical dependencies are as follows:

- Chain: $X \rightarrow Y \rightarrow Z$. In a chain structure, $X$ is a cause that influences $Y$, and $Y$ is a cause of $Z$. This structure implies dependency: $X \not\perp\!\!\!\perp Z$, while $X \not\perp\!\!\!\perp Z|Y$.

- Fork: $X \leftarrow Y \rightarrow Z$. In this structure, $Y$ is the common parent of both $X$ and $Z$, and is therefore referred to as a confounder. A fork structure shows that $X \not\perp\!\!\!\perp Z$, while $X \perp\!\!\!\perp Z|Y$.

- Collider: $X \rightarrow Y \leftarrow Z$. $Y$ is the common child of the two variables, $X$ and $Z$, and is referred to as a collider. The dependency implied by a collider structure is $X \perp\!\!\!\perp Z$, while $X \not\perp\!\!\!\perp Z|Y$.

**Causal Assumptions.** Due to the available data offer only partial insights into the underlying causal mechanisms. Therefore, it is crucial to make certain priors or assumptions about the structure of the causal relationships to enable causality learning. The following assumptions (Pearl, 2009; Pearl et al., 2016) are commonly used:

- **Causal Markovian Condition.** For each variable in a causal model, if given its parents, then it is conditionally independent of all its non-descendants.

- **Causal Sufficient**. The causal sufficiency assumption states that there are no unmeasured common causes of any pair of variables in a causal model. This assumption is crucial in a wide range of literature.

- **Global Markov Condition**. Given a DAG $\mathcal{G}$ and the joint distribution $P$ of all nodes, for any two non-adjacent nodes $X_i$ and $X_j$ in $\mathcal{G}$, and given a set of nodes $Z$, if $X_i$ and $X_j$ are d-separated by $Z$, then $X_i \perp\!\!\!\perp X_j \mid Z$. We say that the distribution $P$ satisfies the global Markov property with respect to the DAG $\mathcal{G}$.

- **Causal Faithfulness**. Given a DAG $\mathcal{G}$ and the joint probability distribution $P$ of all nodes, for any two non-adjacent nodes $X_i$ and $X_j$ in $G$, given a set of nodes $Z$, if $X_i \perp\!\!\!\perp X_j \mid Z$, then the nodes $X_i$ and $X_j$ are d-separated by the node set $Z$. We say that the distribution $P$ satisfies the faithfulness with respect to $\mathcal{G}$. This assumption represents exactly the distributional independence relations implied by d-separation.

With these theories, we combine the time-aware SCM $\mathcal{M}_T$ to give the following proposition, which leads to the definition of the optimal causal predictor.

**Proposition 2** (Conditional Independence). *It is assumed that there are no unmeasured common causes of any pair of variables in $\mathcal{M}_T$, i.e., $\mathcal{M}_T$ is causal sufficient. With the global Markov condition and the causal faithfulness, the conditional independence of latent factors can be derived from d-separation: (i) $Y \not\perp\!\!\!\perp [Z_s^{st}, Z_s^{dy}] \mid [Z_c^{st}, Z_c^{dy}]$. (ii) $Y \perp\!\!\!\perp [Z_s^{st}, Z_s^{dy}] \mid [Z_c^{st}, Z_c^{dy}, Z^d]$.*

*Proof.* The DAG corresponding to the time-aware SCM $\mathcal{M}_T$ is shown in Fig. 2b. It can be observed that there exists three paths from $(Z_s^{st}, Z_s^{dy})$ to $Y$: $Z_s^{st} \leftarrow G \rightarrow Z_c^{st} \rightarrow Y$, $Z_s^{dy} \leftarrow L \rightarrow Z_c^{dy} \rightarrow Y$, and $Z_s^{dy} \leftarrow L \rightarrow Z^d \rightarrow Y$. Note that the causal sufficiency of $\mathcal{M}_T$ guarantees that there are no other unobserved variables, thus all the paths between $(Z_s^{st}, Z_s^{dy})$ and $Y$ are blocked by $Z_c^{st}$, $Z_c^{dy}$ and $Z^d$. Therefore, it follows that $Z_c^{st}$, $Z_c^{dy}$ and $Z^d$ block all the paths from $T$ to $Y$, whereas $Z_c^{st}$ and $Z_c^{dy}$ do not. According to d-separation criterion, it is obtained that $Y \not\perp\!\!\!\perp_{\mathcal{M}_T} [Z_s^{st}, Z_s^{dy}] \mid [Z_c^{st}, Z_c^{dy}]$, $Y \perp\!\!\!\perp_{\mathcal{M}_T} [Z_s^{st}, Z_s^{dy}] \mid [Z_c^{st}, Z_c^{dy}, Z^d]$. Since the global Markov condition and the causal faithfulness guarantee the equivalence between $d$-separation and conditional independence, therefore we have:

$$Y \not\perp\!\!\!\perp [Z_s^{st}, Z_s^{dy}] \mid [Z_c^{st}, Z_c^{dy}], \quad Y \perp\!\!\!\perp [Z_s^{st}, Z_s^{dy}] \mid [Z_c^{st}, Z_c^{dy}, Z^d]. \tag{20}$$

$\square$

From Prop. 2, it follows that for each time domain $\mathcal{D}_t$, if we can obtain the static causal factors $Z_c^{st}$, dynamic causal factors $Z_c^{dy}$, and drift factors $Z^d$ that indicate the state of the causal mechanism in the current time domain, then we can use them to construct a stable causal relationship between features and labels. The predictor built in this way will not be influenced by spurious factors. Additionally, the ideal causal features should capture as much information as possible about the target. Based on the above analysis, we now define the optimal causal predictor.

**Definition 2** (Optimal Causal Predictor). For a time domain $\mathcal{D}_t$, the corresponding drift factors $Z_t^d$ are given. Let $Z_{c,t} = (Z_{c,t}^{st}, Z_{c,t}^{dy})$ be the causal factors in domain $\mathcal{D}_t$ that satisfy $Z_{c,t} \in \arg\max_{Z_{c,t}} I(Y; Z_{c,t}, Z_t^d)$, and $Y \perp\!\!\!\perp [Z_{s,t}^{st}, Z_{s,t}^{dy}] \mid [Z_{c,t}, Z_t^d]$. Then the predictor based on factors $Z_{c,t}$ and $Z_t^d$ is the optimal causal predictor.

The following two lemmas establish a theoretical foundation for learning the causal factors $Z_{c,t}$.

### B.3.2. PROOF OF LEMMA 1

**Lemma 1.** *For a time domain $\mathcal{D}_t$, the static causal factors can be obtained by solving the following objective:*

$$\max_{\Phi_c^{st}} I(Y; \Phi_c^{st}(X_t)), \quad s.t. \ \Phi_c^{st} \in \arg\max_{\Phi_c^{st}} I(\Phi_c^{st}(X_t); \Phi_c^{st}(X_{t-1})|Y). \tag{21}$$

*Proof.* For simplicity, we use $\Phi_{c,t}^{st}$ to represent $\Phi_c^{st}(X_t)$. For a pair of domains $(\mathcal{D}_{t-1}, \mathcal{D}_t)$ sampled from training domains $\{\mathcal{D}_t\}_{t=1}^T$, maximizing conditional MI $I(\Phi_{c,t}^{st}; \Phi_{c,t-1}^{st}|Y)$ is essentially maximizing $I(\Phi_{c,t}^{st}, t; \Phi_{c,t-1}^{st}, t-1|Y)$ (Chen et al., 2022), where $t$ and $t-1$ represent all the information within the corresponding time domain (including both global and local information) and serve as proxy variables for time. Thus, we have

$$
\begin{aligned}
\max_{\Phi_c^{st}} I(\Phi_c^{st}(X_t); \Phi_c^{st}(X_{t-1})|Y) &\iff \max_{\Phi_c^{st}} I(\Phi_{c,t}^{st}, t; \Phi_{c,t-1}^{st}, t-1|Y) \\
&\iff \max_{\Phi_c^{st}} H(\Phi_{c,t}^{st}, t|Y) - H(\Phi_{c,t}^{st}, t|\Phi_{c,t-1}^{st}, t-1, Y).
\end{aligned}
\tag{22}
$$

Let $Z_c^{st}$ and $Z_s^{st}$ denote the ground truth static causal factors and static spurious factors, respectively, while $Z_{lc}^{st} \subset Z_c^{st}$ and $Z_{ls}^{st} \subset Z_s^{st}$ represent the static causal factors and static spurious factors learned by the model. After that, we define $Z_{rc}^{st} = Z_c^{st} - Z_{lc}^{st}$ and $Z_{rs}^{st} = Z_s^{st} - Z_{ls}^{st}$ as the remaining static causal factors and static spurious factors. For term $H(\Phi_{c,t}^{st}, t|Y)$, according to chain rule of entropy:

$$H(\Phi_{c,t}^{st}, t|Y) = H(t|Y) + H(\Phi_{c,t}^{st}|t, Y) = H(\Phi_{c,t}^{st}|t, Y), \tag{23}$$

where the second equality is due to $t$ is determined so that $H(t|Y) = 0$. Suppose that the representations $\Phi_{c,t}^{st}$ learned by the model contain both causal information and spurious information, *i.e.*, $\Phi_{c,t}^{st} = (Z_{lc,t}^{st}, Z_{ls,t}^{st})$, then Eq. (23) can be rewritten as

$$H(\Phi_{c,t}^{st}|t, Y) = H(Z_{lc,t}^{st}, Z_{ls,t}^{st}|t, Y) = H(Z_{lc,t}^{st}|t, Y) + H(Z_{ls,t}^{st}|Z_{lc,t}^{st}, t, Y). \tag{24}$$

Analogously, replacing $\Phi_{c,t}^{st}$ with the ground truth $Z_{c,t}^{st}$ ($Z_{c,t}^{st} = (Z_{lc,t}^{st}, Z_{rc,t}^{st})$) yields

$$H(Z_{c,t}^{st}|t, Y) = H(Z_{lc,t}^{st}, Z_{rc,t}^{st}|t, Y) = H(Z_{lc,t}^{st}|t, Y) + H(Z_{rc,t}^{st}|Z_{lc,t}^{st}, t, Y). \tag{25}$$

Hence, it can be konwn that

$$
\begin{aligned}
\Delta H(\Phi_{c,t}^{st}|t, Y) &= H(Z_{c,t}^{st}|t, Y) - H(\Phi_{c,t}^{st}|t, Y) \\
&= \left(H(Z_{lc,t}^{st}|t, Y) + H(Z_{rc,t}^{st}|Z_{lc,t}^{st}, t, Y)\right) - \left(H(Z_{lc,t}^{st}|t, Y) + H(Z_{ls,t}^{st}|Z_{lc,t}^{st}, t, Y)\right) \\
&= H(Z_{rc,t}^{st}|Z_{lc,t}^{st}, t, Y) - H(Z_{ls,t}^{st}|Z_{lc,t}^{st}, t, Y).
\end{aligned}
\tag{26}
$$

For the second term $H(\Phi_{c,t}^{st}, t|\Phi_{c,t-1}^{st}, t-1, Y)$ in Eq. (22), suppose that $\Phi_{c,t}^{st} = (Z_{lc,t}^{st}, Z_{ls,t}^{st})$, $\Phi_{c,t-1}^{st} = (Z_{lc,t-1}^{st}, Z_{ls,t-1}^{st})$, then it can be derived that

$$
\begin{aligned}
H(\Phi_{c,t}^{st}, t|\Phi_{c,t-1}^{st}, t-1, Y) &= H(Z_{lc,t}^{st}, Z_{ls,t}^{st}, t|Z_{lc,t-1}^{st}, Z_{ls,t-1}^{st}, t-1, Y) \\
&= H(Z_{lc,t}^{st}, t|Z_{ls,t}^{st}, Z_{lc,t-1}^{st}, Z_{ls,t-1}^{st}, t-1, Y) + H(Z_{ls,t}^{st}|Z_{lc,t}^{st}, t, Z_{lc,t-1}^{st}, Z_{ls,t-1}^{st}, t-1, Y).
\end{aligned}
\tag{27}
$$

Similarly, let $Z_{c,t-1}^{st} = (Z_{lc,t-1}^{st}, Z_{rc,t-1}^{st})$, we can get:

$$H(Z_{c,t}^{st}, t | Z_{c,t-1}^{st}, t-1, Y) = H(Z_{lc,t}^{st}, t | Z_{rc,t}^{st}, Z_{lc,t-1}^{st}, Z_{rc,t-1}^{st}, t-1, Y) + H(Z_{rc,t}^{st} | Z_{lc,t}^{st}, t, Z_{lc,t-1}^{st}, Z_{rc,t-1}^{st}, t-1, Y).$$
(28)

Combing with Eq. (27) and Eq. (28), we have:

$$\begin{aligned}
\Delta H(\Phi_{c,t}^{st}, t | \Phi_{c,t-1}^{st}, t-1, Y) &= H(Z_{c,t}^{st}, t | Z_{c,t-1}^{st}, t-1, Y) - H(\Phi_{c,t}^{st}, t | \Phi_{c,t-1}^{st}, t-1, Y) \\
&= H(Z_{lc,t}^{st}, t | Z_{rc,t}^{st}, Z_{lc,t-1}^{st}, Z_{rc,t-1}^{st}, t-1, Y) - H(Z_{lc,t}^{st}, t | Z_{ls,t}^{st}, Z_{lc,t-1}^{st}, Z_{ls,t-1}^{st}, t-1, Y) \\
&\quad + H(Z_{rc,t}^{st} | Z_{lc,t}^{st}, t, Z_{lc,t-1}^{st}, Z_{rc,t-1}^{st}, t-1, Y) - H(Z_{ls,t}^{st} | Z_{lc,t}^{st}, t, Z_{lc,t-1}^{st}, Z_{ls,t-1}^{st}, t-1, Y).
\end{aligned}$$
(29)

Now we can derive the expression for the change in the conditional MI $I(\Phi_{c,t}^{st}; \Phi_{c,t-1}^{st} | Y)$:

$$\begin{aligned}
&I(Z_{c,t}^{st}; Z_{c,t-1}^{st} | Y) - I(\Phi_{c,t}^{st}; \Phi_{c,t-1}^{st} | Y) \\
&= \Delta I(\Phi_{c,t}^{st}; \Phi_{c,t-1}^{st} | Y) \\
&= \Delta H(\Phi_{c,t}^{st} | t, Y) - \Delta H(\Phi_{c,t}^{st}, t | \Phi_{c,t-1}^{st}, t-1, Y) \\
&= \underbrace{\left( H(Z_{rc,t}^{st} | Z_{lc,t}^{st}, t, Y) - H(Z_{rc,t}^{st} | Z_{lc,t}^{st}, t, Z_{lc,t-1}^{st}, Z_{rc,t-1}^{st}, t-1, Y) \right)}_{\text{term 1}} \\
&\quad + \underbrace{\left( H(Z_{lc,t}^{st}, t | Z_{ls,t}^{st}, Z_{lc,t-1}^{st}, Z_{ls,t-1}^{st}, t-1, Y) - H(Z_{lc,t}^{st}, t | Z_{rc,t}^{st}, Z_{lc,t-1}^{st}, Z_{rc,t-1}^{st}, t-1, Y) \right)}_{\text{term 2}} \\
&\quad + \underbrace{\left( H(Z_{ls,t}^{st} | Z_{lc,t}^{st}, t, Z_{lc,t-1}^{st}, Z_{ls,t-1}^{st}, t-1, Y) - H(Z_{ls,t}^{st} | Z_{lc,t}^{st}, t, Y) \right)}_{\text{term 3}}.
\end{aligned}$$
(30)

For term 1 in Eq. (30), since conditioning reduces the entropy for both discrete and continuous variables, we can conclude that the value of term 1 is greater that 0. For term 2, since the similarity between causal factors is generally higher than the similarity between causal factors and spurious factors, the entropy of the second half is smaller than that of the first half. Therefore, the value of term 2 is greater than 0. For term 3, it can be rewritten as $-I(Z_{ls,t}^{st}, Z_{ls,t-1}^{st}, Z_{lc,t-1}^{st}, t-1 | Z_{lc,t}^{st}, t, Y)$. When $t$ is used as a condition, it is equivalent to giving all the information of $\mathcal{D}_t$. At this time, the spurious factors at time $t$ and the factors at time $t-1$ are independent of each other, so the value of term 3 is 0. Based on the above analysis, we have

$$I(Z_{c,t}^{st}; Z_{c,t-1}^{st} | Y) - I(\Phi_{c,t}^{st}; \Phi_{c,t-1}^{st} | Y) > 0.$$
(31)

The above inequality indicates that maximizing the conditional MI $I(\Phi_{c,t}^{st}; \Phi_{c,t-1}^{st} | Y)$ will allow $\Phi_c^{st}(\cdot)$ to capture static causal factors $Z_c^{st}$. Hence, Lem. 1 is proved. □

We have known that $\Phi_c^{dy}(\cdot)$ can extract dynamic category-related information by optimizing the cross-entropy loss between $\Phi_c^{dy}(X)$ and target $Y$. The following Lem. 2 shows that dynamic causal factors $Z_c^{dy}$ can be learned by introducing the intra-domain conditional MI constraints.

### B.3.3. PROOF OF LEMMA 2

**Lemma 2.** *For a time domain $\mathcal{D}_t$, let $\Phi_c^{st,\star}(X_t)$ be the static causal representations learned by the model, then the dynamic causal factors of $X_t$ can be obtained through the following objective:*

$$\max_{\Phi_c^{dy}} I(Y; \Phi_c^{dy}(X_t)), \quad s.t. \ \Phi_c^{dy} \in \arg\max_{\Phi_c^{dy}} I(\Phi_c^{dy}(X_t); \Phi_c^{st,\star}(X_t) | Y).$$
(32)

*Proof.* For simplicity, we use $\Phi_{c,t}^{st}$ and $\Phi_{c,t}^{dy}$ to represent $\Phi_c^{st}(X_t)$ and $\Phi_c^{dy}(X_t)$, respectively. Let $Z_c^{dy}$ and $Z_s^{dy}$ be the ground truth dynamic causal factors and dynamic spurious factors, respectively, while $Z_{lc}^{dy} \subset Z_c^{dy}$ and $Z_{ls}^{dy} \subset Z_s^{dy}$ represent the dynamic causal factors and dynamic spurious factors learned by the model. After that, we define $Z_{rc}^{dy} = Z_c^{dy} - Z_{lc}^{dy}$ and $Z_{rs}^{dy} = Z_s^{dy} - Z_{ls}^{dy}$ as the remaining dynamic causal factors and dynamic spurious factors. For $I(\Phi_{c,t}^{dy}; \Phi_{c,t}^{st,\star} | Y)$, suppose that the representations $\Phi_{c,t}^{dy}$ learned by the model contain both causal information and spurious information, *i.e.*, $\Phi_{c,t}^{dy} = (Z_{lc,t}^{dy}, Z_{ls,t}^{dy})$, then we rewrite it as:

$$I(\Phi_{c,t}^{dy}; \Phi_{c,t}^{st,\star} | Y) = I(Z_{lc,t}^{dy}, Z_{ls,t}^{dy}; Z_{c,t}^{st} | Y) = H(Z_{lc,t}^{dy}, Z_{ls,t}^{dy} | Y) + H(Z_{c,t}^{st} | Y) - H(Z_{lc,t}^{dy}, Z_{ls,t}^{dy}, Z_{c,t}^{st} | Y).$$
(33)

Analogously, $I(Z_{c,t}^{dy}; \Phi_{c,t}^{st,\star}|Y)$ is expanded as:

$$I(Z_{c,t}^{dy}; \Phi_{c,t}^{st,\star}|Y) = H(Z_{lc,t}^{dy}, Z_{rc,t}^{dy}|Y) + H(Z_{c,t}^{st}|Y) - H(Z_{lc,t}^{dy}, Z_{rc,t}^{dy}, Z_{c,t}^{st}|Y).\tag{34}$$

Hence, we have:

$$\begin{aligned}
\Delta I(\Phi_{c,t}^{dy}; \Phi_{c,t}^{st,\star}|Y) &= I(Z_{c,t}^{dy}; \Phi_{c,t}^{st,\star}|Y) - I(\Phi_{c,t}^{dy}; \Phi_{c,t}^{st,\star}|Y) \\
&= \underbrace{\left( H(Z_{lc,t}^{dy}, Z_{rc,t}^{dy}|Y) - H(Z_{lc,t}^{dy}, Z_{ls,t}^{dy}|Y) \right)}_{\Delta H_1} - \underbrace{\left( H(Z_{lc,t}^{dy}, Z_{rc,t}^{dy}, Z_{c,t}^{st}|Y) - H(Z_{lc,t}^{dy}, Z_{ls,t}^{dy}, Z_{c,t}^{st}|Y) \right)}_{\Delta H_2}.
\end{aligned}\tag{35}$$

For $\Delta H_1$, we have:

$$\begin{aligned}
\Delta H_1 &= H(Z_{lc,t}^{dy}, Z_{rc,t}^{dy}|Y) - H(Z_{lc,t}^{dy}, Z_{ls,t}^{dy}|Y) \\
&= \left( H(Z_{lc,t}^{dy}|Y) + H(Z_{rc,t}^{dy}|Z_{lc,t}^{dy}, Y) \right) - \left( H(Z_{lc,t}^{dy}|Y) + H(Z_{ls,t}^{dy}|Z_{lc,t}^{dy}, Y) \right) \\
&= H(Z_{rc,t}^{dy}|Z_{lc,t}^{dy}, Y) - H(Z_{ls,t}^{dy}|Z_{lc,t}^{dy}, Y),
\end{aligned}\tag{36}$$

As for $\Delta H_2$, it can be known that:

$$\begin{aligned}
\Delta H_2 &= H(Z_{lc,t}^{dy}, Z_{rc,t}^{dy}, Z_{c,t}^{st}|Y) - H(Z_{lc,t}^{dy}, Z_{ls,t}^{dy}, Z_{c,t}^{st}|Y) \\
&= \left( H(Z_{rc,t}^{dy}|Z_{lc,t}^{dy}, Z_{c,t}^{st}, Y) + H(Z_{lc,t}^{dy}, Z_{c,t}^{st}|Y) \right) - \left( H(Z_{ls,t}^{dy}|Z_{lc,t}^{dy}, Z_{c,t}^{st}, Y) + H(Z_{lc,t}^{dy}, Z_{c,t}^{st}|Y) \right) \\
&= H(Z_{rc,t}^{dy}|Z_{lc,t}^{dy}, Z_{c,t}^{st}, Y) - H(Z_{ls,t}^{dy}|Z_{lc,t}^{dy}, Z_{c,t}^{st}, Y).
\end{aligned}\tag{37}$$

By substituting Eq. (36) and Eq. (37) into Eq. (35), we obtain:

$$\begin{aligned}
\Delta I(\Phi_{c,t}^{dy}; \Phi_{c,t}^{st,\star}|Y) &= \Delta H_1 - \Delta H_2 \\
&= \left( H(Z_{rc,t}^{dy}|Z_{lc,t}^{dy}, Y) - H(Z_{ls,t}^{dy}|Z_{lc,t}^{dy}, Y) \right) - \left( H(Z_{rc,t}^{dy}|Z_{lc,t}^{dy}, Z_{c,t}^{st}, Y) - H(Z_{ls,t}^{dy}|Z_{lc,t}^{dy}, Z_{c,t}^{st}, Y) \right) \\
&= \underbrace{\left( H(Z_{rc,t}^{dy}|Z_{lc,t}^{dy}, Y) - H(Z_{rc,t}^{dy}|Z_{lc,t}^{dy}, Z_{c,t}^{st}, Y) \right)}_{\text{term 1}} - \underbrace{\left( H(Z_{ls,t}^{dy}|Z_{lc,t}^{dy}, Y) - H(Z_{ls,t}^{dy}|Z_{lc,t}^{dy}, Z_{c,t}^{st}, Y) \right)}_{\text{term 2}}.
\end{aligned}\tag{38}$$

From Eq. (38), since causal factors typically exhibit higher similarity to each other than to spurious factors, for a new given $Z_{c,t}^{st}$, the entropy decay with respect to $Z_{rc,t}^{dy}$ is generally greater than the entropy decay with respect to $Z_{ls,t}^{dy}$. Therefore, it can be derived that

$$I(Z_{c,t}^{dy}; \Phi_{c,t}^{st,\star}|Y) - I(\Phi_{c,t}^{dy}; \Phi_{c,t}^{st,\star}|Y) = \Delta I(\Phi_{c,t}^{dy}; \Phi_{c,t}^{st,\star}|Y) > 0,\tag{39}$$

Hence, dynamic causal factors $Z_{c,t}^{dy}$ is the maximizer of Eq. (32). This completes the proof. □

### B.3.4. PROOF OF THE MAIN RESULT

**Theorem 2.** *Let $\Phi_c^{st}(X_t)$, $\Phi_c^{dy}(X_t)$ and $Z_t^d$ denote the representations and drift factors at the time domain $\mathcal{D}_t$, obtained by training the network through the optimization of $\mathcal{L}_{SYNC}$. Then the predictor constructed upon these components is the optimal causal predictor as defined in Def. 2.*

*Proof.* As $\mathcal{L}_{SYNC} = \mathcal{L}_{evolve} + \alpha_1 \mathcal{L}_{MI} + \alpha_2 \mathcal{L}_{causal}$, according to Lem. 1 and Lem. 2, by minimizing losses $\mathcal{L}_{MI}$, $\mathcal{L}_{causal}$ and the cross-entropy loss in $\mathcal{L}_{evolve}$, $\Phi_c^{st}(X_t)$ and $\Phi_c^{dy}(X_t)$ will gradually approach the ground truth static causal factors $Z_c^{st}$ and dynamic causal factors $Z_c^{dy}$, which exclude any spurious factors. In addition, optimizing the loss $\mathcal{L}_{evolve}$ make the drift factors $Z_t^d$ approach the drift of causal mechanisms. According to d-separation, We can conclude that the predictor built on top of these satisfies $Y \perp\!\!\!\perp [Z_s^{st}, Z_s^{dy}] \mid [\Phi_c^{st}(X_t), \Phi_c^{dy}(X_t), Z_t^d]$. Meanwhile, the mutual information between representations and the target can be maximized by minimizing the cross-entropy loss in $\mathcal{L}_{evolve}$. Therefore, it can be concluded that the resulting predictor satisfies the criteria for the optimal causal predictor as defined in Def. 2. This completes the proof. □

# C. Implementation Details

## C.1. Mini-batch Weighted Sampling

Mini-batch weighted sampling (MWS) strategy, proposed by (Chen et al., 2018b), introduces a biased negative entropy estimator based on the principle of importance sampling. Although biased, this estimator is straightforward to use for estimation and does not require any additional hyperparameters. Suppose that the distribution of the dataset samples is $p(n)$, the dataset size is $N$, and let $\mathcal{B}_M = \{n_1, \cdots, n_M\}$ be a mini-batch of $M$ indices where each element is sampled i.i.d. from $p(n)$. Let $p(\mathcal{B}_M)$ be the probability of any mini-batch $\mathcal{B}_M$ sampled from the dataset, then

$$
\begin{aligned}
\mathbb{E}_{q(\boldsymbol{z})}[\log q(\boldsymbol{z})] &= \mathbb{E}_{q(\boldsymbol{z},n)}[\log q(\boldsymbol{z}(n))] \\
&= \mathbb{E}_{q(\boldsymbol{z},n)}[\log \mathbb{E}_{n' \sim p(n)} q(\boldsymbol{z}(n)|n')] \\
&= \mathbb{E}_{q(\boldsymbol{z},n)}[\log \mathbb{E}_{\mathcal{B}_M \sim p(\mathcal{B}_M)}[\frac{1}{M} \sum_{j=1}^{M} q(\boldsymbol{z}(n)|n_j)]] \\
&\geq \mathbb{E}_{q(\boldsymbol{z},n)}[\log \mathbb{E}_{\mathcal{B}_M \sim p(\mathcal{B}_M|n)}[\frac{p(\mathcal{B}_M)}{p(\mathcal{B}_M|n)} \sum_{j=1}^{M} q(\boldsymbol{z}(n)|n_j)]] \\
&= \mathbb{E}_{q(\boldsymbol{z},n)}[\log \mathbb{E}_{\mathcal{B}_M \sim p(\mathcal{B}_M|n)}[\frac{1}{NM} \sum_{j=1}^{M} q(\boldsymbol{z}(n)|n_j)]] \,,
\end{aligned}
\tag{40}
$$

therefore, $\mathbb{E}_{q(\boldsymbol{z})} \log q(\boldsymbol{z})$ can be estimated using the following estimator:

$$
\mathbb{E}_{q(\boldsymbol{z})} \log q(\boldsymbol{z}) = \frac{1}{M} \sum_{i=1}^{M} [\log[\frac{1}{NM} \sum_{j=1}^{M} q(\boldsymbol{z}(n_i)|n_j)]] \,.
\tag{41}
$$

## C.2. Gumbel-Softmax Trick

To sample a $k$-hot vector from an $N$-dimensional vector, we use the Gumbel-Softmax trick (Jang et al., 2017) and perform $k$ samplings without replacement. Specifically, let $\boldsymbol{s}$ be a score vector of dimension $N$, for the $i$-th ($i \in [k]$) sampling, using the Gumbel-Softmax trick, the sampled vector $\boldsymbol{m}^i$ is obtained by

$$
\boldsymbol{m}_j^i = \frac{\exp((\log \boldsymbol{s}_j^i + \xi_j^i)/\tau)}{\sum_{j'=1}^{N} \exp((\log \boldsymbol{s}_{j'}^i + \xi_{j'}^i)/\tau)}, \quad \xi_j^i = -\log(-\log u_j^i), u_j^i \sim \text{Uniform}(0, 1) \,,
\tag{42}
$$

where $\tau$ is the temperature hyperparameter. Following a common setting (Chen et al., 2018a), we set $\tau = 0.5$. After each sampling, we find the position of the largest element $p = \text{argmax}_{j \in [N]} \boldsymbol{m}_j^i$ and set $\boldsymbol{s}[p]$ to a very small value to acquire scores $\boldsymbol{s}^{i+1}$, then continue with the next sampling. This process continues for $k$ samplings, and the sampled $k$-hot vector $\boldsymbol{m} = \text{Gumbel-Softmax}(\boldsymbol{s}, k)$ is obtained by $\boldsymbol{m}_j = \max_{l \in [k]} \boldsymbol{m}_j^l$.

## C.3. Network Structure

For the static variational encoding network $q_\psi$, we utilize a feature extractor to implement. The dynamic variational encoding network $q_\theta$ can be implemented using a feature extractor with the same architecture as $q_\psi$ but without sharing network parameters, followed by an LSTM network. The prior $p(\boldsymbol{z}_t^{dy}|\boldsymbol{z}_{<t}^{dy})$ is obtained by $F^{dy}$, using a one-layer LSTM network. We implement $q_\zeta$ by a dynamic inference network, which takes the one-hot code of $y_t$ as the input and output the categorical distribution. The prior $p(\boldsymbol{z}_t^d|\boldsymbol{z}_{<t}^d)$ is acquired in a manner similar way to $p(\boldsymbol{z}_t^{dy}|\boldsymbol{z}_{<t}^{dy})$, and is modeled by a prior netweotk $F^d$. In order to reconstruct the data $\boldsymbol{x}_{1:T}$ and the labels $y_{1:T}$, we utilize the appropriate decoder $D$ and linear classifier $W$. To ensure fairness, the feature extractor and decoder in each task are used the same as LSSAE (Qin et al., 2022). The masker is implemented by a 3-layer MLP which is randomly initialized. During training stage, the sampling of $\boldsymbol{z}_t^{st}$ and $\boldsymbol{z}_t^{dy}$ are performed using the reparameterization trick (Kingma & Welling, 2014), and $\boldsymbol{z}_t^d$ is acquired using the Gumbel-Softmax reparameterization trick (Jang et al., 2017).

## C.4. Model Inference

In this work, to make the prediction of $\boldsymbol{x}_t$ sampled from the target domain $\mathcal{D}_t$, we need to leverage the static causal representation $\Phi_{c,t}^{st}$, the dynamic causal representation $\Phi_{c,t}^{dy}$ and the drift factor $Z_t^d$ simultaneously. Therefore, during inference stage, we use $q_\psi$ to infer $\Phi_t^{st}$, and $\Phi_{c,t}^{st}$ is obtained through the masker $m_c^{st}$, *i.e.*, $\Phi_{c,t}^{st} = m_c^{st}(\Phi_t^{st})$. For $Z_t^d$, we employ the trained network of prior $p(\boldsymbol{z}_t^{dy}|\boldsymbol{z}_{<t}^{dy})$ to infer without requirement of $q_\zeta$. To get $\Phi_{c,t}^{dy}$, we store the state variables of $q_\theta$ from the last training domain in the hidden state bank $\mathcal{B}$ at training time. Then, we randomly select a test batch from $\mathcal{B}$ and send to $q_\theta$ to infer $\Phi_t^{dy}$ from $\boldsymbol{x}_t$, and $\Phi_{c,t}^{dy} = m_c^{dy}(\Phi_t^{dy})$. At the same time, the state variables in $\mathcal{B}$ is updated to the state variables of the LSTM network in the current domain, in preparation for the prediction of the next domain. Ultimately, $\Phi_{c,t}^{st}$, $\Phi_{c,t}^{dy}$ and $Z_t^d$ are feed into the classifier $W$ for prediction.

# D. Algorithm of SYNC

The training and testing procedures of SYNC are shown in Algorithm 1 and 2.

---

**Algorithm 1** Training procedure for SYNC

---

1: **Input:** sequential source labeled datasets $\mathcal{S}$; static variational encoding network $q_\psi$; dynamic variational encoding networks $q_\theta, q_\zeta$ and their corresponding prior networks $F^{dy}, F^d$; decoder $D$ and classifier $W$; masker $m_c^{st}$ and $m_c^{dy}$; hidden state bank $\mathcal{B}$; total number of epoch $N_{epoch}$ and number of iterations per epoch $N_{it}$.
2: Randomly initialize all the networks.
3: **for** epoch in $1, 2, \cdots, N_{epoch}$ **do**
4:     Set the hidden state bank $\mathcal{B} = \emptyset$.
5:     **for** it in $1, 2, \cdots, N_{it}$ **do**
6:         Sample training date $(\boldsymbol{x}_{1:T}, y_{1:T})$ from $\mathcal{S}_{1:T}$.
7:         Assign $\boldsymbol{z}_0^{dy} \leftarrow \boldsymbol{0}, \boldsymbol{z}_0^d \leftarrow \boldsymbol{0}$, generate prior distributions $p(\boldsymbol{z}_t^{dy}|\boldsymbol{z}_{<t}^{dy}), p(\boldsymbol{z}_t^d|\boldsymbol{z}_{<t}^d), t = 1, ..., T$ via $F^{dy}, F^d$.
8:         Generate posterior distribution $q(\boldsymbol{z}_t^{st}|\boldsymbol{x}_t)$ via $q_\psi$, generate posterior distribution $q(\boldsymbol{z}_t^{dy}|\boldsymbol{z}_{<t}^{dy}, \boldsymbol{x}_t), t = 1, 2, ..., T$ and hidden state $\boldsymbol{st}_T$ via $q_\theta$, generate posterior distribution $q(\boldsymbol{z}_t^d|\boldsymbol{z}_{<t}^d, y_t), t = 1, 2, ..., T$ via $q_\zeta$.
9:         Send $\hat{\boldsymbol{z}}_{1:T}^{st}$ and $\hat{\boldsymbol{z}}_{1:T}^{dy}$ sampled from the posterior distributions of $q(\boldsymbol{z}_t^{st}|\boldsymbol{x}_t), t = 1, 2, ..., T$ and $q(\boldsymbol{z}_t^{dy}|\boldsymbol{z}_{<t}^{dy}, \boldsymbol{x}_t), t = 1, 2, ..., T$, along with the means of the posterior distributions $\boldsymbol{\mu}_{1:T}^{st}$ and $\boldsymbol{\mu}_{1:T}^{dy}$, to the corresponding mask module.
10:         ▷ Calculate $\mathcal{L}_{\text{evolve}}$ for $q_\psi, q_\theta, q_\zeta, F^{dy}, F^d, D, W, m_c^{st}$ and $m_c^{dy}$.
11:         ▷ Calculate $\mathcal{L}_{\text{MI}}$ for $q_\psi, q_\theta, q_\zeta, F^{dy}, F^d$.
12:         ▷ Calculate $\mathcal{L}_{\text{causal}}$ for $m_c^{st}$ and $m_c^{dy}$.
13:         Update all modules by the summary of these loss.
14:     **end for**
15: **end for**

---

---

**Algorithm 2** Testing procedure for SYNC

---

1: **Input:** sequential target datasets $\mathcal{T}$, static variational encoding network $q_\psi$; dynamic variational encoding networks $q_\theta$; prior network $F^d$; masker $m_c^{st}$ and $m_c^{dy}$; classifier $W$; hidden state bank $\mathcal{B}$.
2: Create a new bank $\mathcal{B}'$.
3: **for** $\mathcal{D}_t \in \mathcal{T}$ **do**
4:     Set $\mathcal{B}' = \emptyset$.
5:     **for** $\boldsymbol{x} \in \mathcal{D}_t$ **do**
6:         Randomly sample a hidden state $\boldsymbol{st}$ from the hidden state bank $\mathcal{B}$.
7:         Obtain $\boldsymbol{\mu}^{dy}$ and $\boldsymbol{st}'$ by feeding $\boldsymbol{x}$ and $\boldsymbol{st}$ into $q_\theta$. Store $\boldsymbol{st}'$ into $\mathcal{B}'$.
8:         Obtain $\boldsymbol{\mu}^{st}$ by feeding $\boldsymbol{x}$ into $q_\psi$. Sample $\boldsymbol{z}^d$ via $F^d$.
9:         Send $\boldsymbol{\mu}^{st}, \boldsymbol{\mu}^{dy}$ to the corresponding mask module, and use the resulting representations along with $\boldsymbol{z}^d$ to make prediction via $W$.
10:     **end for**
11:     Assign $\mathcal{B} \leftarrow \mathcal{B}'$.
12: **end for**

---

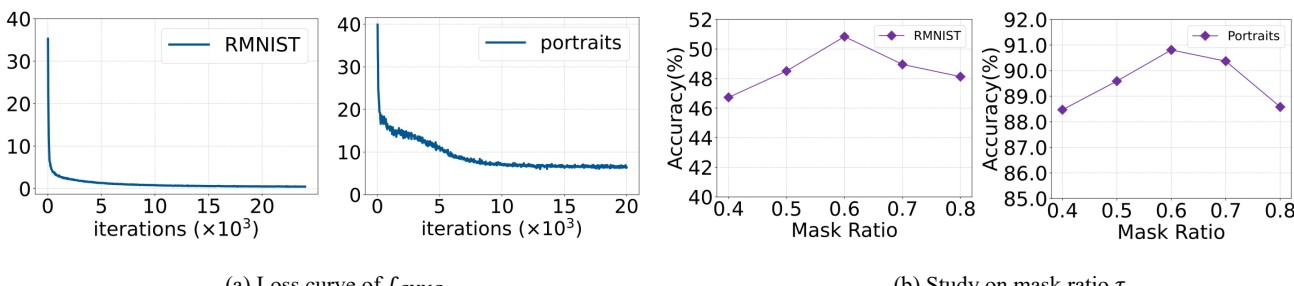

(a) Loss curve of $\mathcal{L}_{\text{SYNC}}$.

(b) Study on mask ratio $\tau$.

Figure 7: (a) The loss curve of $\mathcal{L}_{\text{SYNC}}$ on RMNIST and Portraits during the training phase. (b) Ablation study of SYNC to the mask ratio $\tau$ on RMNIST and Portraits.

## E. More Experimental Details

### E.1. More Details on Datasets

**Circle** (Pesaranghader & Viktor, 2016) contains evolving 30 domains (15 source domains, 5 validation domains, and 10 target domains), where the instance are sampled from 30 2D Gaussian distributions.

**Sine** (Pesaranghader & Viktor, 2016) includes 24 evolving domains (12 source domains, 4 validation domains, and 8 target domains), which is achieved by extending and rearranging the original dataset.

**Rotated MNIST (RMNIST)** (Ghifary et al., 2015) is composed of MNIST digits with various rotations. We follow the approach outlined in (Qin et al., 2022) and extend it to 19 evolving domains (10 source domains, 3 validation domains, and 6 target domains) by applying the rotations with degree of $\{0°, 15°, 30°, \cdots, 180°\}$ in order on each domain.

**Portraits** (Yao et al., 2022a) is a real-word dataset consisting of photos of American high school seniors over 108 years (1905-2013) across 26 states. We divide it into 34 evolving domains (19 source domains, 5 validation domains, and 10 target domains) by a fixed internal over time.

**Caltran** (Hoffman et al., 2014) is a surveillance dataset captured by fixed traffic cameras deployed at intersections. It is split into 34 domains (19 source domains, 5 validation domains, and 10 target domains) by time to predict the type of scene.

**PowerSupply** (Dau et al., 2019) is created by an Italian electricity company and contains 30 evolving domains (15 source domains, 5 validation domains, and 10 target domains) for predicting the current power supply based on the hourly records.

**ONP** (Fernandes et al., 2015) dataset documents articles collected from the Mashable website within two years (2013-2015), aiming to predict the number of shares in social networks. This dataset is divided into 24 domains (12 source domains, 4 validation domains, and 8 target domains) according to month.

### E.2. Training Details

Table 3: Training details on different datasets.

| Dataset | B | Epochs | Optimizer | Learning Rate | $\alpha_1$ | $\alpha_2$ | $\tau$ | $N$ |
|---------|-----|--------|-----------|---------------|-----------|-----------|--------|------|
| Circle | 64 | 30 | Adam | 5e-6 | 1 | 0.02 | 0.6 | 20 |
| Sine | 64 | 50 | Adam | 1e-5 | 1 | 0.001 | 0.6 | 32 |
| RMNIST | 48 | 120 | Adam | 5e-4 | 0.005 | 0.001 | 0.6 | 32 |
| Portraits | 24 | 100 | Adam | 5e-5 | 0.05 | 0.002 | 0.6 | 32 |
| Caltran | 24 | 100 | Adam | 1e-5 | 0.005 | 0.002 | 0.6 | 32 |
| PowerSupply | 64 | 50 | Adam | 5e-6 | 0.001 | 0.01 | 0.6 | 32 |
| ONP | 64 | 50 | Adam | 5e-6 | 0.001 | 0.01 | 0.6 | 32 |

Training details on different datasets are shown in Table 3, where **B** denotes the batch size, $\alpha_1$ and $\alpha_2$ represent the trade-off hyper-parameter for the loss function $\mathcal{L}_{\text{MI}}$ and $\mathcal{L}_{\text{causal}}$, respectively. $\tau$ is the mask ratio of the masker and $N$ represents the dimension of the latent space. All experiments in this work are performed on a single NVIDIA GeForce RTX 4090 GPU with 24GB memory, using the PyTorch packages. Following (Qin et al., 2022), the intermediate domains are utilized as validation set for model selection.

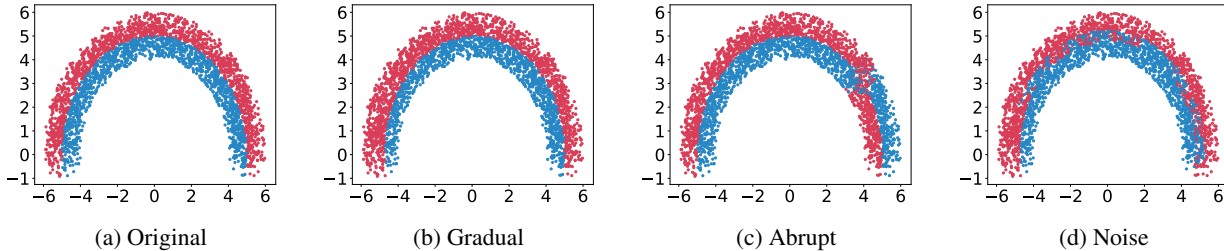

(a) Original          (b) Gradual          (c) Abrupt          (d) Noise

Figure 8: Visualization of decision boundary, where the positive and negative classes are colored in red and blue. (a)-(d) show the decision boundary of Circle-Orignal, Circle-Gradual, Circle-Abrupt and Circle-Noise, respectively.

## E.3. Additional Results

### E.3.1. COMPUTATIONAL COST

To evaluate the computational cost of our approach, we conduct experiments on the RMNIST and Portraits datasets, recording the memory cost and runtime per iteration. As shown in Table 4 and Table 5, it can be found that our method is comparable among the considered methods. Specifically, the two indicators of our method are roughly similar to those of LSSAE and MMD-LSAE, with the memory cost lower than that of MMD-LSAE and the runtime per iteration lower than that of GI. Moreover, it is worth noting that SYNC consistently outperforms all baseline methods, achieving an average accuracy improvement of at least $1.2\%$ and the lowest accuracy improvement of $5.3\%$ across all datasets. This suggests that only a slight increase in computational cost is required to achieve performance improvements.

Table 4: Comparison of memory cost (GB) on different datasets.

| Dataset | LSSAE | MMD-LSAE | GI | SYNC |
|---|---|---|---|---|
| RMNIST | 5.06 | 5.24 | 3.81 | 5.19 |
| Portraits | 17.01 | 17.20 | 14.48 | 17.14 |

Table 5: Comparison of runtime per iteration (s) during the training phase on different datasets.

| Dataset | LSSAE | MMD-LSAE | GI | SYNC |
|---|---|---|---|---|
| RMNIST | 0.12 | 0.13 | 2.80 | 0.17 |
| Portraits | 0.35 | 0.36 | 7.84 | 0.42 |

### E.3.2. CONVERGENCE ANALYSIS

In Fig. 7 (a), we plot the loss curve of our method during training phase. From the results, it can be observed that the total loss of SYNC decreases steadily until convergence on RMNIST and Portraits. This suggests that, although our loss function comprises multiple components, the optimization path formed by their combination remains relatively smooth, with the components working synergistically to promote convergence. By optimizing $\mathcal{L}_{\text{SYNC}}$, the model can progressively capture the underlying evolving patterns in the data distribution and learn time-aware causal representations, thereby mitigating the influence of spurious correlations that may arise from data bias, ultimately enhancing the temporal generalization.

### E.3.3. STUDY ON DIFFERENT MASK RATIO

We conduct an experiment on RMNSIT and Portraits to explore the impact of mask ratio $\tau$ in Fig. 7 (b). It can be found that SYNC is a little bit sensitive to $\tau$. Based on empirical results, a mask ratio of $0.6$ proves more effective for extracting causal factors, thereby enhancing generalization performance. Model performance improves when the proportion of true causal factors within the mixed factors aligns more closely with the selected mask ratio.

### E.3.4. MORE DISTRIBUTION DRIFT TYPES

In order to further verify the robustness of our proposed method, we follow (Qin et al., 2022; 2023) and conduct experiments on three additional circle-based distribution drift datasets (Circle-Gradual, Circle-Abrupt and Circle-Noise) to demonstrate

Table 6: The comparison of the accuracy (%) between SYNC and other baselines. "Wst" denotes the worst performance of the dataset on all test domains, and "Avg" denotes the average performance of the dataset on all test domains. The best result are marked in **bold**.

| Method | Gradual | | Abrupt | | Noise | | Overall | |
|---|---|---|---|---|---|---|---|---|
| | Wst | Avg | Wst | Avg | Wst | Avg | Wst | Avg |
| LSSAE | 32.0 | 53.1 | 44.0 | 57.8 | 29.0 | 53.8 | 35.0 | 54.9 |
| MMD-LSAE | 34.4 | 65.5 | 45.0 | **68.7** | 32.0 | 53.8 | 37.1 | 62.7 |
| SYNC | **49.0** | **67.2** | **60.0** | 66.7 | **69.0** | **77.0** | **59.3** | **70.3** |

the effectiveness of SYNC in more complex scenarios. As shown in Fig. 8, the decision boundary of Circle-Gradual changes gradually, and decision boundary of Circle-Abrupt changes suddenly at a certain moment, while the decision boundary of Circle-Noise fluctuates irregularly due to the influence of noise. These three drift types can simply simulate slow variations, drastic fluctuations, and irregular variations in real-world data distribution. We evaluate our method with LSSAE (Qin et al., 2022) and MMD-LSAE (Qin et al., 2023) on these synthetic datasets. As shown in Table 6, our method significantly outperforms the baselines in both metrics, achieving at least 12.2% and 7.6% improvements in Wst and Avg on average across all datasets, respectively. This suggests that SYNC has the potential to handle more complex and diverse drift types.

### E.4. Full Experimental Results

We provide complete experimental results of SYNC and other baselines. It can be found that in most scenarios, our approach achieve the state-of-the-art performance, showing the effectiveness of SYNC in non-stationary tasks. It is worth noting that for the experiment of SDE-EDG on PowerSupply, we report the results using the same backbone architecture as LSSAE to ensure a fair comparison.

Table 7: Circle. We show the results on each target domain by domain index.

| ALGORITHM | 21 | 22 | 23 | 24 | 25 | 26 | 27 | 28 | 29 | 30 | Avg. |
|---|---|---|---|---|---|---|---|---|---|---|---|
| ERM | $53.9 \pm 3.5$ | $55.8 \pm 4.8$ | $53.9 \pm 5.2$ | $44.7 \pm 6.3$ | $56.9 \pm 4.4$ | $47.8 \pm 5.8$ | $41.9 \pm 7.5$ | $41.7 \pm 5.9$ | $54.2 \pm 3.0$ | $47.8 \pm 5.7$ | 49.9 |
| Mixup | $48.6 \pm 3.8$ | $51.7 \pm 4.0$ | $49.4 \pm 4.5$ | $43.6 \pm 5.8$ | $56.9 \pm 4.4$ | $47.8 \pm 5.8$ | $41.9 \pm 7.5$ | $41.7 \pm 5.9$ | $54.2 \pm 3.0$ | $47.8 \pm 5.7$ | 48.4 |
| MMD | $50.0 \pm 3.9$ | $53.6 \pm 4.4$ | $55.0 \pm 4.3$ | $51.9 \pm 6.0$ | $60.8 \pm 3.9$ | $49.7 \pm 7.2$ | $41.9 \pm 7.5$ | $41.7 \pm 5.9$ | $54.2 \pm 3.0$ | $47.8 \pm 5.7$ | 50.7 |
| MLDG | $57.8 \pm 3.6$ | $57.7 \pm 5.0$ | $55.3 \pm 4.9$ | $46.4 \pm 6.8$ | $56.9 \pm 4.4$ | $47.8 \pm 5.8$ | $41.9 \pm 7.5$ | $41.7 \pm 5.9$ | $54.2 \pm 3.0$ | $47.8 \pm 5.7$ | 50.8 |
| RSC | $45.3 \pm 3.6$ | $51.4 \pm 3.8$ | $49.4 \pm 4.5$ | $43.6 \pm 5.8$ | $56.9 \pm 4.4$ | $47.8 \pm 5.8$ | $41.9 \pm 7.5$ | $41.7 \pm 5.9$ | $54.2 \pm 3.0$ | $47.8 \pm 5.7$ | 48.0 |
| MTL | $61.4 \pm 2.2$ | $57.2 \pm 6.4$ | $53.3 \pm 5.1$ | $48.3 \pm 6.2$ | $56.9 \pm 4.8$ | $49.2 \pm 3.7$ | $43.3 \pm 5.0$ | $45.8 \pm 2.8$ | $54.2 \pm 5.7$ | $42.2 \pm 4.9$ | 51.2 |
| Fish | $51.7 \pm 3.7$ | $53.1 \pm 3.7$ | $49.4 \pm 4.5$ | $43.6 \pm 5.8$ | $56.9 \pm 4.4$ | $47.8 \pm 5.8$ | $41.9 \pm 7.5$ | $41.7 \pm 5.9$ | $54.2 \pm 3.0$ | $47.8 \pm 5.7$ | 48.8 |
| CORAL | $65.3 \pm 3.2$ | $63.9 \pm 4.4$ | $60.0 \pm 4.8$ | $56.4 \pm 6.0$ | $60.2 \pm 4.3$ | $47.8 \pm 5.8$ | $41.9 \pm 7.5$ | $41.7 \pm 5.9$ | $54.2 \pm 3.0$ | $47.8 \pm 5.7$ | 53.9 |
| AndMask | $42.8 \pm 3.4$ | $50.6 \pm 4.1$ | $49.4 \pm 4.5$ | $43.6 \pm 5.8$ | $56.9 \pm 4.4$ | $47.8 \pm 5.8$ | $41.9 \pm 7.5$ | $41.7 \pm 5.9$ | $54.2 \pm 3.0$ | $49.7 \pm 5.4$ | 47.9 |
| DIVA | $81.3 \pm 3.5$ | $76.3 \pm 4.2$ | $74.7 \pm 4.6$ | $56.7 \pm 5.1$ | $67.0 \pm 6.1$ | $62.3 \pm 5.1$ | $62.0 \pm 5.6$ | $66.3 \pm 4.1$ | $70.3 \pm 5.6$ | $62.0 \pm 4.2$ | 67.9 |
| IRM | $57.8 \pm 3.9$ | $59.4 \pm 5.4$ | $56.9 \pm 4.9$ | $48.1 \pm 7.4$ | $57.5 \pm 4.3$ | $47.8 \pm 5.8$ | $41.9 \pm 7.5$ | $41.7 \pm 5.9$ | $54.2 \pm 3.0$ | $47.8 \pm 5.7$ | 51.3 |
| IIB | $79.0 \pm 3.3$ | $68.0 \pm 4.1$ | $58.0 \pm 4.5$ | $44.0 \pm 3.2$ | $55.0 \pm 4.1$ | $49.0 \pm 4.2$ | $42.0 \pm 4.1$ | $45.0 \pm 2.5$ | $55.0 \pm 3.3$ | $44.0 \pm 3.0$ | 53.9 |
| iDAG | $52.0 \pm 3.4$ | $53.0 \pm 3.7$ | $51.0 \pm 4.1$ | $44.0 \pm 3.6$ | $55.0 \pm 4.2$ | $49.0 \pm 3.3$ | $42.0 \pm 3.9$ | $45.0 \pm 2.8$ | $55.0 \pm 3.4$ | $44.0 \pm 3.6$ | 49.0 |
| GI | $72.0 \pm 4.9$ | $62.0 \pm 4.9$ | $61.0 \pm 4.9$ | $58.0 \pm 4.9$ | $56.0 \pm 4.7$ | $49.0 \pm 4.7$ | $42.0 \pm 4.2$ | $45.0 \pm 4.5$ | $55.0 \pm 4.0$ | $44.0 \pm 4.1$ | 54.4 |
| LSSAE | $95.8 \pm 1.9$ | $95.6 \pm 2.1$ | $93.5 \pm 2.9$ | $96.3 \pm 1.8$ | $83.8 \pm 5.2$ | $74.3 \pm 3.6$ | $51.9 \pm 5.6$ | $52.3 \pm 8.1$ | $46.5 \pm 9.2$ | $48.4 \pm 5.3$ | 73.8 |
| DDA | $71.0 \pm 0.7$ | $56.0 \pm 0.0$ | $51.0 \pm 0.0$ | $44.0 \pm 2.8$ | $55.0 \pm 0.7$ | $49.0 \pm 2.1$ | $42.0 \pm 7.1$ | $45.0 \pm 0.7$ | $55.0 \pm 3.5$ | $44.0 \pm 4.2$ | 51.2 |
| DRAIN | $48.0 \pm 2.1$ | $52.0 \pm 0.7$ | $54.0 \pm 3.5$ | $47.0 \pm 1.4$ | $58.0 \pm 3.5$ | $52.0 \pm 3.5$ | $45.0 \pm 4.2$ | $48.0 \pm 4.9$ | $58.0 \pm 4.9$ | $47.0 \pm 2.8$ | 50.7 |
| MMD-LSAE | $93.0 \pm 2.1$ | $89.0 \pm 1.3$ | $83.0 \pm 2.6$ | $84.0 \pm 2.0$ | $84.0 \pm 4.4$ | $88.0 \pm 3.5$ | $79.0 \pm 5.0$ | $76.0 \pm 6.4$ | $54.0 \pm 6.1$ | $65.0 \pm 5.0$ | 79.5 |
| CTOT | $86.0 \pm 1.0$ | $88.0 \pm 1.7$ | $89.0 \pm 1.5$ | $87.0 \pm 1.0$ | $77.0 \pm 2.5$ | $79.0 \pm 1.2$ | $72.0 \pm 1.2$ | $65.0 \pm 3.0$ | $55.0 \pm 0.9$ | $54.0 \pm 1.2$ | 75.2 |
| SDE-EDG | $95.0 \pm 0.7$ | $99.0 \pm 1.4$ | $98.0 \pm 1.4$ | $94.0 \pm 1.4$ | $95.0 \pm 1.4$ | $93.0 \pm 7.8$ | $74.0 \pm 5.7$ | $66.0 \pm 0.7$ | $45.0 \pm 6.4$ | $56.0 \pm 4.5$ | 81.5 |
| SYNC | $87.0 \pm 1.7$ | $95.0 \pm 2.1$ | $87.0 \pm 2.1$ | $94.0 \pm 1.5$ | $88.0 \pm 3.2$ | $84.0 \pm 3.3$ | $85.0 \pm 4.5$ | $80.0 \pm 5.5$ | $67.0 \pm 6.0$ | $80.0 \pm 4.7$ | 84.7 |

Table 8: Sine. We show the results on each target domain by domain index.

| ALGORITHM | 17 | 18 | 19 | 20 | 21 | 22 | 23 | 24 | Avg. |
|---|---|---|---|---|---|---|---|---|---|
| ERM | 71.4 ± 6.1 | 91.0 ± 1.5 | 81.6 ± 2.4 | 53.4 ± 2.9 | 51.1 ± 6.7 | 54.3 ± 4.7 | 49.5 ± 4.8 | 51.7 ± 5.0 | 63.0 |
| MIXUP | 63.1 ± 5.9 | 93.5 ± 1.7 | 80.6 ± 3.8 | 52.8 ± 2.9 | 60.3 ± 7.2 | 54.2 ± 2.7 | 49.5 ± 4.4 | 49.3 ± 8.0 | 62.9 |
| MMD | 57.0 ± 4.2 | 57.1 ± 4.1 | 47.6 ± 5.4 | 50.0 ± 1.8 | 55.1 ± 6.7 | 54.4 ± 4.7 | 49.5 ± 4.8 | 51.7 ± 5.0 | 55.8 |
| MLDG | 69.2 ± 4.2 | 67.7 ± 4.1 | 52.1 ± 5.4 | 50.7 ± 1.8 | 51.1 ± 6.7 | 54.3 ± 4.7 | 49.5 ± 4.8 | 51.7 ± 5.0 | 63.2 |
| RSC | 61.3 ± 6.6 | 83.5 ± 1.9 | 84.5 ± 2.6 | 52.8 ± 2.8 | 55.1 ± 6.7 | 54.4 ± 4.7 | 49.5 ± 4.8 | 51.7 ± 5.0 | 61.5 |
| MTL | 70.6 ± 6.6 | 91.6 ± 1.2 | 79.9 ± 3.4 | 51.0 ± 4.7 | 60.3 ± 7.6 | 53.6 ± 5.2 | 49.5 ± 5.3 | 46.9 ± 5.9 | 62.9 |
| Fish | 66.1 ± 6.9 | 82.0 ± 2.7 | 87.5 ± 2.4 | 55.2 ± 3.0 | 51.1 ± 6.7 | 54.3 ± 4.7 | 49.5 ± 4.8 | 51.7 ± 5.0 | 62.3 |
| CORAL | 60.0 ± 5.3 | 57.1 ± 4.2 | 48.6 ± 6.4 | 50.7 ± 1.8 | 49.7 ± 6.2 | 48.6 ± 4.6 | 46.3 ± 5.0 | 51.7 ± 5.0 | 51.6 |
| AndMASK | 44.2 ± 5.1 | 42.9 ± 4.2 | 54.2 ± 7.0 | 71.9 ± 1.9 | 86.4 ± 3.2 | 90.4 ± 2.9 | 88.1 ± 3.4 | 76.4 ± 3.7 | 69.3 |
| DIVA | 79.0 ± 6.6 | 60.8 ± 1.9 | 47.6 ± 2.6 | 50.0 ± 2.8 | 55.1 ± 6.7 | 51.9 ± 4.7 | 38.6 ± 4.8 | 40.4 ± 5.0 | 52.9 |
| IRM | 66.9 ± 6.2 | 81.1 ± 3.2 | 88.5 ± 3.0 | 56.6 ± 6.0 | 57.2 ± 5.8 | 53.7 ± 5.1 | 49.5 ± 2.2 | 51.7 ± 5.4 | 63.2 |
| IIB | 56.0 ± 5.4 | 79.1 ± 2.2 | 86.9 ± 1.3 | 58.3 ± 4.0 | 55.1 ± 4.9 | 54.4 ± 5.4 | 49.5 ± 3.6 | 50.7 ± 4.1 | 61.3 |
| iDAG | 86.0 ± 3.2 | 63.0 ± 4.3 | 49.4 ± 3.7 | 50.0 ± 3.8 | 55.1 ± 4.5 | 54.3 ± 5.1 | 49.6 ± 3.8 | 50.0 ± 3.3 | 57.1 |
| GI | 77.0 ± 0.7 | 84.2 ± 4.4 | 89.1 ± 0.6 | 60.9 ± 3.6 | 55.1 ± 0.8 | 53.5 ± 6.0 | 49.8 ± 4.1 | 52.0 ± 2.8 | 65.2 |
| LSSAE | 93.0 ± 1.7 | 86.9 ± 0.7 | 69.2 ± 1.5 | 63.8 ± 3.8 | 68.8 ± 2.5 | 76.8 ± 4.8 | 63.9 ± 1.3 | 49.0 ± 3.1 | 71.4 |
| DDA | 43.0 ± 0.7 | 47.2 ± 0.6 | 74.2 ± 1.6 | 89.6 ± 1.1 | 75.5 ± 1.1 | 70.0 ± 7.1 | 59.1 ± 2.2 | 74.0 ± 1.4 | 66.6 |
| DRAIN | 43.0 ± 4.9 | 44.6 ± 4.7 | 69.3 ± 0.9 | 89.6 ± 1.1 | 82.8 ± 4.8 | 78.5 ± 2.5 | 70.2 ± 4.1 | 92.0 ± 5.7 | 71.3 |
| MMD-LSAE | 43.0 ± 3.2 | 43.9 ± 4.4 | 60.7 ± 2.3 | 79.7 ± 1.9 | 96.9 ± 0.7 | 98.8 ± 0.2 | 90.5 ± 2.3 | 57.3 ± 5.4 | 71.4 |
| CTOT | 43.2 ± 1.9 | 52.6 ± 1.2 | 62.1 ± 0.9 | 87.4 ± 1.2 | 78.4 ± 2.8 | 71.6 ± 1.2 | 77.4 ± 1.3 | 65.3 ± 2.7 | 67.3 |
| SDE-EDG | 99.0 ± 0.7 | 96.9 ± 0.6 | 90.0 ± 4.2 | 88.8 ± 0.6 | 64.9 ± 2.2 | 52.4 ± 0.3 | 43.0 ± 4.9 | 42.3 ± 4.0 | 72.2 |
| SYNC | 67.0 ± 4.0 | 89.0 ± 1.6 | 84.5 ± 3.2 | 79.8 ± 1.8 | 75.5 ± 2.8 | 68.9 ±1.3 | 60.0 ± 2.7 | 84.0 ± 3.1 | 76.0 |

Table 9: RMNIST. We show the results on each target domain denoted by rotation angle.

| ALGORITHM | 130° | 140° | 150° | 160° | 170° | 180° | Avg. |
|---|---|---|---|---|---|---|---|
| ERM | 56.8 ± 0.9 | 44.2 ± 0.8 | 37.8 ± 0.6 | 38.3 ± 0.8 | 40.9 ± 0.8 | 43.6 ± 0.8 | 43.6 |
| MIXUP | 61.3 ± 0.7 | 47.4 ± 0.8 | 39.1 ± 0.7 | 38.3 ± 0.7 | 40.5 ± 0.8 | 42.8 ± 0.9 | 44.9 |
| MMD | 59.2 ± 0.9 | 46.0 ± 0.8 | 39.0 ± 0.7 | 39.3 ± 0.8 | 41.6 ± 0.7 | 43.7 ± 0.8 | 44.8 |
| MLDG | 57.4 ± 0.7 | 44.5 ± 0.9 | 37.5 ± 0.8 | 37.5 ± 0.8 | 39.9 ± 0.8 | 42.0 ± 0.9 | 43.1 |
| RSC | 54.1 ± 0.9 | 41.9 ± 0.8 | 35.8 ± 0.7 | 37.0 ± 0.8 | 39.8 ± 0.8 | 41.6 ± 0.8 | 41.7 |
| MTL | 54.8 ± 0.9 | 43.1 ± 0.8 | 36.4 ± 0.8 | 36.3 ± 0.8 | 39.1 ± 0.9 | 40.9 ± 0.8 | 41.7 |
| FISH | 60.8 ± 0.8 | 47.8 ± 0.8 | 39.2 ± 0.8 | 37.6 ± 0.7 | 39.0 ± 0.8 | 40.7 ± 0.7 | 44.2 |
| CORAL | 58.8 ± 0.9 | 46.2 ± 0.8 | 38.9 ± 0.7 | 38.5 ± 0.8 | 41.3 ± 0.8 | 43.5 ± 0.8 | 44.5 |
| ANDMASK | 53.5 ± 0.9 | 42.9 ± 0.8 | 37.8 ± 0.7 | 38.6 ± 0.8 | 40.8 ± 0.8 | 43.2 ± 0.8 | 42.8 |
| DIVA | 58.3 ± 0.8 | 45.0 ± 0.8 | 37.6 ± 0.8 | 36.9 ± 0.7 | 38.1 ± 0.8 | 40.1 ± 0.8 | 42.7 |
| IRM | 47.7 ± 0.9 | 38.5 ± 0.7 | 34.1 ± 0.7 | 35.7 ± 0.8 | 37.8 ± 0.8 | 40.3 ± 0.8 | 39.0 |
| IIB | 59.4 ± 0.7 | 49.5 ± 0.8 | 42.4 ± 0.9 | 38.5 ± 0.8 | 39.6 ± 0.7 | 40.3 ± 0.8 | 45.0 |
| iDAG | 56.7 ± 0.9 | 44.6 ± 0.8 | 39.8 ± 0.9 | 39.8 ± 0.7 | 41.6 ± 0.8 | 41.9 ± 0.9 | 44.1 |
| GI | 61.6 ± 0.9 | 46.4 ± 0.9 | 39.2 ± 0.8 | 40.0 ± 0.8 | 40.1 ± 0.8 | 40.1 ± 0.7 | 44.6 |
| LSSAE | 64.1 ± 0.8 | 51.6 ± 0.8 | 43.4 ± 0.8 | 38.6 ± 0.7 | 40.3 ± 0.8 | 40.4 ± 0.8 | 46.4 |
| DDA | 60.7 ± 0.8 | 50.0 ± 0.8 | 42.6 ± 0.8 | 39.6 ± 0.8 | 38.0 ± 0.8 | 39.7 ± 0.8 | 45.1 |
| DRAIN | 59.5 ± 0.8 | 45.4 ± 0.8 | 40.2 ± 0.7 | 37.2 ± 0.7 | 39.6 ± 0.8 | 41.0 ± 0.7 | 43.8 |
| MMD-LSAE | 65.8 ± 0.8 | 53.6 ± 0.8 | 46.6 ± 0.8 | 43.1 ± 0.8 | 42.9 ± 0.8 | 43.5 ± 0.8 | 49.2 |
| CTOT | 68.2 ± 0.7 | 55.3 ± 0.8 | 44.9 ± 0.9 | 36.7 ± 0.8 | 32.3 ± 0.8 | 31.7 ± 0.8 | 44.8 |
| SDE-EDG | 75.1 ± 0.8 | 61.3 ± 0.9 | 49.8 ± 0.8 | 49.8 ± 0.8 | 39.7 ± 0.7 | 39.7 ± 0.9 | 52.6 |
| SYNC | 65.1 ± 0.7 | 52.8 ± 0.9 | 48.5 ± 1.1 | 45.8 ± 0.9 | 46.3 ± 1.3 | 46.6 ± 0.9 | 50.8 |

Table 10: Portraits. We show the results on each target domain by domain index.

| ALGORITHM | 25 | 26 | 27 | 28 | 29 | 30 | 31 | 32 | 33 | 34 | Avg. |
|---|---|---|---|---|---|---|---|---|---|---|---|
| ERM | 75.5 ± 0.9 | 83.8 ± 0.9 | 88.5 ± 0.8 | 93.3 ± 0.7 | 93.4 ± 0.6 | 92.1 ± 0.7 | 90.6 ± 0.8 | 84.3 ± 0.9 | 88.5 ± 0.9 | 87.9 ± 1.4 | 87.8 |
| MIXUP | 75.5 ± 0.9 | 83.8 ± 0.9 | 88.5 ± 0.8 | 93.3 ± 0.7 | 93.4 ± 0.6 | 92.1 ± 0.7 | 90.6 ± 0.8 | 84.3 ± 0.9 | 88.5 ± 0.9 | 87.9 ± 1.4 | 87.8 |
| MMD | 74.0 ± 1.0 | 83.8 ± 0.8 | 87.2 ± 0.8 | 93.0 ± 0.7 | 93.0 ± 0.6 | 91.9 ± 0.7 | 90.9 ± 0.7 | 84.7 ± 1.4 | 88.3 ± 0.9 | 85.8 ± 1.8 | 87.3 |
| MLDG | 76.4 ± 0.8 | 85.5 ± 0.9 | 90.1 ± 0.7 | 94.3 ± 0.6 | 93.5 ± 0.6 | 92.0 ± 0.7 | 90.8 ± 0.8 | 85.6 ± 1.1 | 89.3 ± 0.8 | 87.6 ± 1.6 | 88.5 |
| RSC | 75.2 ± 0.9 | 84.7 ± 0.8 | 87.9 ± 0.7 | 93.3 ± 0.7 | 92.5 ± 0.7 | 91.0 ± 0.7 | 90.0 ± 0.7 | 84.6 ± 1.2 | 88.2 ± 0.8 | 85.8 ± 1.9 | 87.3 |
| MTL | 78.2 ± 0.9 | 86.5 ± 0.8 | 90.9 ± 0.8 | 94.2 ± 0.7 | 93.8 ± 0.6 | 92.0 ± 0.7 | 91.2 ± 0.7 | 86.0 ± 1.2 | 89.3 ± 0.8 | 87.4 ± 1.4 | 89.0 |
| Fish | 78.6 ± 0.9 | 86.9 ± 0.8 | 89.5 ± 0.8 | 93.5 ± 0.7 | 93.3 ± 0.6 | 92.1 ± 0.6 | 91.1 ± 0.7 | 86.2 ± 1.3 | 88.7 ± 0.9 | 87.7 ± 1.6 | 88.8 |
| CORAL | 74.6 ± 0.9 | 84.6 ± 0.8 | 87.9 ± 0.8 | 93.3 ± 0.6 | 92.7 ± 0.7 | 91.5 ± 0.7 | 90.7 ± 0.7 | 84.6 ± 1.5 | 88.1 ± 0.9 | 85.9 ± 1.9 | 87.4 |
| AndMASK | 62.0 ± 1.1 | 70.8 ± 1.1 | 67.0 ± 1.2 | 70.2 ± 1.1 | 75.2 ± 1.1 | 74.1 ± 1.0 | 72.7 ± 1.1 | 64.7 ± 1.6 | 77.3 ± 1.1 | 74.9 ± 2.1 | 70.9 |
| DIVA | 76.2 ± 1.0 | 86.6 ± 0.8 | 88.8 ± 0.8 | 93.5 ± 0.7 | 93.1 ± 0.6 | 91.6 ± 0.6 | 91.1 ± 0.7 | 84.7 ± 1.3 | 89.1 ± 0.8 | 87.0 ± 1.5 | 88.2 |
| IRM | 74.2 ± 0.9 | 83.5 ± 0.9 | 88.5 ± 0.8 | 91.0 ± 0.8 | 90.4 ± 0.7 | 87.3 ± 0.8 | 87.0 ± 0.9 | 80.4 ± 1.5 | 86.7 ± 0.9 | 85.1 ± 1.8 | 85.4 |
| IIB | 78.1 ± 1.3 | 87.2 ± 1.1 | 91.8 ± 1.2 | 95.8 ± 0.8 | 94.9 ± 0.7 | 92.2 ± 1.4 | 91.7 ± 1.2 | 87.6 ± 1.1 | 89.5 ± 1.4 | 88.4 ± 1.5 | 89.7 |
| iDAG | 79.8 ± 1.1 | 86.2 ± 1.4 | 91.3 ± 0.8 | 94.0 ± 0.7 | 92.6 ± 0.9 | 89.3 ± 1.1 | 89.8 ± 1.3 | 87.4 ± 1.4 | 87.9 ± 1.1 | 88.4 ± 1.3 | 88.6 |
| GI | 77.8 ± 1.2 | 86.6 ± 1.3 | 90.8 ± 1.1 | 95.3 ± 1.3 | 93.1 ± 1.2 | 89.3 ± 1.1 | 88.9 ± 1.2 | 84.1 ± 1.8 | 87.7 ± 1.0 | 87.5 ± 2.0 | 88.1 |
| LSSAE | 77.7 ± 0.9 | 87.1 ± 0.8 | 90.8 ± 0.7 | 94.3 ± 0.6 | 94.3 ± 0.6 | 92.2 ± 0.6 | 91.2 ± 0.7 | 86.7 ± 1.1 | 89.6 ± 0.8 | 86.9 ± 1.4 | 89.1 |
| DDA | 76.0 ± 1.0 | 85.6 ± 0.8 | 88.6 ± 0.8 | 93.6 ± 0.6 | 92.9 ± 0.7 | 92.9 ± 0.6 | 90.3 ± 0.8 | 84.3 ± 1.2 | 88.7 ± 0.8 | 85.9 ± 1.2 | 87.9 |
| DRAIN | 77.7 ± 0.8 | 86.2 ± 0.8 | 90.6 ± 0.6 | 94.8 ± 0.5 | 94.4 ± 0.6 | 92.8 ± 0.7 | 92.2 ± 0.6 | 87.2 ± 1.2 | 89.9 ± 0.8 | 87.9 ± 1.1 | 89.4 |
| MMD-LSAE | 80.9 ± 0.9 | 88.6 ± 0.8 | 92.8 ± 0.7 | 95.0 ± 0.6 | 94.9 ± 0.5 | 91.0 ± 0.9 | 92.3 ± 0.6 | 88.1 ± 1.2 | 90.4 ± 0.8 | 89.7 ± 1.1 | 90.4 |
| CTOT | 77.9 ± 0.9 | 83.0 ± 0.8 | 79.2 ± 0.8 | 88.0 ± 0.7 | 90.4 ± 0.6 | 89.8 ± 0.8 | 87.9 ± 0.8 | 83.9 ± 1.6 | 87.8 ± 0.8 | 86.3 ± 1.1 | 86.4 |
| SDE-EDG | 78.6 ± 0.8 | 86.6 ± 0.9 | 90.1 ± 0.8 | 94.8 ± 0.6 | 94.5 ± 0.6 | 93.3 ± 0.7 | 92.1 ± 0.7 | 87.9 ± 1.3 | 89.6 ± 0.9 | 89.0 ± 1.1 | 89.6 |
| SYNC | 81.0 ± 1.1 | 88.8 ± 0.9 | 93.5 ± 1.2 | 96.0 ± 0.7 | 95.7 ± 0.6 | 93.1 ± 0.9 | 92.6 ± 0.8 | 87.8 ± 1.3 | 90.7 ± 0.9 | 89.0 ± 1.2 | 90.8 |

Table 11: Caltran. We show the results on each target domain by domain index.

| ALGORITHM | 25 | 26 | 27 | 28 | 29 | 30 | 31 | 32 | 33 | 34 | Avg. |
|---|---|---|---|---|---|---|---|---|---|---|---|
| ERM | 29.9 ± 3.5 | 88.4 ± 2.1 | 61.1 ± 3.5 | 56.3 ± 3.2 | 90.0 ± 1.6 | 60.1 ± 2.5 | 55.5 ± 3.5 | 88.8 ± 2.4 | 57.1 ± 3.5 | 50.5 ± 5.2 | 66.3 |
| MIXUP | 53.6 ± 3.9 | 89.0 ± 2.0 | 61.8 ± 2.4 | 55.7 ± 2.9 | 88.2 ± 2.1 | 58.6 ± 3.0 | 52.3 ± 3.7 | 88.6 ± 2.7 | 57.1 ± 3.0 | 55.1 ± 4.3 | 66.0 |
| MMD | 30.2 ± 2.1 | 92.7 ± 1.7 | 56.4 ± 3.7 | 39.1 ± 3.2 | 93.6 ± 1.7 | 52.1 ± 3.2 | 42.8 ± 3.0 | 92.1 ± 2.2 | 42.1 ± 3.8 | 29.4 ± 3.8 | 57.1 |
| MLDG | 54.8 ± 4.1 | 88.6 ± 2.6 | 62.2 ± 3.6 | 55.1 ± 4.1 | 88.3 ± 1.7 | 60.9 ± 4.3 | 51.7 ± 2.6 | 89.0 ± 1.9 | 56.5 ± 3.4 | 55.3 ± 4.8 | 66.2 |
| RSC | 57.2 ± 3.0 | 88.4 ± 2.6 | 62.6 ± 3.0 | 56.5 ± 3.7 | 88.0 ± 2.4 | 59.4 ± 3.0 | 51.9 ± 2.9 | 90.0 ± 2.0 | 59.4 ± 2.9 | 56.0 ± 3.1 | 67.0 |
| MTL | 64.2 ± 3.0 | 87.2 ± 2.5 | 64.9 ± 3.9 | 60.0 ± 4.8 | 84.5 ± 2.2 | 60.6 ± 3.5 | 52.6 ± 3.7 | 83.9 ± 2.9 | 58.2 ± 4.1 | 65.7 ± 5.6 | 68.2 |
| Fish | 61.1 ± 3.5 | 88.2 ± 1.5 | 64.7 ± 4.0 | 57.9 ± 3.1 | 88.3 ± 2.2 | 59.9 ± 3.0 | 57.5 ± 2.7 | 87.4 ± 2.8 | 57.7 ± 3.7 | 63.0 ± 6.1 | 68.6 |
| CORAL | 50.4 ± 3.0 | 90.8 ± 2.0 | 61.2 ± 3.8 | 55.2 ± 3.5 | 92.0 ± 1.7 | 55.8 ± 3.0 | 52.0 ± 3.8 | 90.9 ± 1.6 | 56.8 ± 2.4 | 50.9 ± 5.6 | 65.7 |
| AndMASK | 30.0 ± 2.2 | 92.7 ± 1.7 | 56.2 ± 3.8 | 39.1 ± 3.2 | 93.6 ± 1.7 | 51.6 ± 3.2 | 42.6 ± 2.9 | 92.1 ± 2.2 | 41.2 ± 3.7 | 29.9 ± 3.6 | 56.9 |
| DIVA | 60.6 ± 2.9 | 90.1 ± 1.7 | 67.5 ± 3.1 | 58.9 ± 3.5 | 88.4 ± 2.8 | 58.7 ± 3.3 | 53.8 ± 3.6 | 89.8 ± 1.7 | 61.8 ± 4.8 | 62.0 ± 3.4 | 69.2 |
| IRM | 46.4 ± 3.7 | 90.8 ± 1.7 | 60.8 ± 3.4 | 52.9 ± 3.1 | 91.8 ± 1.7 | 56.6 ± 3.1 | 52.1 ± 2.9 | 90.9 ± 2.6 | 55.6 ± 3.9 | 43.1 ± 5.5 | 64.1 |
| IIB | 62.2 ± 4.1 | 93.8 ± 1.1 | 68.0 ± 2.9 | 55.1 ± 3.3 | 91.4 ± 1.8 | 60.8 ± 3.0 | 54.8 ± 3.1 | 89.4 ± 2.2 | 58.4 ± 3.7 | 68.1 ± 5.1 | 69.3 |
| iDAG | 58.3 ± 3.8 | 91.2 ± 1.2 | 68.5 ± 3.2 | 56.2 ± 2.8 | 88.8 ± 1.9 | 63.2 ± 3.1 | 53.9 ± 2.2 | 89.1 ± 2.6 | 69.4 ± 3.5 | 53.5 ± 4.1 | 69.7 |
| GI | 68.8 ± 3.1 | 86.6 ± 1.9 | 65.5 ± 3.2 | 60.6 ± 4.3 | 88.8 ± 2.4 | 58.5 ± 3.6 | 53.1 ± 2.8 | 88.7 ± 2.1 | 63.7 ± 2.9 | 73.0 ± 5.1 | 70.7 |
| LSSAE | 63.4 ± 3.4 | 92.1 ± 2.0 | 62.6 ± 4.7 | 58.8 ± 4.4 | 92.9 ± 1.6 | 62.0 ± 3.9 | 54.3 ± 3.0 | 92.1 ± 2.2 | 60.5 ± 3.8 | 67.4 ± 3.6 | 70.6 |
| DDA | 31.0 ± 3.4 | 92.6 ± 1.8 | 56.8 ± 0.9 | 59.0 ± 2.8 | 94.0 ± 2.2 | 61.7 ± 2.3 | 52.9 ± 2.3 | 92.9 ± 2.3 | 57.8 ± 5.3 | 62.9 ± 5.8 | 66.1 |
| DRAIN | 66.4 ± 3.3 | 83.8 ± 1.0 | 65.7 ± 2.2 | 62.8 ± 3.2 | 77.9 ± 3.3 | 62.3 ± 3.7 | 55.7 ± 3.7 | 78.9 ± 3.1 | 60.6 ± 3.8 | 75.7 ± 5.6 | 69.0 |
| MMD-LSAE | 61.3 ± 3.7 | 87.4 ± 1.9 | 65.7 ± 3.3 | 60.4 ± 4.9 | 85.7 ± 2.7 | 60.4 ± 3.1 | 56.9 ± 2.6 | 85.2 ± 2.4 | 58.3 ± 2.6 | 75.0 ± 4.9 | 69.6 |
| CTOT | 48.0 ± 1.4 | 88.0 ± 1.0 | 64.0 ± 1.7 | 54.0 ± 2.1 | 93.0 ± 3.6 | 57.0 ± 1.9 | 48.0 ± 3.2 | 85.0 ± 2.6 | 61.0 ± 2.8 | 71.0 ± 2.6 | 66.9 |
| SDE-EDG | 70.5 ± 2.9 | 88.8 ± 4.9 | 66.1 ± 2.8 | 55.1 ± 2.6 | 85.1 ± 3.6 | 59.5 ± 4.4 | 58.6 ± 3.3 | 88.2 ± 3.4 | 68.9 ± 3.5 | 72.2 ± 5.7 | 71.3 |
| SYNC | 58.8 ± 2.6 | 90.7 ± 1.3 | 71.1 ± 3.1 | 59.4 ± 4.5 | 91.3 ± 2.3 | 63.5 ± 3.5 | 65.0 ± 2.5 | 90.7 ± 2.1 | 68.8 ± 3.3 | 62.6 ± 4.5 | 72.2 |

Table 12: PowerSupply. We show the results on each target domain by domain index.

| ALGORITHM | 21 | 22 | 23 | 24 | 25 | 26 | 27 | 28 | 29 | 30 | Avg. |
|---|---|---|---|---|---|---|---|---|---|---|---|
| ERM | 69.8 ± 1.4 | 70.0 ± 1.4 | 69.2 ± 1.3 | 64.4 ± 1.5 | 85.8 ± 1.0 | 76.0 ± 1.3 | 70.1 ± 1.5 | 69.8 ± 1.5 | 69.0 ± 1.3 | 65.5 ± 1.5 | 71.0 |
| MIXUP | 69.6 ± 1.4 | 69.5 ± 1.5 | 68.3 ± 1.5 | 64.3 ± 1.5 | 87.1 ± 1.0 | 76.6 ± 1.3 | 70.1 ± 1.4 | 69.2 ± 1.3 | 68.1 ± 1.5 | 65.0 ± 1.6 | 70.8 |
| MMD | 70.0 ± 1.3 | 69.7 ± 1.4 | 68.7 ± 1.4 | 64.8 ± 1.5 | 85.6 ± 1.0 | 76.1 ± 1.3 | 70.0 ± 1.5 | 69.5 ± 1.4 | 68.7 ± 1.3 | 65.6 ± 1.5 | 70.9 |
| MLDG | 69.7 ± 1.4 | 69.7 ± 1.5 | 68.6 ± 1.5 | 64.6 ± 1.5 | 86.4 ± 1.1 | 76.3 ± 1.4 | 70.1 ± 1.4 | 69.4 ± 1.3 | 68.4 ± 1.5 | 65.6 ± 1.5 | 70.8 |
| RSC | 69.9 ± 1.4 | 69.6 ± 1.4 | 68.6 ± 1.4 | 64.4 ± 1.5 | 86.6 ± 1.0 | 76.3 ± 1.3 | 70.0 ± 1.5 | 69.4 ± 1.4 | 68.4 ± 1.3 | 65.4 ± 1.5 | 70.9 |
| MTL | 69.6 ± 1.4 | 69.4 ± 1.5 | 68.2 ± 1.6 | 64.2 ± 1.5 | 87.4 ± 1.2 | 76.6 ± 1.3 | 69.9 ± 1.5 | 69.1 ± 1.5 | 68.2 ± 1.5 | 64.6 ± 1.4 | 70.7 |
| Fish | 69.7 ± 1.4 | 69.4 ± 1.4 | 68.2 ± 1.4 | 64.2 ± 1.4 | 87.3 ± 1.0 | 76.6 ± 1.3 | 69.9 ± 1.5 | 69.2 ± 1.5 | 68.2 ± 1.3 | 65.2 ± 1.5 | 70.8 |
| CORAL | 69.9 ± 1.4 | 69.7 ± 1.5 | 68.9 ± 1.4 | 64.6 ± 1.4 | 86.1 ± 1.0 | 76.3 ± 1.3 | 70.0 ± 1.5 | 69.5 ± 1.5 | 68.8 ± 1.3 | 65.7 ± 1.5 | 71.0 |
| ANDMASK | 69.9 ± 1.4 | 69.4 ± 1.4 | 68.2 ± 1.3 | 64.0 ± 1.4 | 87.4 ± 0.9 | 76.7 ± 1.3 | 70.0 ± 1.5 | 69.1 ± 1.5 | 68.0 ± 1.3 | 64.7 ± 1.5 | 70.7 |
| DIVA | 69.7 ± 1.4 | 69.5 ± 1.3 | 68.2 ± 1.4 | 63.9 ± 1.5 | 87.5 ± 1.0 | 76.5 ± 1.3 | 69.9 ± 1.5 | 69.1 ± 1.5 | 68.1 ± 1.3 | 64.7 ± 1.5 | 70.7 |
| IRM | 69.8 ± 1.4 | 69.5 ± 1.4 | 68.3 ± 1.4 | 64.5 ± 1.4 | 87.2 ± 0.9 | 76.5 ± 1.3 | 70.0 ± 1.5 | 69.1 ± 1.5 | 68.2 ± 1.3 | 65.0 ± 1.4 | 70.8 |
| IIB | 69.4 ± 1.3 | 69.5 ± 1.4 | 68.2 ± 1.4 | 64.5 ± 1.4 | 86.9 ± 1.1 | 76.4 ± 1.2 | 69.9 ± 1.4 | 69.0 ± 1.5 | 68.1 ± 1.3 | 65.1 ± 1.4 | 70.8 |
| iDAG | 70.7 ± 1.4 | 70.4 ± 1.4 | 70.0 ± 1.2 | 66.1 ± 1.3 | 83.5 ± 1.1 | 76.3 ± 1.1 | 70.4 ± 1.4 | 70.4 ± 1.3 | 68.9 ± 1.3 | 66.7 ± 1.4 | 71.2 |
| GI | 70.2 ± 1.4 | 71.0 ± 1.4 | 70.5 ± 1.5 | 69.6 ± 1.5 | 80.7 ± 1.1 | 68.4 ± 1.3 | 72.9 ± 1.5 | 72.0 ± 1.3 | 71.8 ± 1.3 | 66.5 ± 1.5 | 71.4 |
| LSSAE | 70.0 ± 1.4 | 69.8 ± 1.4 | 69.0 ± 1.5 | 65.4 ± 1.4 | 85.1 ± 1.1 | 76.0 ± 1.4 | 70.1 ± 1.7 | 69.9 ± 1.3 | 69.0 ± 1.6 | 66.3 ± 1.4 | 71.1 |
| DDA | 69.8 ± 1.6 | 72.4 ± 1.5 | 70.5 ± 1.5 | 63.8 ± 1.5 | 83.7 ± 1.2 | 73.1 ± 1.2 | 70.1 ± 1.3 | 71.4 ± 1.5 | 70.5 ± 1.7 | 63.4 ± 1.2 | 70.9 |
| DRAIN | 70.1 ± 1.3 | 70.0 ± 1.0 | 69.3 ± 1.1 | 65.5 ± 1.5 | 83.6 ± 1.0 | 75.8 ± 1.7 | 70.3 ± 1.3 | 69.8 ± 1.5 | 68.9 ± 1.9 | 66.4 ± 1.2 | 71.0 |
| MMD-LSAE | 69.9 ± 1.4 | 74.3 ± 1.3 | 71.8 ± 1.5 | 65.2 ± 1.4 | 80.1 ± 1.4 | 70.0 ± 1.8 | 70.7 ± 1.5 | 74.0 ± 1.4 | 72.4 ± 1.4 | 66.0 ± 1.6 | 71.4 |
| CTOT | 70.3 ± 1.1 | 71.4 ± 1.4 | 73.7 ± 1.2 | 70.4 ± 1.2 | 72.0 ± 1.2 | 63.6 ± 1.6 | 75.2 ± 1.2 | 71.1 ± 1.1 | 72.8 ± 1.2 | 70.6 ± 1.2 | 71.1 |
| SDE-EDG | 69.8 ± 1.1 | 69.5 ± 1.3 | 68.2 ± 1.1 | 64.1 ± 1.2 | 87.4 ± 1.1 | 76.5 ± 1.2 | 70.0 ± 1.5 | 69.0 ± 1.5 | 68.0 ± 1.2 | 65.0 ± 1.1 | 70.8 |
| SYNC | 70.2 ± 1.3 | 72.5 ± 1.2 | 71.7 ± 1.1 | 66.9 ± 1.4 | 80.0 ± 1.2 | 73.7 ± 1.4 | 70.6 ± 1.2 | 72.7 ± 1.4 | 71.1 ± 1.2 | 67.3 ± 1.1 | 71.7 |

Table 13: ONP. We show the results on each target domain by domain index.

| ALGORITHM | 17 | 18 | 19 | 20 | 21 | 22 | 23 | 24 | Avg. |
|---|---|---|---|---|---|---|---|---|---|
| ERM | 66.6 ± 1.1 | 66.7 ± 1.2 | 66.3 ± 1.1 | 67.0 ± 1.1 | 67.0 ± 1.1 | 64.2 ± 1.0 | 64.9 ± 1.1 | 64.6 ± 1.1 | 65.9 |
| Mixup | 67.0 ± 1.2 | 67.3 ± 1.1 | 66.1 ± 1.1 | 67.2 ± 1.1 | 66.7 ± 1.2 | 64.1 ± 1.1 | 64.5 ± 1.0 | 65.0 ± 1.1 | 66.0 |
| MMD | 58.9 ± 1.1 | 52.5 ± 1.3 | 54.6 ± 1.0 | 52.8 ± 1.2 | 48.1 ± 1.1 | 47.1 ± 1.0 | 49.2 ± 1.1 | 47.4 ± 1.2 | 51.3 |
| MLDG | 67.0 ± 1.2 | 67.0 ± 1.1 | 66.0 ± 1.1 | 66.3 ± 1.1 | 66.8 ± 1.1 | 63.9 ± 1.0 | 65.0 ± 1.1 | 65.4 ± 1.1 | 65.9 |
| RSC | 64.9 ± 1.2 | 65.2 ± 1.2 | 65.0 ± 1.1 | 66.0 ± 1.1 | 65.5 ± 1.1 | 62.6 ± 1.1 | 64.7 ± 1.1 | 63.3 ± 1.1 | 64.7 |
| MTL | 67.1 ± 1.1 | 66.5 ± 1.2 | 65.7 ± 1.1 | 66.0 ± 1.0 | 66.5 ± 1.0 | 63.4 ± 1.0 | 64.8 ± 1.1 | 65.2 ± 1.2 | 65.6 |
| Fish | 67.2 ± 1.2 | 66.8 ± 1.2 | 65.7 ± 1.1 | 66.8 ± 1.2 | 67.3 ± 1.1 | 63.2 ± 1.0 | 65.1 ± 1.2 | 64.7 ± 1.1 | 65.9 |
| CORAL | 66.5 ± 1.2 | 66.9 ± 1.2 | 65.9 ± 1.1 | 66.6 ± 1.1 | 67.0 ± 1.1 | 63.8 ± 1.0 | 64.5 ± 1.1 | 64.9 ± 1.1 | 65.8 |
| AndMask | 59.5 ± 1.2 | 56.1 ± 1.2 | 56.4 ± 1.1 | 56.0 ± 1.2 | 52.4 ± 1.2 | 51.9 ± 1.1 | 53.0 ± 1.1 | 51.2 ± 1.2 | 54.6 |
| DIVA | 67.8 ± 1.1 | 67.2 ± 1.3 | 66.4 ± 1.1 | 66.7 ± 1.2 | 67.3 ± 1.1 | 63.4 ± 1.1 | 65.0 ± 1.1 | 64.5 ± 1.2 | 66.0 |
| IRM | 66.1 ± 1.1 | 65.1 ± 1.2 | 64.7 ± 1.1 | 65.2 ± 1.2 | 64.9 ± 1.2 | 61.9 ± 1.1 | 63.4 ± 1.2 | 64.3 ± 1.1 | 64.5 |
| IIB | 66.6 ± 1.2 | 67.5 ± 1.2 | 66.8 ± 1.1 | 67.3 ± 1.1 | 66.4 ± 1.1 | 63.2 ± 1.2 | 66.4 ± 1.2 | 65.1 ± 1.1 | 66.3 |
| iDAG | 68.3 ± 1.2 | 67.8 ± 1.2 | 67.8 ± 1.1 | 66.2 ± 1.2 | 67.0 ± 1.2 | 63.8 ± 1.1 | 66.2 ± 1.3 | 63.9 ± 1.2 | 66.4 |
| GI | 67.6 ± 1.2 | 66.9 ± 1.1 | 66.3 ± 1.1 | 66.4 ± 1.3 | 67.1 ± 1.1 | 64.4 ± 1.2 | 64.9 ± 1.2 | 63.6 ± 1.2 | 65.9 |
| LSSAE | 64.7 ± 1.3 | 66.2 ± 1.4 | 66.6 ± 1.0 | 67.1 ± 1.0 | 67.6 ± 1.0 | 64.5 ± 1.0 | 64.9 ± 1.1 | 66.4 ± 1.1 | 66.0 |
| DDA | 67.7 ± 1.2 | 66.2 ± 1.1 | 66.6 ± 1.2 | 66.3 ± 1.2 | 66.8 ± 1.2 | 63.7 ± 1.1 | 64.7 ± 1.3 | 64.7 ± 1.1 | 65.8 |
| DRAIN | 60.9 ± 1.1 | 60.7 ± 1.2 | 59.8 ± 1.2 | 61.2 ± 1.1 | 61.6 ± 1.2 | 60.6 ± 1.2 | 61.9 ± 1.1 | 61.8 ± 1.1 | 65.8 |
| MMD-LSAE | 61.8 ± 1.0 | 66.4 ± 1.3 | 66.7 ± 1.1 | 67.2 ± 1.2 | 67.8 ± 0.9 | 64.6 ± 1.0 | 65.4 ± 1.1 | 65.4 ± 1.2 | 66.4 |
| CTOT | 62.7 ± 1.4 | 66.3 ± 1.3 | 65.5 ± 0.8 | 66.5 ± 1.2 | 63.9 ± 1.1 | 63.2 ± 1.2 | 65.3 ± 1.0 | 61.1 ± 1.2 | 64.3 |
| SDE-EDG | 67.8 ± 1.2 | 66.8 ± 0.8 | 65.6 ± 1.0 | 66.7 ± 1.0 | 66.9 ± 0.9 | 63.1 ± 1.1 | 64.7 ± 1.1 | 64.6 ± 1.1 | 65.6 |
| SYNC | 66.4 ± 1.2 | 66.0 ± 1.0 | 65.0 ± 1.1 | 65.9 ± 1.1 | 66.9 ± 1.0 | 64.9 ± 1.2 | 64.6 ± 1.1 | 64.6 ± 1.2 | 65.6 |

