# OpenReview forum: "Learning Time-Aware Causal Representation for Model Generalization in Evolving Domains"
_ICML.cc/2025/Conference — ICML 2025 poster_

### Official Review · Reviewer_c4gP · 2025-03-10

**Overall Recommendation:** 3

**Summary:**

To solve the problem of poor evolving domain generalization caused by spurious correlation between data and targets across domains, this paper proposes a time-aware structural causal model with static-dynamic causal representation learning (SYNC). SYNC introduces mutual information to constrain the model to learn static-dynamic causal representations, produces good causal predictors by preserving intra-class compactness of causal factors both across and within domains. The results show that SYNC has better time generalization.

The author's response addressed most of my concerns about this article, so I update the score to weak accept.

**Claims And Evidence:**

Most of the claims in this paper are clear and convincing, but I still have some concerns:
Modeling the drift factors $Z^d$ may not be necessary. In the structural cause model, the drift factor $Z^d$ and the dynamic factor $Z^{dy}$ are very similar. They are both affected by the local variable L and they both affect the label Y, which means that their distributions both shift as L changes, and $Z^d$ can be viewed as a dynamic factor.

**Essential References Not Discussed:**

N/A

**Experimental Designs Or Analyses:**

I have checked the experimental designs and analyses of this paper, including the predicted performance comparison experiment of all algorithms on all datasets, and the ablation experiments of SYNC on the RMNIST dataset. This paper also shows the decision boundary of the algorithms, the change of decision accuracy in different domains and the independence of dynamic and static representations. Although the authors present a large number of results, I still have some concerns:
1. This paper only shows the results of the algorithms in the last few domains. The results of the algorithms in all domains need to be compared to demonstrate the generalization ability of SYNC in different domains.
2. Figures 5 and 6 compare the results of only a few algorithms, and these algorithms are not the best baselines for the corresponding datasets. Please show the results of other baselines, or explain the reasons for presenting results from only those baselines.

**Methods And Evaluation Criteria:**

1. In Eq. (1), $z_{<t}^{dy}$ is the condition for the prior distribution and the posterior distribution of the dynamic factor $z_t^{dy}$, but we do not know the prior distribution $p(z_{<t}^{dy})$ when we infer the posterior distribution of $z_t^{dy}$. I think the true posterior distribution of $z_t^{dy}$ is $q_{\theta}(z_{t}^{dy}|x_{<t})$ or $q_{\theta}(z_{t}^{dy}|x_{≤t})$.
2. Eq. (3) minimizes the mutual information between static and dynamic representations to disentangle $z^{st}_t$ and $z^{dy}_t$, but Eq. (8) maximizes their mutual information to align static and dynamic representations of the same category within the domain. These two goals seem to be in complete conflict.
3. Eq. (11) uses the label $y_t$ to learn the drift factor $z^d$, and then $z^d$ is used to predict the label $\hat{y}_t$. There can be serious information leakage, because the true label and the predicted label may be directly correlated by the model. In extreme cases, the drift factor learned by the model may be invalid.

**Other Comments Or Suggestions:**

1. There are some symbol errors in the paper that need to be corrected. For example, $Z^{gc}$ and $Z^{lc}$ in the formula below definition 3.1 and “variant E” in the ablation study.

**Other Strengths And Weaknesses:**

Strengths：
1. This paper is important for the study of evolving domain generalization. It introduces causal learning to solve the problem of spurious correlation between data and targets modeled by existing models.
2. This paper is innovative to some extent, introducing mutual information to constrain the model to learn dynamic and static representations.

Weakness：
1. The contribution of this paper to solving spurious associations is limited. Although this paper proposes a structural causal model that divides potential factors into causal and spurious factors, this paper uses mutual information to constrain the model to disentangle dynamic and static representations, without disentangling causal and spurious factors. Therefore, whether the proposed model solves the spurious association needs further proof.
2. The method in this paper is not reasonable. For example, the posterior distribution of $z^{dy}_t$ in Eq. (1) is conditional on unknown variables, Eq. (3) and (8) conflict, and Eq. (11) has label information leakage.
3. The experimental results in this paper are incomplete. Authors should present the results of algorithms in all domains of the dataset, and should highlight the results compared to the optimal baseline under different datasets.

**Questions For Authors:**

1. Can the author explain in more detail how DYNC disentangles causal factors and spurious factors? I think the author overemphasizes disentangled dynamic and static representations and ignores the more important goal, which is to eliminate the spurious associations between data and targets.

**Relation To Broader Scientific Literature:**

This paper attempts to solve the problem of modeling spurious associations between data and targets across domains. Prior to this, LSSAE and MMD-LSAE had attempted to model evolving patterns.

**Theoretical Claims:**

I have checked the proofs of Proposition 3.3 and Theorem 3.6 provided in this paper, and the proof of Theorem 3.6 May be wrong. Taking the true label as the model input can cause the true label and the predicted label to be directly associated by the model and cause the model to learn invalid representations.

---

> ### Author Rebuttal · Authors · 2025-04-01
>
> > Q1. Modeling the drift factors may be not necessary.
>
> R1. We argue that modeling $Z^d$ is indispensable. Although both $Z^d$ and $Z^{dy}$ are influenced by $L$ and contribute to $Y$, $Z^{dy}$ operates in the feature space while $Z^d$ pertains to the label space which is designed to capture mechanism drifts. To be clear, $Z^d$ essentially characterizes the evolution of the classifier itself, which is crucial in EDG, as also emphasized in [1-2]. Furthermore, we conduct an ablation study on $Z^d$ and show result below, which clearly demonstrates the necessity of modeling drift factors.
>
> [1] LSSAE, ICML’22.
>
> [2] DDA, AAAI’23.
>
> ||RMNIST|Portraits|Overall
> |-|-|-|-
> |SYNC|50.8|90.8|70.8
> |SYNC w/o $Z^d$|47.6|89.9|68.8
>
> > Q2. Regarding the posterior distribution model for $z_t^{dy}$.
>
> R2. We clarify that the posterior $q_{\theta}(z_t^{dy}|z_{<t}^{dy},x_t)$ in our model is valid.
>
> In VAE frameworks, the variational posterior $q_\theta(z_t^{dy}|z_{<t}^{dy},x_t)$ is naturally learned through the inference network. The conditional dependency on $z_{<t}^{dy}$ reflects a data-driven modeling of temporal dynamics, independent of prior knowledge about $p(z_{<t}^{dy})$. Crucially, while both modeling $z_{<t}^{dy}$ and $x_{<t}^{dy}$ are reasonable, $z_{<t}^{dy}$ represents a ​low-dimensional encoding of $x_{<t}^{dy}$, compressed via LSTM hidden states to bypass high-dimensional data handling. This approach aligns with established sequence modeling practices [3], where latent variables effectively capture temporal dependencies.
>
> [3] S3VAE, CVPR’20.
>
> > Q3. About conflicting optimization goals.
>
> R3. We believe that the reviewer may have some misunderstandings. First, the misunderstanding fundamentally arises from a misinterpretation of the collider structure $Z^{st} \rightarrow Y \leftarrow Z^{dy}$, where $Z^{st} \perp Z^{dy}$, $Z^{st} \not \perp Z^{dy}|Y$ hold. This is a well-established principle in causal science, commonly referred to as the collider effect. Therefore, these two objectives are reasonable. In addition, the second objective does not directly take the static and dynamic factors as input. Instead, it operates on the representations processed by the causalizer.
>
> > Q4. Misunderstanding about information leakage during inference.
>
> R4. We respectfully disagree with the reviewer's view. During inference, static and dynamic causal factors are obtained by corresponding VAE encoder and causalizer, and the drift factors are inferred solely from the learned prior $p(z_t^d|z_{<t}^d)$, without access to ground-truth labels. Hence, the concern about possible information leakage may be misplaced. Please refer to Appendix. D for full details about training and inference procedures of our approach.
>
> > Q5. Regarding for showing results in domains.
>
> R5. We stress that EDG is inherently designed for future-domain generalization. Accordingly, following the common practice adopted by most existing EDG methods such as [1, 4], we evaluate on the last third of the domains to assess this capability. However, to address potential concerns, we also report the average test performance across all domains on Circle and RMNIST, demonstrating the superiority of SYNC.
>
> ||MMD-LSAE|SDE-EDG|SYNC
> |-|-|-|-
> |Circle|92.3|91.1|93.0
> |RMNIST|81.2|81.7|82.0
>
> [4] SDE-EDG, ICLR’24.
>
> > Q6. Lack of sufficient comparison with other baselines in Figures 5 and 6.
>
> R6. For Fig. 6(a-b), LSSAE [1] is chosen as the baseline since other methods don't explicitly model both static and dynamic factors. Although MMD-LSAE, an extension of LSSAE, considers these factors, it models them deterministically, making mutual information (MI) computation infeasible under our MWS-based estimation scheme. For the remaining figures, we have added the optimal baselines and redrawn them, as shown in Fig. 3 and Fig. 4 of https://anonymous.4open.science/r/2150-1CBD. The results demonstrate that SYNC still outperforms other methods, as claimed.
>
> [5] MMD-LSAE, TPAMI'23.
>
> > Q7. Clarification of this work's contribution to resolving spurious correlations.
>
> R7. Our method disentangles static and dynamic causal factors to learn time-aware causal representations and address spurious correlations. It works as follows: First, MI $I(z_t^{st},z_t^{dy})$ is minimized to disentangle the factors. Then, Proposition 3.3, Lemma 3.4, and Lemma 3.5 guide the learning process. Specifically, static causal factors can be identified by maximizing conditional MI between static components of consecutive domains given the class (Eq. (4)). Dynamic causal factors can be obtained by anchoring on static causal factors and maximizing conditional MI within the same domain given both the class and static causal factors (Eq. (8)). Based on these theoretical results, we use causalizer modules to extract finer-grained static and dynamic causal factors by optimizing the above objectives.
>
> > Q8. Regarding for some symbol errors.
>
> R8. Thanks. We will fix these errors and thoroughly revise the manuscript.

---

> > ### Comment · Reviewer_c4gP · 2025-04-02
> >
> > Thank you for your response, which answered some of my questions. But I still have some concerns:
> >
> > 1. I admit that it is necessary to infer the drift factor $Z^d$, which helps the model understand cross-domain information, but SYNC does not distinguish between $Z^d$ and $Z^{dy}$. Similar to SYNC inferring static and dynamic factors from the overall data and past data, what designs have the authors introduced to ensure that $Z^d$ and $Z^{dy}$ learn different information?
> >
> > 2. I understand that min-max mutual information can ensure that the causality between static factor $Z^{st}$, dynamic factor $Z^{dy}$ and label $y$ conforms to the collider structure. In general, when we train a model with a min-max goal, we optimize some parameters to maximize the goal, and optimize others to minimize the goal, such as adversarial learning (or vice versa). Does SYNC do something similar? Maximizing mutual information and minimizing mutual information in SYNC seem to be done by updating the same set of parameters. I suggest a more reasonable approach to max-min optimization.
> >
> > 3. I believe SYNC does not introduce the real label into the model, but Figure 4 explicitly input $y_{1:T}$ into the model, is this a drawing error? Or does the arrow from $y_{1:T}$ to the model mean something else?
> >
> > 4. I understand the main contribution of this article. However, this paper has repeatedly emphasized that dynamic factors and static factors are composed of causal factors and spurious factors, which brings confusion. In fact, SYNC does not make much design to distinguish causal factors from spurious factors.

---

> > > ### Author Response · Authors · 2025-04-03
> > >
> > > Thanks for your feedback, we will solve the remaining concerns as follows.
> > >
> > > > Q1. The design that ensures $Z^d$ and $Z^{dy}$ learn distinct information.
> > >
> > > R1. Thanks for your comment. Here we explain the modeling of $Z^d$ and $Z^{dy}$ in detail.
> > >
> > > $Z^d$ characterizes the temporal changes in causal factors' influence on the target. Therefore, $Z^d$ contains the structural information of category space and indirectly models the classifier's state changes over time. Specifically, In a $C$-class classification problem, $Z^d$ is modeled as a vector in $\mathbb{R}^C$. To learn the temporal evolution of $Z^d$, we develop a network $q_{\zeta}$ that takes the historical variables $z_{<t}^d$ and the one-hot vector of label $y_t$ as inputs and outputs the current state $z_t^d$. By optimizing $\mathcal{L}_{\text{mp}}$, we can constrain $Z^d$ to contain the category space structure information and learn the evolving pattern.
> > >
> > > Unlike $Z^d$, the dynamic factor $Z^{dy}\in \mathbb{R}^D$ contains feature space semantic information, where $D$ is the dimension of latent features. We develop a network $q_{\theta}$ to capture the evolving pattern of $Z^{dy}$. In our objectives, the reconstruction loss of data and labels preserves the semantic information of feature in $Z^{dy}$, while the KL divergence between $q_{\theta}$ and the prior $p(z_t^{dy}|z_{<t}^{dy})$ aids in learning evolving patterns.
> > >
> > > Overall, we design $q_{\zeta}$ and $q_{\theta}$ with specific losses to ensure that $Z^d$ learns the category space structure information, while $Z^{dy}$ retains the feature space semantic information. The results below show that $Z^d$ and $Z^{dy}$ learn different information.
> > >
> > > ||RMNIST|Portraits|Overall
> > > |-|-|-|-
> > > |SYNC|50.8|90.8|70.8
> > > |SYNC w/o $Z^d$|47.6|89.9|68.8
> > > |SYNC w/o $Z^{dy}$|47.1|89.6|68.4
> > >
> > > > Q2. About max-min optimization.
> > >
> > > R2. Thanks. Unlike adversarial training, which maximizes and minimizes the same loss across different parameters, our method optimizes **distinct loss functions**. Specifically, we minimize the mutual information (MI) loss $I(Z_t^{st},Z_t^{dy})$ to disentangle static and dynamic factors. The conditional MI loss $I(\Phi_c^{dy}(X_t);Z_{c,t}^{st}|Y)$ between dynamic causal factors and anchored static causal factors is maximized to extract the dynamic causal factors, which is implemented by minimizing the supervised contrastive loss. Therefore, our method performs max-min optimization on different losses.
> > >
> > > In addition, we clarify that these two losses update different sets of parameters. The MI loss updates the network $q_{\psi}$ and $q_{\theta}$, while the conditional MI loss above updates $\Phi_c^{dy}$, which includes the feature extraction component of $q_{\theta}$ and a MLP-based masker.
> > >
> > > > Q3. About $y_{1:T}$ in Figure 4.
> > >
> > > R3. Thanks for your response. This paper adopts a well-designed EDG setting [1-2], where data and labels from $T$ temporally ordered domains $\mathcal{D}\_t^{train}=\\{(x_{i,t},y_{i,t})\\}\_{i=1}^{N_t}$ are used during training, and predictions are made for future unlabeled domain $\mathcal{D}\_t^{test}=\\{x_{i,t} \\}\_{i=1}^{N_t}$ starting from $T+1$ during inference. Figure 4 illustrates the training process of SYNC, where the input $y_{1:T}$ is used to model the posterior network $q_{\zeta}$ to approximate the prior $p(z_t^d|z_{<t}^d)$, which is used to infer the drift factor during the test phase. In future versions, we will improve Figure 4 and its caption to clearly differentiate between the training and inference processes.
> > >
> > > [1] LSSAE, ICML’22.
> > >
> > > [2] SDE-EDG, ICLR’24.
> > >
> > > > Q4. The design to distinguish causal factors from spurious factors.
> > >
> > > R4. Thanks for your feedback. We clarify that our method is effectively designed to separate causal factors from spurious ones. It models static and dynamic factors in the time domain,  and then further dividing them into causal and spurious factors at a finer granularity. Namely, we have $Z^{st}=[Z_c^{st},Z_s^{st}]$ and $Z^{dy}=[Z_c^{dy},Z_s^{dy}]$.
> > >
> > > To differentiate $Z_c^{st}$ from $Z_s^{st}$ and $Z_c^{dy}$ from $Z_s^{dy}$, we design two maskers: $m_c^{st}$ and $m_c^{dy}$. For each masker $m_c^{\cdot}$ ($\cdot$ denotes “st” or “dy”), it takes factors $Z^{\cdot}$ as input and outputs a 0-1 mask. This mask is then element-wise multiplied with the corresponding $Z^{\cdot}$ to model causal factors. Specifically, using $q_{\psi}^{ext}$ and $q_{\theta}^{ext}$ to learn $Z^{st}$ and $Z^{dy}$, the static and dynamic causal factors are denoted as $\Phi_c^{st}(X)$ and $\Phi_c^{dy}(X)$, respectively, where $\Phi_c^{st}=m_c^{st}\circ q_{\psi}^{ext}$ and $\Phi_c^{dy}=m_c^{dy}\circ q_{\theta}^{ext}$.
> > >
> > > After that, Lemma 3.4 and Lemma 3.5 ensure that optimizing $\Phi_c^{st}$ and $\Phi_c^{dy}$ via Eq. (4) and Eq. (8) allows the model to extract causal factors from mixed factors, separating them from spurious factors.
> > >
> > > Finally, the results shown in R2 of Reviewer RWjv demonstrates that our method effectively learns both static and dynamic causal factors.

---

### Official Review · Reviewer_LQYB · 2025-03-10

**Overall Recommendation:** 3

**Summary:**

This paper addresses the challenge of generalizing deep models in evolving domains, where data distributions shift dynamically over time. The author claims that exiting Evolving Domain Generalization (EDG) approaches suffer from spurious correlations, which degrade their generalization ability. To mitigate this, the authors propose a time-aware Structural Causal Model (SCM) that explicitly models dynamic causal factors and causal mechanism drifts. They further introduce Static-DYNamic Causal Representation Learning (SYNC), an approach that integrates a sequential Variational Autoencoder (VAE) and information-theoretic objectives to learn time-aware causal representations. Experiments on synthetic and real-world datasets demonstrate the superiority of the proposed method over existing EDG techniques.

## update after rebuttal
I believe most of my concerns were solved by the authors during the rebuttal phase. So I tend to keep my score

**Claims And Evidence:**

Yes

**Essential References Not Discussed:**

I seem the authors attempted to discuss more on details, but there is still a lack of discussion, more details see questions listed.

**Experimental Designs Or Analyses:**

Kinds of, I would say the datasets and the tasks are set correctly. and the results of each experiment also show the effectiveness of the proposed method. However, some visualization can still be improved.

**Methods And Evaluation Criteria:**

Yes,  I would say the datasets and the tasks are set correctly.

**Other Comments Or Suggestions:**

Please check the questions above.

**Other Strengths And Weaknesses:**

Pros:
1.	The experiments are sufficient, which visible performance improvements.
2.	The method is technically sound, especially for considering the time factor of modelling causal reasoning. I think the idea is reasonable.
3.	The functions and theoretical proving seems good.

Cons:
1.	The idea of using time factor for causal reasoning is reasonable, but not being expressed very clear. The background of why time matters for building correlations is still not clear enough. Directly saying that the time factor may cause spurious correlations, is not so convincible. but I think there is still space to even talk deeper, with more concrete examples.
2.	The figure 1 can be improved with some day time images which can be recognized correctly, to show the difference between results of daytime and nighttime images. Then tell the trained model learned the spurious correlations.
3.	The idea of capturing the time factor during causal reasoning is not the first time, what is the main difference between the proposal modelling and the existing methods who claim they also using time factors to avoid spurious correlations?
4.	From the shown visualization results, there is potential overfitting to temporal trends, e.g., every image shows the prediction hot map just in front of the camera.  The model might learn domain-specific time factors rather than true causal mechanisms, since we can not really tell from an un-explainable DNN. This may lead to reduced robustness when faced with unseen distributions that deviate significantly from training trends. Do you have an idea of how to conquer this?

**Questions For Authors:**

Please check the questions above.

**Relation To Broader Scientific Literature:**

I think from what I know, and also the way the authors claimed, I think this method has a potential to be adapted for different/more extensive applications.

**Theoretical Claims:**

Yes, No clear mistakes are found in the proofs.

---

> ### Author Rebuttal · Authors · 2025-04-01
>
> > Q1. Further analysis of spurious correlation caused by time factors and more concrete examples.
>
> R1. Thanks. Herein, we provide a formal characterization of the spurious correlation problem. In the proposed time-aware SCM, the time factor constitutes a latent confounder whose components $G$ and $L$ establish backdoor paths between static and dynamic spurious factors and the label, namely $Z_s^{st} \leftarrow G \rightarrow Y$ and $Z_s^{dy} \leftarrow L \rightarrow Z^d \rightarrow Y$, thereby introducing spurious correlations. Under this configuration, naively maximizing mutual information between features and labels (that is, minimize the cross-entropy loss) will cause the model to learn spurious features. Specifically, the model may erroneously utilize brightness features with strong temporal correlations as discriminative cues for vehicle detection in Figure. 1, rather than learning essential shape semantics. To make the example more concrete, we provide more visualization results and analysis. Please refer to R2.
>
> > Q2. Improvement on visualization to tell the trained model learned the spurious correlations.
>
> R2. Thanks. To more clearly demonstrate the effectiveness of our method, we have enhanced the visualization and presented it in Fig. 2 of https://anonymous.4open.science/r/2150-1CBD. It can be found that in nighttime images, LSSAE [1] identifies the image as “No Car” based on lighting, while our method correctly focuses on the semantic information of the car, identifying it as “Car”. In daytime images, although LSSAE successfully classifies the image as car, it relies on environmental factors, whereas our method correctly identifies it based on the car's semantic information. Therefore, the trained model may learn the spurious correlations.
>
> [1] LSSAE, ICML 2022.
>
> > Q3. Comparison with methods using time factors to avoid spurious correlations.
>
> R3. Thanks. Time-series causal modeling constitutes the most pertinent research direction to this work. Most existing time series causal methods [2-4] construct temporal SCMs that incorporate causal factors and aim to learn causal representations based on the properties of these factors. However, due to the complexity of dynamic scenes, the behavior and nature of causal factors also become intricate, often requiring strong assumptions, such as the reversibility of the generation function and an additive noise model.
>
> In contrast to them, meticulous deliberation has been conducted regarding the majority of factors in our method, wherein causal variables are explicitly decomposed into static and dynamic components for joint modeling. This modeling approach allows our method to learn complete causal representations by first learning easily obtainable static causal factors by and using them as anchors to learn dynamic causal factors, without requiring stringent assumptions, thus endowing it with extensibility to more complex scenarios. Furthermore, the integration of causal mechanism drift enables better adaptation to the underlying data distribution.
>
> [2] Causal-HMM, CVPR’21.
>
> [3] TDRL, ICLR’22.
>
> [4] CtrlNS, NeurIPS’24.
>
> > Q4. Visualization improvements and the idea for mitigating significant deviations from the trend.
>
> R4. Thanks. In the displayed visualization results, the similarity in the hot map positions may be attributed to the fact that the vehicle's position is directly in front of the camera. We have improved the visualization, as shown in Fig. 2 of https://anonymous.4open.science/r/2150-1CBD.
>
> When faced with unseen distributions significantly deviating from the training trends, the performance of EDG methods inevitably deteriorates, as they are designed under the assumption of slow and regular distribution changes. However, our approach incorporates the learning of static causal factors, enabling the model to maintain relatively stable generalization capabilities. To illustrate this, we randomly reorder the test domains of RMNIST and evaluate different methods. Specifically, we keep the training and validation sets unchanged, while rearranging the domains in the original test set, which previously arrived sequentially from 130°, 140°, ..., 180°, into 170°, 140°, 180°, 160°, 130°, 150°. The results in the table show that our method still outperforms the baselines. In addition, techniques such as out-of-distribution detection can be used to identify moments of significant distribution shifts and adjust the model to mitigate the impact of abrupt and violent shifts.
>
> Finally, we also conduct an experiment on the more challenging FMoW dataset as shown in R2 of Reviewer GuDA. it can be found that our approach remains effective in generalizing in more challenging scenarios, exhibits the potential for deployment in complex real-world applications.
>
> || LSSAE | SDE-EDG | SYNC |
> | --- | --- | --- | --- |
> | RMNIST_Reorder (Wst./Avg.) | 35.7/40.8 | 28.1/42.0 | 38.1/43.4 |

---

> > ### Comment · Reviewer_LQYB · 2025-04-01
> >
> > I appreciate the authors' detailed responses and have reviewed both their replies and the feedback provided by other reviewers. Overall, my initial concerns have been largely addressed, although some issues remain unresolved. Nevertheless, I acknowledge the potential for these concerns to be adequately addressed in future revisions. Given that several points raised by other reviewers also warrant consideration, I am inclined to maintain my original scores at this stage.

---

> > > ### Author Response · Authors · 2025-04-02
> > >
> > > We greatly appreciate your feedback and are pleased that our rebuttal has largely addressed your initial concerns. We will incorporate your valuable suggestions into the manuscript in future revisions and remain open to any further inquiries or discussions that may arise.

---

### Official Review · Reviewer_RWjv · 2025-03-12

**Overall Recommendation:** 3

**Summary:**

This paper proposes a framework called Static-DYNamic Causal Representation Learning (SYNC) to deal with distributional drift in dynamic environments for generalization. By designing a time-aware Structural Causal Model (SCM), SYNC models dynamic causal factors and causal mechanism drifts, leveraging a sequential Variational Autoencoder (VAE) framework combined with information-theoretic objectives to learn time-aware causal representations. Theoretical analysis demonstrates that this method effectively mitigates spurious correlations and learns the optimal causal predictors for each time domain. Experimental results on synthetic and real-world datasets validate its superiority and broad applicability in dynamic, non-stationary environments.

## update after rebuttal
All my concerns have now been addressed. As previously suggested, I hope the authors can thoroughly revise the manuscript by adding a detailed description of the ablation study and carefully correcting the typos.

**Claims And Evidence:**

N/A

**Essential References Not Discussed:**

Yes

**Experimental Designs Or Analyses:**

N/A

**Methods And Evaluation Criteria:**

N/A

**Other Comments Or Suggestions:**

- In Line 189, left half of the page. I guess X and Y should be $X := f^x(Z^{st}_c,Z^{st}_s,Z^{dy}_c,Z^{dy}_s, \epsilon_x)$ $Y := f^y(Z^{st}_c,Z^{dy}_c,Z^d, \epsilon_y)$
- Lines 253-257, right half of the page. The maker should be $m^{st}_c$ and $\Phi^{st}_c = m^{st}_c \circ q^{ext}\_{\psi}$

**Other Strengths And Weaknesses:**

**Strength:**

1. The framework is conceptually solid and interesting. The introduced SCM module integrates both static and dynamic causal factors, expanding causal representation learning to dynamic, non-stationary environments.
2. This paper provides rigorous theoretical analysis, proving that SYNC can learn optimal causal predictors for each time domain and mitigate spurious correlations effectively.
3. Extensive experiments have been done on various synthetic and real-world datasets and the proposed method exhibits excellent generalization performance under various types of domain shifts.

**Weakness:**

1. The significance of Proposition 3.3 in learning dynamic causal representations is unclear. While (i) and (ii) seem reasonable here, what is the their connection towards the SCM?
2. Has an ablation study been conducted on the loss term $\mathcal{L}_{causal}$? What is detailed implementation of Variants A-D? I cannot find them in the main paper and appendix.
3. The manuscript contains numerous typographical errors, thus I recommend the author to revise their manuscript thorough to improve clarity and readability.

**Questions For Authors:**

N/A

**Relation To Broader Scientific Literature:**

No, just in AI research community.

**Theoretical Claims:**

N/A

---

> ### Author Rebuttal · Authors · 2025-04-01
>
> > Q1. The significance of Proposition 3.3 and its connection to SCM.
>
> R1. Thanks. Here we provide a detailed explanation of Proposition 3.3.
>
> **The Significance for learning causal representation:** The two points of Proposition 3.3 respectively guide the learning of static and dynamic causal representations.
>
> From Proposition 3.3 (i), it can be found that for various static factors in two consecutive temporal domains, the static causal factors $Z_c^{st}$ are those that maximize the conditional mutual information (CMI) under a given category. Since we extract static causal factors using network $\Phi_c^{st}$, static causal representations can be learned by maximizing the CMI $I(\Phi_c^{st}(X_t);\Phi_c^{st}(X_{t-1})|Y)$. Proposition 3.3 (ii) shows that among various dynamic factors within the same temporal domain, the dynamic causal factors $Z_c^{dy}$ are those that maximize the CMI with the static causal factors under a given category. Therefore, static causal factors $Z_c^{st}$ can serve as anchors to facilitate the learning of dynamic causal factors $Z_c^{dy}$ by maximizing $I(\Phi_c^{dy}(X_t);Z_{c,t}^{st}|Y)$.
>
> **Connection towards the SCM:** Although Proposition 3.3 can be derived straightforwardly using information theory, its inspiration actually comes from SCM and can be intuitively explained within the SCM framework. Since the two points of Proposition 3.3 are quite similar, for the sake of simplicity, we will explain using the second point. As shown in the time-aware SCM in Figure. 2, there is a clear collider structure and a fork structure, namely $Z_c^{st}\rightarrow Y \leftarrow Z_c^{dy}$ and $Z_s^{dy} \leftarrow L \rightarrow Z_c^{dy}$. According to d-separation, when conditioned on $Y$, both $Z_c^{dy}$ and $Z_s^{dy}$ are related to $Z_c^{st}$. Consider a boundary case where $Z_c^{dy}$ and $Z_s^{dy}$ are independent, this implies that the backdoor path between $Z_c^{dy}$ and $Z_s^{dy}$ is blocked, leading to independence between $Z_c^{st}$ and $Z_s^{dy}$, while $Z_c^{st}$ remains related to $Z_c^{dy}$. Proposition 3.3 generalizes this observation, under certain entropy inequalities, static causal factors are more strongly related to dynamic causal factors than dynamic spurious factors.
>
> > Q2. More details about ablation study.
>
> R2. Thanks. Here we explain the ablation study in detail and illustrate the contribution of the modeled causal factors to performance. As the table shown below, Variant A serves as the base method trained solely with evolving pattern loss $\mathcal{L}\_{\text{evolve}}$. Variant B builds upon the base method by additionally training with MI loss $\mathcal{L}\_{\text{MI}}$, serving as an ablation for $\mathcal{L}\_{\text{causal}}$. Variants C and D build upon Variant B by incorporating additional training with static causal loss $\mathcal{L}\_{\text{stc}}$ and dynamic causal loss $\mathcal{L}\_{\text{dyc}}$, respectively. We conducted additional ablation experiments on the Portraits dataset, excluding RMNIST, and recorded their worst-case performance (W) and average performance (A). The results are presented below.
>
> First, both Variant C and Variant D show performance improvements over Variant B, validating the effectiveness of modeling static and dynamic causal factors.
>
> Then, it is clear that Variant C achieves a greater improvement in worst-case performance compared to Variant D, indicating that static causal factors can ensure stable generalization under continuous distribution shifts. However, focusing solely on them ignores evolving pattern in EDG, limiting further generalization gains. Learning dynamic causal factors captures features related to task evolving over time, enabling to generalize better to the current distribution. Variant D outperforms Variant C in average performance and provides evidence for this claim.
>
> Finally, SYNC learns static and dynamic causal representations jointly, achieving the best performance, demonstrating their significant contribution to overall performance.
>
> ||| RMNIST (W/A) | Portraits (W/A) | Overall (W/A) |
> |-|-|-|-|-
> |Variant A | base | 40.5/44.1 | 78.1/89.2 | 59.3/66.7 |
> |Variant B | base+$\mathcal{L}\_{\text{MI}}$| 41.9/45.7 | 78.3/89.4 | 60.1/67.5 |
> |Variant C | base+$\mathcal{L}\_{\text{MI}}$+$\mathcal{L}\_{\text{stc}}$ | 44.1/48.7 | 79.8/89.9 | 62.0/69.3 |
> | Variant D | base+$\mathcal{L}\_{\text{MI}}$+$\mathcal{L}\_{\text{dyc}}$ | 42.9/49.3 | 79.1/90.4 | 61.0/69.8 |
> | SYNC | base+$\mathcal{L}\_{\text{MI}}$+$\mathcal{L}\_{\text{stc}}$+$\mathcal{L}\_{\text{dyc}}$ | 45.8/50.8 | 81.0/90.8 | 63.4/70.8 |
> > Q3. Typographical errors.
>
> R3. Thanks. As your suggestion, we will thoroughly revise the manuscript to improve clarity and readability.

---

### Official Review · Reviewer_GuDA · 2025-03-12

**Overall Recommendation:** 3

**Summary:**

This paper proposes SYNC, a method for improving temporal generalization in evolving domains by explicitly disentangling static and dynamic causal representations. It introduces a sequential variational autoencoder (VAE) with mutual information minimization constraints to separate static and dynamic causal factors. Experimental evaluations on synthetic and real-world datasets show that SYNC achieves better generalization performance than existing causal and non-causal domain generalization methods.

**Claims And Evidence:**

Claim: *SYNC achieves improved temporal generalization by modeling dynamic causal factors and causal mechanism drifts.*

This is partially supported. The experiments indeed show improvement over various baseline methods. However, results lack deeper analyses into why and how each causal factor contributes to performance.

Claim of optimal causal predictor: Theoretically argued, but practically the verification is incomplete and lacks rigorous validation.

**Essential References Not Discussed:**

"*Continuous Temporal Domain Generalization.*" Zekun Cai, Guangji Bai, Renhe Jiang, Xuan Song, Liang Zhao. NeurIPS 2024.

**Experimental Designs Or Analyses:**

1. Datasets selected (Circle, Sine, RMNIST, Caltran) are standard but somewhat simplistic or dated. How representative are these of realistic, high-dimensional, complex evolving domain generalization scenarios? For instance, how does the proposed method perform on WILDS benchmarks from Stanford (https://wilds.stanford.edu/datasets/#fmow).

2. Limited exploration of scalability (the proposed method's overhead, complexity, and memory cost are largely omitted). There is only one relatively limited comparison of computational cost in the appendix currently.

**Methods And Evaluation Criteria:**

1. The current evaluation is primarily accuracy-based. Given the goal is generalization in evolving domains, metrics reflecting robustness (such as performance variance across evolving domains or robustness metrics under shifts) should also be considered.

2. Good coverage of existing EDG and DG methods, but lacks a detailed comparison with e.g., transformer-based methods [1,2].

[1] Domain Transformer: Predicting Samples of Unseen, Future Domains (2021)

[2] Vision Transformers in Domain Adaptation and Domain Generalization (2024)

**Other Comments Or Suggestions:**

Please refer to my comments above. My final score will depend on other reviewers' comments and rebuttal discussion, and I will adjust my score accordingly.

**Other Strengths And Weaknesses:**

S1. Clear methodological motivation, novel integration of causal modeling into evolving domains.

S2. The theoretical analyses round up a rigorous paper.

S2. Promising empirical performance improvements.

W1. Over-complex method without sufficient empirical or theoretical justification for complexity.

W2. Insufficient practical robustness checks (e.g., sensitivity to hyperparameters, data complexity).

W3. Limited clarity and depth in experimental analysis, especially regarding scalability and real-world application feasibility.

**Questions For Authors:**

1. How robust is SYNC to violations of SCM assumptions, specifically regarding causal factor disentanglement and mechanism drifts?

2. What is the computational complexity of SYNC, particularly the scalability of sequential VAE and MI computations?

3. Could you also compare with the transformer-based DG methods given their relevance?

4. Could you empirically evaluate the proposed method and other methods on more real-world benchmarks such as WILDS?

**Relation To Broader Scientific Literature:**

The paper relates well to domain generalization and causal representation learning literature, clearly distinguishing itself from static causal DG methods.

**Theoretical Claims:**

In this paper, the author presents two theoretical claims. Optimality of causal predictor for each domain is theoretically stated, but practical relevance and conditions for this optimality (e.g., assumptions like SCM correctness, Markov condition, faithfulness) remain unvalidated empirically. How sensitive is the method to violations of SCM assumptions?

---

> ### Author Rebuttal · Authors · 2025-04-01
>
> > Q1. The contribution of causal factors to performance.
>
> R1. Thanks. We perform a further analysis of causal factors. Please refer to R2 of Reviewer RWjv for details.
>
> > Q2. Model assumptions and their impact on performance.
>
> R2. Thanks. Given the challenging of the EDG problem, most existing methods make appropriate assumptions to derive their objectives. To name a few, LSSAE [1] assumes the latent variable's Markov property, while DDA [2] assumes consistent evolution across domains. Our approach adopts a causal perspective, relying only on the fundamental SCM assumptions like Markov condition, to build a reasonable SCM for modeling time causal factors. These widely used assumptions have been validated in real-world scenarios in literature [3-5]. Besides, our method generally outperforms other methods on real-world datasets in our paper, supporting the assumptions.
>
> Since causal sufficiency and faithfulness are relatively important in our assumptions (violating causal sufficiency disrupts static-dynamic independence, while violating faithfulness affects conditional dependencies), we conduct experiments to evaluate their impact. Results below show little performance change. Moreover, incorporating these relaxed assumptions improves performance.
>
> Finally, we conducted experiments on the more challenging FMoW. The results below show that SYNC outperforms others, demonstrating its applicability in more complex scenarios.
>
> ||RMNIST|Portraits
> |-|-|-
> |SYNC|49.6|89.9
> |SYNC w/o sufficiency|48.9|89.2
> |SYNC w/o faithfulness|50.8|90.8
>
> ||LSSAE|SDE-EDG|SYNC
> |-|-|-|-
> |FMoW|42.8|44.2|46.6
>
> [1] LSSAE, ICML’22.
>
> [2] DDA, AAAI’23.
>
> [3] MatchDG, ICML’21.
>
> [4] CIRL, CVPR’22.
>
> [5] iDAG, ICCV’23.
>
> > Q3. Comparison with transformer-based DG methods.
>
> R3. Thanks. We discuss transformer-based DG methods here. [5] generates data from unseen domains using CycleGAN, while [6] investigates the deployment of vision transformers (ViT) in DA and DG. However, neither addresses the DG using transformers. Here we compare SYNC with two highly cited transformer-based DG methods, i.e., DoPrompt [5] and GMoE [6]. While ViT helps capture robust semantic features, they fail to consider the continuous structure in EDG. The results below show SYNC's superior temporal generalization.
>
> ||GMoE|DoPrompt|SYNC
> |-|-|-|-
> |Portraits|87.9|88.2|90.8
> |Caltran|70.1|70.4|72.2
>
> [5] DoTra, IJCNN’22.
>
> [6] Vision...Generalization, Neural Comput Appl’24.
>
> [7] DoPrompt, arXiv.
>
> [8] GMoE, ICLR’23.
>
> > Q4. Evaluation on more real-world benchmarks.
>
> R4. Thanks. We evaluate our method and baselines on FMoW and the results are detailed in R2. It is known that SYNC remains effective in generalizing in more challenging scenarios, exhibits the potential for deployment in complex real-world applications.
>
> > Q5. Computational complexity and scalability analysis.
>
> R5. Thanks. Our network largely follows LSSAE [1] and MMD-LSAE [9], with the only addition being two MLP-based maskers. Similar to them, the time complexity and memory complexity of sequential VAE are $\mathcal{O}(T\cdot B\cdot [\sum\_{l=1}^LH^{l}W^{l}C^{l-1}C^l(K^l)^2+D^2])$ and $\mathcal{O}(T\cdot B\cdot [\sum\_{l=1}^LH^lW^lC^l+D])$ respectively, where $T$ is the number of time domains, $B$ represents batch size, $D$ is the dimension of latent features. For the
> $l$-th layer of the decoder, $H^l$ and $W^l$ denote the output feature map size, $C^l$ represents the output channel, and $K^l$ denotes the kernel size. For the MI loss function, the time complexity and memory complexity are $\mathcal{O}(T\cdot B^2\cdot (D+1))$ and $\mathcal{O}(T\cdot B\cdot D)$, respectively. In the implementation, $D$ and $K$ are set to a relatively small value ($D=32, K=5$) and the complexity is acceptable.
>
> Besides, we conduct an experiment on FMoW using DenseNet-121. The results show the effectiveness of our approach in challenging real-world scenarios. Additionally, we record memory usage and runtime per iteration. The results below show that our method requires almost the same cost to achieve better performance.
>
> ||LSSAE|MMD-LSAE|GI|SYNC
> |-|-|-|-|-
> |FMoW|35.2G/0.79s|35.3G/0.80s|28.6G/12.6s|35.3G/0.87s
>
> [9] MMD-LSAE, TPAMI'23.
> > Q6. Comparison with Koodos [10].
>
> R6. Thanks. Koodos considers the continuous temporal DG and models the evolving pattern via Koopman theory. Our method is tailored to the EDG and enhance temporal generalization by tackling spurious correlations through time-aware causal representations. We evaluate Koodos on two datasets and the results below verify the effectiveness of SYNC.
>
> ||Circle|RMNIST|
> |-|-|-
> |Koodos|81.4|44.6
> |SYNC|84.7|50.8
>
> [10] Koodos, NeurIPS’24.
>
> > Q7. Practical robustness checks.
>
> R7. Thanks. We conduct a sensitivity analysis experiment on $\alpha\_1$ and $\alpha\_2$, the results are shown in Fig. 1 of https://anonymous.4open.science/r/2150-1CBD, indicating insensitivity within a certain range. Additionally, our evaluation on FMoW shows the method's adaptability to more complex scenarios.

---

> > ### Comment · Reviewer_GuDA · 2025-04-04
> >
> > Thanks to the authors for the rebuttal. I’ve read through your responses as well as the other reviewers' comments. Most of my main concerns, especially regarding the empirical validation and efficiency of the proposed method, have been addressed.
> > Given the clarifications and the effort put into the rebuttal, I’m increasing my score from 2 to 3.

---

> > > ### Author Response · Authors · 2025-04-04
> > >
> > > Thank you very much for your detailed feedback and valuable suggestions. We are pleased to hear that our responses have addressed your main concerns. In future revisions, we will carefully incorporate your suggestions into the manuscript. Once again, we sincerely appreciate your thoughtful and constructive engagement with our work.

---

### Decision · Program_Chairs · 2025-05-01

**Decision:**

Accept (poster)

**Comment:**

This paper introduces one method for improving temporal generalization in evolving domains by explicitly disentangling static and dynamic causal representations. The authors have provided comprehensive responses to the reviewers' concerns through effective revisions that significantly enhance the paper's clarity and argumentative depth. After revision, the paper provided sufficient experiments and good visible performance. Overall, considering the core strengths and drawbacks of the proposed method, I recommend accepting this paper for publication.